# Exchangeability of GNN Representations with Applications to Graph Retrieval

**Kartik Nair**[*]
Carnegie Mellon University
ksnair@cs.cmu.edu

**Indradyumna Roy**
IIT Bombay
indraroy15@cse.iitb.ac.in

**Soumen Chakrabarti**
IIT Bombay
soumen@cse.iitb.ac.in

**Anirban Dasgupta**
IIT Gandhinagar
anirbandg@iitgn.ac.in

**Abir De**
IIT Bombay
abir@cse.iitb.ac.in

## Abstract

We discover a probabilistic symmetry, called exchangeability, in graph neural networks (GNNs). Specifically, we show that the trained node embedding computed using a large family of graph neural networks, learned under standard optimization tools, are exchangeable random variables. This implies that the probability density of the node embeddings remains invariant with respect to a permutation applied on their dimension axis. This results in identical distribution across the elements of the graph representations. Such a property enables approximation of transportation-based graph similarities by Euclidean similarities between the sorted embedding elements in fixed dimension. This allows us to propose a unified locality-sensitive hashing (LSH) framework that supports diverse relevance measures for graphs. Experiments show that our method provides more effective LSH than baselines. Code can be found at https://rebrand.ly/graphhash.

## 1 Introduction

In their seminal work, Hecht-Nielsen (1990) first demonstrated that the output of multi-layer perceptrons (MLPs) remains invariant under suitable permutations of the weight matrices across layers. Since then, such weight-space symmetries have been widely recognized, and have resurfaced with the advent of deep learning (Neyshabur et al., 2015b; Freeman et al., 2016; Brea et al., 2019). Recent works (Bui Thi Mai et al., 2020; Godfrey et al., 2022) characterized such symmetries for different activation functions. Beyond academic interest, weight space symmetries underpin several practical advances: for example, they enhance model training (Neyshabur et al., 2015b), equivariant architecture design (Cohen et al., 2016; Maron et al., 2019; Navon et al., 2023), enable model merging (Peña et al., 2022; Ainsworth et al., 2022), motivate data augmentation (Schürholt et al., 2021), *etc*. They also yield deeper characterizations of geometry and loss landscapes (Brea et al., 2019; Simsek et al., 2021; Entezari et al., 2021). These works focus on algebraic symmetry, and treat them in isolation from training. This leaves unaddressed the probabilistic symmetry structures in GNNs that emerge naturally during standard training, starting with random model initialization.

**Our contributions**  We aspire for a broad understanding of exchangeability in a wide variety of message passing graph neural network (GNN) topologies. This is of particular interest to us because of the recent research focus on locality-sensitive hashing, indexing, and scalable graph retrieval using such neural representations (Roy et al., 2022; Chakraborty et al., 2025). Some specific cases of exchangeability in multi-layer perceptrons and CNNs have been reported independently in contemporaneous work (Yi et al., 2025a), but without connections to hashing and search.

—*Characterization of exchangeability:*  We establish a new property of GNNs: under standard conditions, the elements of node embeddings computed by a trained GNN are exchangeable random variables, where the randomness is induced by the initialization of model parameters. Let $\boldsymbol{x}(u) \in \mathbb{R}^D$ denote the embedding of node $u$, produced by a trained GNN. Then, the joint distribution of its components $\boldsymbol{x}(u)[1], \ldots, \boldsymbol{x}(u)[D]$ is invariant under any permutation of the embedding dimensions $d \in [D]$. This has a significant consequence: the components $\boldsymbol{x}(u)[1], \ldots, \boldsymbol{x}(u)[D]$ are identically distributed random variables. Therefore, when averaged across multiple random seeds, the expected embedding matrix $\mathbb{E}[[\boldsymbol{x}(u)]_{u \in V}]$ collapses to a rank one matrix.

---

[*]Work done while at IIT Bombay

We would like to highlight that, we show such exchangeability holds for a wide spectrum of GNNs and graph transformers; and several optimizers, *e.g.*, SGD, Adam. In view of GNNs' known propensity for spatial oversmoothing (Roth et al., 2024) and recent discoveries of output rank collapse of transformers (Dong et al., 2023; Naderi et al., 2025), and sequential state space models (Joseph et al., 2025), this result is of independent interest.

—*Applications to graph retrieval:* In neural graph retrieval, the goal is to find corpus graphs $C = \{G_c\}$ most relevant to a query graph $G_q$. Recent studies (Jain et al., 2024; Zhuo et al., 2022; Fey et al., 2020) make it clear that transportation-based relevance distance between node embeddings performs significantly better than single-vector aggregation and graph kernels (Roy et al., 2022; Zhuo et al., 2022). Exchangeability enables efficient graph retrieval in two steps:

**(1)** Approximating transportation similarity with 1-D Euclidean approximations: Consider embeddings in one dimension ($D = 1$). In this case, the transportation distance between two sets can be solved exactly by sorting the points in each set and matching them in order. For example, suppose we use a GNN to produce one-dimensional embeddings $x(u) \in \mathbb{R}$. Then, given two graphs $G_q$ and $G_c$, each with $n$ nodes, the transportation distance between their embedding sets is $\text{Transport}(\{x^{(q)}(u)\}, \{x^{(c)}(u)\}) = \|\text{SORT}(\{x^{(q)}(u)\}) - \text{SORT}(\{x^{(c)}(u)\})\|$. In higher dimensions ($D > 1$), however, computing transportation-based distance (or transportation-based similarity) is substantially more complex, with exact algorithms scaling for $n$ nodes as $O(n^3)$ and often requiring $O(n^2)$ approximations such as Sinkhorn iterations. Exchangeability provides a way around this: since embedding coordinates are identically distributed, each dimension yields a concentrated estimate of the underlying transportation-based similarity. Instead of solving the full high-dimensional transportation-based similarity, we approximate it by aggregating $D$ simple Euclidean similarities across dimensions, thereby reducing "transportation distance between high dimensional vector sets" to an estimate based on per-dimension sorted orders, which is more amenable to indexing.

**(2)** Locality sensitive hashing (LSH) for graphs: LSH enables sublinear-time retrieval by hashing similar objects into the same bucket (Gionis et al.; Indyk et al., 1998; Charikar, 2002). Exchangeability lets us approximate costly transportation-based similarity with simple Euclidean similarity across embedding dimensions, making existing LSH schemes directly applicable. Notably, LSH for asymmetric transportation-based similarity has remained unexplored; our approximation provides the first principled approach, leveraging Roy et al. (2023). This yields a unified LSH framework that supports diverse graph relevance measures, from subgraph matching to graph edit distance with general costs.

## 2 PRELIMINARIES

**Notation** For a graph $G = (V, E)$, we denote $\boldsymbol{A}$ as its $n \times n$ adjacency matrix. We write $[\cdot]_+ = \max\{\cdot, 0\}$ as the hinge or ReLU function, $\mathcal{P}_n$ as the set of $n \times n$ permutation matrices and $[n] = \{1, .., n\}$ for any integer $n$. We denote $\boldsymbol{P}$ and $\boldsymbol{\pi}$ to indicate $n$ and $D$ dimensional permutation matrices, respectively, which are applied on the nodes and their embedding vectors respectively. $[\![\bullet]\!] \in \{0, 1\}$ is indicator function. In the context of graph retrieval, we denote a query graph as $G_q$, a corpus graph as $G_c$ with $|V_q| = |V_c| = n$ after padding with suitable number of nodes; and, the set of corpus graphs as $C$. We also use $\boldsymbol{A}_q$ and $\boldsymbol{A}_c$ to denote their $n \times n$ adjacency matrices. We use $p(\cdot)$ to denote the density of any random variable. Given a group $\mathcal{G}$, a function $f$ is $\mathcal{G}$-equivariant ($\mathcal{G}$-invariant) if $f(g\boldsymbol{x}) = gf(\boldsymbol{x})$ (resp., $f(g\boldsymbol{x}) = f(\boldsymbol{x})$) for all $g \in \mathcal{G}$.

**Node embedding computation using GNN** Given the number of message passing steps (or layers) $K$ and the dimension of node embeddings $D$, a graph neural network (GNN$_\theta$) computes node embeddings $\boldsymbol{x}_k(u) = \text{GNN}_\theta(G) \in \mathbb{R}^D$ for $u \in V$ using $K$ message passing steps. For brevity, we drop $K$ to write $\boldsymbol{x}(u) = \boldsymbol{x}_K(u)$. We compute the embedding matrices $\boldsymbol{X} \in \mathbb{R}^{n \times D}$ as $\boldsymbol{X} = [\boldsymbol{x}(u)]_{u \in [n]}$. $\boldsymbol{X}[:, d] \in \mathbb{R}^n$ denotes the $d$-th column of $\boldsymbol{X}$. The operator $\text{SORT}(\cdot)$ sorts an input vector in decreasing order. In the context of graph retrieval, we denote $\boldsymbol{x}^{(q)}(u)$ and $\boldsymbol{x}^{(c)}(u')$ to denote embeddings of node $u \in [n]$ and $u' \in [n]$ in the query and corpus graphs $G_q$ and $G_c$, respectively. Similarly, we use $\boldsymbol{X}^{(q)} = [\boldsymbol{x}^{(q)}(u)]_{u \in [n]} \in \mathbb{R}^{n \times D}$ and $\boldsymbol{X}^{(c)} = [\boldsymbol{x}^{(c)}(u')]_{u' \in [n]} \in \mathbb{R}^{n \times D}$ to denote the embedding matrices.

The parameters $\theta$ of the GNN are learned by minimizing a task specific loss function, which we denote as $\text{loss}(\theta)$. We assume that weights in $\theta$ are initialized via iid sampling from popular distributions, and then some popular gradient-based update recipes are used for training.

**Exchangeability**   Exchangeability implies that the joint density of the elements within a vector is permutation invariant with respect to the ordering of the elements.

**Definition 1** (Exchangeability (Aldous, 1985))**.** *Let $\boldsymbol{Y}_d \in \mathbb{R}^n$ be random vectors for $d \in [D]$. We say $\boldsymbol{Y}_1, ..., \boldsymbol{Y}_D$ are* exchangeable*, if for all permutations $\pi : [D] \to [D]$, the probability density functions of the sequence of vectors $\{\boldsymbol{Y}_1, ..., \boldsymbol{Y}_D\}$ is the same as that of $\{\boldsymbol{Y}_{\pi(1)}, ..., \boldsymbol{Y}_{\pi(D)}\}$, i.e.,*
$$p_{\boldsymbol{Y}_1,...,\boldsymbol{Y}_D}(\boldsymbol{y}_1, ...\boldsymbol{y}_D) = p_{\boldsymbol{Y}_{\pi(1)},...,\boldsymbol{Y}_{\pi(D)}}(\boldsymbol{y}_1, ...\boldsymbol{y}_D) \text{ for all realizations: } \boldsymbol{Y}_d = \boldsymbol{y}_d \text{ for } d \in [D].$$

**Order statistics**   For a vector $\boldsymbol{a}$, we denote its order statistics by $\text{SORT}(\boldsymbol{a})$, obtained by sorting its entries in decreasing order. For the node embedding matrix $\boldsymbol{X}$, we will frequently use $\text{SORT}(\boldsymbol{X}[:, d])$—the order statistics of the $d$-th embedding dimension across all nodes.

**Overview of our analysis**   **(1)** Distinct from algebraic symmetry, we characterize a new type of probabilistic symmetry in the node embeddings $\boldsymbol{X}$ of a graph $G$, which is computed using a trained GNN starting with random model initialization. Specifically, we show that $\boldsymbol{X}[:, 1], ..., \boldsymbol{X}[:, D]$ are exchangeable random variables, where the randomness is induced by the initialization of the model. **(2)** Given a query–corpus graph pair $(G_q, G_c)$, we exploit this property to approximate the transportation-based similarity between $\boldsymbol{X}^{(q)}$ and $\boldsymbol{X}^{(c)}$ using Euclidean similarity between the order statistics $\text{SORT}(\boldsymbol{X}^{(q)}[:, d])$ and $\text{SORT}(\boldsymbol{X}^{(c)}[:, d])$ for $d \in [D]$. **(3)** Building upon the proposal of Roy et al. (2023), we develop a unified LSH (Charikar, 2002) method for several graph relevance measures using the Fourier transform on the order statistics vectors. We further show that the resulting algorithm is a valid LSH for the original transportation-based graph similarity.

## 3   EXCHANGEABILITY OF GNN REPRESENTATIONS

In this section, we characterize the probabilistic symmetry of node representations, explicitly incorporating the effect of model training. Specifically, given the node representation matrix $\boldsymbol{X} = [\boldsymbol{x}(u)]_{u \in [n]} \in \mathbb{R}^{n \times D} = \text{GNN}_\theta(G)$, we show that $\boldsymbol{X}[:, 1], ..., \boldsymbol{X}[:, D]$ are exchangeable random variables (Definition 1) across the axis of the embedding dimension, where $\boldsymbol{X}[:, d] = [\boldsymbol{x}(u)[d]]_{u \in [n]}$. We first describe the setting for our analysis, followed by a high level explanation on why exchangeability will hold. Finally, we present the formal characterization.

### 3.1   SETTING

We provide the four components of our settings. We emphasize that they are presented primarily for technical completeness. They are not restrictive and, in fact, capture a broad class of settings.

**(1) Broad class of GNN architectures**   We consider the a wide variety of GNN architectures, which are listed in Appendix F. This list includes gated GNN (Gilmer et al., 2017), GIN (Xu et al., 2019), GAT (Veličković et al., 2018), GCN (Kipf et al., 2017). Our analysis is likely to extend beyond these cases, and also applies to graph transformers (Appendix F).

**(2) IID initialization of the parameters within a layer**   The entries of the parameter matrix within each layer are initialized in an i.i.d manner. This covers standard model initialization schemes, including Kaiming (He et al., 2015) and Xavier initialization (Glorot et al., 2010).

**(3) Permutation invariance of loss function**   We consider loss functions that are invariant to permutations of elements in the node embedding vectors. This condition holds naturally in several settings, including graph retrieval. Here, the loss, whether binary cross-entropy or pairwise ranking, depends on the similarity between $(G_q, G_c)$ via the transportation plan between $\boldsymbol{X}^{(q)}$ and $\boldsymbol{X}^{(c)}$. Since this similarity is invariant under permutations of embedding elements, the loss is likewise permutation-invariant. This also applies to link prediction, when the similarity between nodes $u$ and $v$ is computed as the dot product $\boldsymbol{x}(u)^\top \boldsymbol{x}(v)$, which is permutation invariant w.r.t. elements of $\boldsymbol{x}$.

**(4) Broad class of optimizers**   Our results hold for a broad class of gradient-based optimizers, *viz.*, SGD (Zhang, 2004), Adam (Kingma et al., 2015), *etc*.

### 3.2   WHY EXCHANGEABILITY HOLDS: A HIGH LEVEL EXPLANATION

**Exchangeability among initialized model parameters**   Training begins with i.i.d. initialization of the parameter matrices. Formally, consider a weight matrix $\boldsymbol{\Theta}$ whose entries are drawn i.i.d. from a common distribution. Its joint distribution is invariant to column permutations: for any permutation matrix $\boldsymbol{\pi}$, $p(\boldsymbol{\Theta}) = p(\boldsymbol{\Theta}\boldsymbol{\pi})$. When $\boldsymbol{\Theta}$ is applied to an input row vector $\boldsymbol{x}$, the output $\boldsymbol{x}' = \boldsymbol{x}\boldsymbol{\Theta}$ is *equivariant* to column permutations of $\boldsymbol{\Theta}$: $\boldsymbol{\Theta} \mapsto \boldsymbol{\Theta}\boldsymbol{\pi} \implies \boldsymbol{x}' \mapsto \boldsymbol{x}'\boldsymbol{\pi}$. Although permuting $\boldsymbol{\Theta}$

changes the values of $\boldsymbol{x}'$, an i.i.d. initialization ensures that all permutations are equally likely, so the distribution of $\boldsymbol{x}'$ is invariant: $p(\boldsymbol{x}') = p(\boldsymbol{x}'\boldsymbol{\pi})$. This statistical symmetry is precisely what we mean by exchangeability of hidden units at initialization. Nonlinear activations $\sigma$, such as sigmoid or $\tanh$, being identical and applied pointwise, preserve this symmetry.

**Exchangeability in MLP Training**   Consider a two-layer MLP with weights $\boldsymbol{\Psi}, \boldsymbol{\Theta}$ and nonlinear activations $\sigma$, which maps an input row feature vector $\mathbf{feat}$ to an output representation $\boldsymbol{x}$ via $\boldsymbol{x} = \sigma(\sigma(\mathbf{feat}\,\boldsymbol{\Psi})\boldsymbol{\Theta})$. As discussed, at initialization ($t = 0$), exchangeability holds by construction: the entries of $\boldsymbol{\Theta}_0$ ($\boldsymbol{\Theta}$ at $t = 0$) are i.i.d., so $p(\boldsymbol{\Theta}_0) = p(\boldsymbol{\Theta}_0\boldsymbol{\pi})$, and consequently $p(\boldsymbol{x}) = p(\boldsymbol{x}\boldsymbol{\pi})$. As noted in Section 3.1 (3), the loss function is invariant to permutations of the embedding dimensions. With all other randomness fixed by seeding, permuting the columns of $\boldsymbol{\Theta}_0$ yields identical losses and hence equivariant gradients. Consequently, the training trajectories are permutation-equivariant: for any $\boldsymbol{\pi}$, $\boldsymbol{\Theta}_0 \mapsto \boldsymbol{\Theta}_0\boldsymbol{\pi} \implies \boldsymbol{\Theta}_t \mapsto \boldsymbol{\Theta}_t\boldsymbol{\pi}$ for all epochs $t$. Combining $p(\boldsymbol{\Theta}_0) = p(\boldsymbol{\Theta}_0\boldsymbol{\pi})$ at initialization, with permutation-equivariant training dynamics, we obtain $p(\boldsymbol{\Theta}_t) = p(\boldsymbol{\Theta}_t\boldsymbol{\pi})$ and hence $p(\boldsymbol{x}) = p(\boldsymbol{x}\boldsymbol{\pi})$ for all $t \geq 0$.

### 3.3   FORMAL CHARACTERIZATION OF EXCHANGEABILITY

**Overview**   Here, we seek to establish the afore-mentioned arguments for GNN to prove the exchangeability of the elements of the node embeddings. We prove this using four steps:

**(1)** Permutation induced parameter transformation on GNN (Lemma 2):   Given $\text{GNN}_\theta$ with parameter set $\theta$, consider any permutation $\boldsymbol{\pi} \in \mathcal{P}_D$. We show that there exists a bijective transformation $\Gamma_{\boldsymbol{\pi}}$ on $\theta$ such that, for $\theta' = \Gamma_{\boldsymbol{\pi}}(\theta)$, the elements of the node embeddings are permuted by $\boldsymbol{\pi}$, *i.e.*, $\boldsymbol{X} \mapsto \boldsymbol{X}\boldsymbol{\pi}$. We refer to $\Gamma_{\boldsymbol{\pi}}$ as a permutation-inducing transformation corresponding to $\boldsymbol{\pi}$.

**(2)** Gradient equivariance (Lemma 3):   We show that the gradient of loss is equivariant with respect to a permutation inducing transformation $\Gamma_{\boldsymbol{\pi}}$.

**(3)** Invariance of the probability density of model parameters (Lemma 4):   We show that at any stage of training, the model parameters are exchangeable— the probability density of the parameters $\theta$ remains invariant to the transformation $\Gamma_{\boldsymbol{\pi}}$.

**(4)** Result on exchangeability (Theorem 5):   Using (1–3), we show that $\boldsymbol{X}[:, 1], .. \boldsymbol{X}[:, D]$ are exchangeable.

**Warmup: Constructing $\Gamma_{\boldsymbol{\pi}}$ for 2-layer MLP**   We are given an MLP of the form $\boldsymbol{x} = \sigma(\sigma(\mathbf{feat}\,\boldsymbol{\Psi})\boldsymbol{\Theta})$. If we want to reorder $\boldsymbol{x}$ by a given permutation $\boldsymbol{\pi}$, we will transform $\boldsymbol{\Theta} \mapsto \boldsymbol{\Theta}\boldsymbol{\pi}$, which will result in $\boldsymbol{x} \mapsto \boldsymbol{x}\boldsymbol{\pi}$. Equivalently, suppose we write $\theta = [\boldsymbol{\Psi}^\top, \boldsymbol{\Theta}]$, then, we can introduce a bijection $\Gamma_{\boldsymbol{\pi}}$ by $\Gamma_{\boldsymbol{\pi}}(\theta) := \theta \, \text{Diag}(\mathbb{I}, \boldsymbol{\pi})$, which will result in output equivariance $\boldsymbol{x} \mapsto \boldsymbol{x}\boldsymbol{\pi}$.

**Permutation induced parameter transformation on GNN**   Constructing a similar transformation $\Gamma_{\boldsymbol{\pi}}$ is more involved for GNNs. The difficulty stems from the iterative message passing protocol: permutations of parameters in one layer propagate through neighborhood aggregations, which can entangle the symmetry across layers and makes it hard to identify $\Gamma_{\boldsymbol{\pi}}$ for popular GNNs, *e.g.*, gated GNN (Li et al., 2016), (Gilmer et al., 2017) which is widely used in graph retrieval (Li et al., 2019; Roy et al., 2022; Jain et al., 2024). Nevertheless, in the following, we formally establish that such transformations can indeed be derived for GNNs (proven in Appendix E).

**Lemma 2.** *Given a graph $G$ and a GNN architecture $\text{GNN}_\theta$ described in Appendix F, let the node embedding matrix of $G$ be $\boldsymbol{X} = \text{GNN}_\theta(G) \in \mathbb{R}^{n \times D}$. Then, for any permutation matrix $\boldsymbol{\pi} \in \mathcal{P}_D$, there exists a bijective transformation $\Gamma_{\boldsymbol{\pi}}$ with $|\text{Det}\,(\partial\Gamma_{\boldsymbol{\pi}}(\theta)/\partial\theta)| = 1$ such that $\boldsymbol{X}\boldsymbol{\pi} = \text{GNN}_{\Gamma_{\boldsymbol{\pi}}(\theta)}(G)$. We call $\Gamma_{\boldsymbol{\pi}}$ a* model transformation induced by permutation $\boldsymbol{\pi}$.

Given this characterization, we seek to reduce the problem of establishing exchangeability to establishing probabilistic symmetries in the model parameters $\theta$ with respect to the transformation $\Gamma_{\boldsymbol{\pi}}$.

**Equivariance of gradient under permutation induced parameter transformation**   Since the loss function is invariant to any permutation $\pi$ of the node embeddings, it is also invariant to the transformation $\Gamma_{\boldsymbol{\pi}}$ on $\theta$ (Lemma 2). As a result, the corresponding loss landscape exhibits symmetry under $\Gamma_{\boldsymbol{\pi}}$. This symmetry, in turn, implies an equivariance property for the gradient, as formalized below (proven in Appendix E).

**Lemma 3** (Gradient equivariance). *Given the setting described in Section 3.1. Let $\Gamma_{\boldsymbol{\pi}}$ be the transformation on the GNN parameters $\theta$, induced by a permutation $\boldsymbol{\pi}$, as introduced in Lemma 2. We denote the loss function as $\text{loss}(\theta)$. Then the gradient of the loss $\nabla_\theta \text{loss}(\theta)$ is equivariant under transformation $\Gamma_{\boldsymbol{\pi}}$ of the parameters $\theta$.*

**Invariance of probability density of model parameters under the transformation** $\Gamma_{\boldsymbol{\pi}}$   Suppose we shuffle the initial parameters within a layer. Then, from the gradient equivariance property (Lemma 3) the resultant trajectory $\{\theta_t \,|\, t \geq 0\}$ of $\theta$ at different epochs $t$, will undergo an equivariant transformation with respect to a permutation-induced bijection $\Gamma_{\boldsymbol{\pi}}$. Since $p(\theta_0) = p(\Gamma_{\boldsymbol{\pi}}(\theta_0))$, the observation will lead to invariance of the probability density of $\theta_t$ for $t \geq 0$ too, as stated below (proven in Appendix E).

**Lemma 4** (Invariance of density of $\Gamma_{\boldsymbol{\pi}}(\theta)$). *Given the setting described in Section 3.1. Let $\{\theta_t \,|\, t \geq 0\}$ be the trajectory of the parameter $\theta$ of a GNN across different training epochs $t \geq 0$. Then, we have: $p(\theta_t) = p(\Gamma_{\boldsymbol{\pi}}(\theta_t))$ for all $t \geq 0$.*

**Key results on exchangeability**   Using Lemmas 2–4, we can show our key exchangeability results, stated as follows (proven in Appendix E).

**Theorem 5** (Exchangeability of embedding elements). *Given the setting described in Section 3.1. Then, $\boldsymbol{X} = \mathrm{GNN}_\theta(G)$ are exchangeable random variables, where the randomness is induced by the model initialization prior to training. That is, $p(\boldsymbol{X}) = p(\boldsymbol{X}\boldsymbol{\pi})$.*

Note that the above theorem can also be generalized for a joint distribution over multiple graphs. For example, in graph retrieval, is necessary to compute the joint distribution of the embeddings of the query and corpus graph pairs $(G_q, G_c)$. In such cases, we have the following result (proven in Appendix E).

**Proposition 6.** *Given two graphs $G_q, G_c$, let the settings in Section 3.1 hold true. Specifically, let us assume that the loss function be invariant to simultaneous permutations of the embeddings $\boldsymbol{X}^{(q)} = \mathrm{GNN}_\theta(G_q)$ and $\boldsymbol{X}^{(c)} = \mathrm{GNN}_\theta(G_c)$. Then, $\boldsymbol{Y} = [\boldsymbol{X}^{(q)}; \boldsymbol{X}^{(c)}] \in \mathbb{R}^{2n \times D}$ satisfies $p(\boldsymbol{Y}) = p(\boldsymbol{Y}\boldsymbol{\pi})$.*

**Scope of the result**   We imposed a few simplifying assumptions only for brevity. In fact, our exchangeability results continue to hold even when these conditions are not explicitly met, including architectures that incorporate more complex operations such as normalization layers. Moreover, our results remain valid even when the loss itself is not permutation-invariant. This is because such losses may still exhibit invariance under a joint transformation consisting of (i) a permutation of intermediate representations; and, (ii) a corresponding permutation-induced transformation of the parameters in the subsequent layer (Appendix E.1.6).

## 4   APPLICATIONS TO GRAPH RETRIEVAL

**Graph retrieval**   In graph retrieval, we are given a large number of corpus graphs $C = \{G_c\}$ and the goal is to *efficiently* find out top-$b$ graphs that are relevant to a given query $G_q$. In a typical real-world application, the corpus database contains large number of graphs, necessitating efficient indexing and retrieval mechanisms, akin to other retrieval tasks. In this section, we exploit exchangeability to design a locality-sensitive hashing (LSH) method (Gionis et al.; Indyk et al., 1998; Charikar, 2002) that accommodates a wide variety of transportation-based graph distance measures in a unified framework. This would allow us to return the set of relevant items in a query time that is sublinear in the number of corpus items $|C|$.

We proceed in two steps: **(1)** We leverage our results on exchangeability (Theorem 5 and Proposition 6) to approximate the transportation-based graph similarity using Euclidean similarity, which is suited for LSH. **(2)** We build upon the proposal of (Roy et al., 2023) to design LSH for such approximate Euclidean similarity, which is also a valid LSH for the true transportation-based Euclidean similarity.

### 4.1   USE OF EXCHANGEABILITY TO DERIVE SIMILARITY OF GRAPHS IN EUCLIDEAN SPACE

**Transportation-based relevance distance between graphs**   It is well established in the literature (Roy et al., 2022; Zhuo et al., 2022; Fey et al., 2020; Jain et al., 2024; Bommakanti et al., 2024) that transport distance between sets of node embeddings across query and corpus graphs results in better accuracy than graph kernels or pooled single-vector representation. These works have proposed different notions of transportation distances, *e.g.*, hinge distance for subgraph matching (Roy et al., 2022), graph edit distance (Jain et al., 2024; Zhuo et al., 2022, GED), *etc*. We unify these distances under a common relevance distance, computed using a function $\rho$ convex, potentially asymmetric

and decomposable between dimensions, *i.e.*, $\rho(\boldsymbol{x}) = \sum_{d \in [D]} \rho(\boldsymbol{x}[d])$.

$$\Delta(G_c, G_q) = \min_{\boldsymbol{P} \in \mathcal{P}_n} \sum_{u,u'} \sum_{d \in [D]} \rho\big(\boldsymbol{x}^{(q)}(u)[d] - \boldsymbol{x}^{(c)}(u')[d]\big) \cdot \boldsymbol{P}[u, u'] \tag{1}$$

If $\rho(\bullet) = [\bullet]_+$, then $\Delta(G_c, G_q)$ captures the hinge distance for subgraph isomorphism (Roy et al., 2022); if $\rho(\bullet) = e_\ominus \times [\bullet]_+ + e_\oplus \times [-\bullet]_+$ for some $e_\ominus, e_\oplus > 0$, then $\Delta(G_c, G_q)$ captures GED, where $e_\ominus$ and $e_\oplus$ denote the costs of edge deletion and addition, respectively (Jain et al., 2024).

**Distance to similarity** Suppose the elements of the node embeddings are bounded by $x_{\max}$. Given cost function $\rho$, we compute $\rho_{\max} = \max_{x,x' \in [-x_{\max}, x_{\max}]} \rho(x - x')$. We define a score function $s(x) = \rho_{\max} - \rho(x)$, which converts the transportation-based distance in Eq. (1) to the following transportation-based similarity measure.

$$\text{sim}(G_c, G_q) = \max_{\boldsymbol{P} \in \mathcal{P}_n} \sum_{u,u'} \sum_{d \in [D]} s\big(\boldsymbol{x}^{(q)}(u)[d] - \boldsymbol{x}^{(c)}(u')[d]\big) \cdot \boldsymbol{P}[u, u']. \tag{2}$$

**Approximation of transportation-based similarity into Euclidean similarity** Owing to the random initialization of the parameters $\theta$, $\boldsymbol{x}^{(q)}(u)$ and $\boldsymbol{x}^{(c)}(u')$ are random variables, which makes $\text{sim}(G_c, G_q)$ a random scalar. Now, $\text{sim}(G_c, G_q)$ is not amenable to indexing and search. To tackle this, we approximate this similarity using a simpler Euclidean similarity $\text{sim}_d(G_c, G_q)$, focusing on a single dimension $d$. This approximate similarity is also a random variable, due to the parameter initialization, but more amenable to approximate nearest neighbor search. As we will see shortly, $\text{sim}_d(G_c, G_q)$ serves as a scaled approximation of $\text{sim}(G_c, G_q)$ with high probability.

Proposition 6 suggests that the node embedding pairs of $G_q$ and $G_c$ are exchangeable across dimensions *i.e.*, if $\boldsymbol{Y} = [\boldsymbol{X}^{(q)}; \boldsymbol{X}^{(c)}]$, then we have: $p(\boldsymbol{Y}) = p(\boldsymbol{Y}\boldsymbol{\pi})$ for any permutation $\boldsymbol{\pi}$. This means that the elements of the embeddings have an identical distribution across different dimensions. This also yields an identical distribution in the output of the score function $s(\cdot)$ across different embedding dimensions. This, in turn, allows us to approximate the score by evaluating it in any one dimension $d$:

$$\text{sim}_d(G_c, G_q) = \max_{\boldsymbol{P} \in \mathcal{P}_n} \sum_{u,u'} s\left(\boldsymbol{x}^{(q)}(u)[d] - \boldsymbol{x}^{(c)}(u')[d]\right) \cdot \boldsymbol{P}[u, u'] \tag{3}$$

By restricting Eq. (3) to a single dimension $d \in [D]$, the problem reduces to transportation cost between scalars. This — together with the property that $s(\cdot)$ is concave (as $\rho$ is convex) — allows us to simplify Eq. (3) (Appendix E) into a similarity function between the order statistics or the sorted vector of the node embedding elements in a fixed dimension. Specifically, we compute the order statistics: $\text{SORT}(\boldsymbol{X}^{(q)}[:, d])$ and $\text{SORT}(\boldsymbol{X}^{(c)}[:, d])$ and express the similarity function for dimension $d$ in Eq. (3) as the similarity between these order statistics:

$$\text{sim}_d(G_c, G_q) = s\big(\text{SORT}(\boldsymbol{X}^{(q)}[:, d]) - \text{SORT}(\boldsymbol{X}^{(c)}[:, d])\big) \tag{4}$$

As the distance function $\rho$ is decomposable $\rho(\boldsymbol{x}) = \sum_d \rho(\boldsymbol{x}[d])$, the score function satisfies: $s(\boldsymbol{x}) = \sum_d s(\boldsymbol{x}[d])$. Hence, we overload $s(\bullet)$ as a function on scalars in Eq. (3), as well as vectors in Eq. (4).

As exchangeability results in an identical distribution of the above similarity across the dimension $d$, we will have the following concentrations (Proven in Appendix E):

**Proposition 7.** For any $\epsilon > 0, \delta > 0$, setting $D > \frac{1}{\epsilon^2 \delta}$ ensures that, for some $\beta_0 = O_D(1)$, we have:

$$\Pr\left(\big|\text{sim}(G_c, G_q)/D - \text{sim}_d(G_c, G_q)\big| \le \epsilon\right) \ge 1 - \beta_0 \delta. \tag{5}$$

### 4.2 LOCALITY SENSITIVE HASHING OF GRAPHS

**Locality sensitive hashing** Locality Sensitive Hashing (LSH) maps queries and corpus items to the same bucket with high probability when they are similar, and with low probability otherwise (Gionis et al.; Indyk et al., 1998; Charikar, 2002; Neyshabur et al., 2015a). This enables retrieving relevant graphs from $\{G_c\}$ by searching only within the bucket where $G_q$ gets hashed.

**Why will existing approaches not work?** If $s(\cdot)$ in Eq. (4) were a symmetric Euclidean distance, we could directly apply existing LSH methods, such as grid-based projections for $L_1$ (Andoni et al., 2006) or line projections for $L_2$ (Datar et al., 2004). However, various common graph similarities are inherently asymmetric (refer to the examples below Eq. (1)). To address this limitation, we propose a new framework for LSH of graphs, starting with the definition of asymmetric-LSH for graphs under a general similarity measure (Neyshabur et al., 2015a).

**Definition 8.** *Given $Q, C$, the domain of query and corpus graphs and a similarity measure* sim : $C \times Q \to \mathbb{R}$. *A distribution over mappings* $\mathcal{F} : Q \to \mathbb{N}$ *and* $\mathcal{H} : C \to \mathbb{N}$ *is called a* $(S_0, \gamma S_0, p, p')$-*asymmetric LSH (ALSH) if, with* $p > p'$ *and* $\gamma \in (0, 1)$*, the following conditions are satisfied.*

$$(1) \ \Pr_{f \sim \mathcal{F}, h \sim \mathcal{H}}(f(G_q) = h(G_c)) \geq p, \ if \ \mathrm{sim}(G_c, G_q) \geq S_0, \qquad (6)$$

$$(2) \ \Pr_{f \sim \mathcal{F}, h \sim \mathcal{H}}(f(G_q) = h(G_c)) \leq p', \ if \ \mathrm{sim}(G_c, G_q) \leq \gamma S_0.$$

**Intuition behind our approach**  Suppose we estimate two vectors $\widehat{\boldsymbol{T}}_{q,d}$ and $\widehat{\boldsymbol{T}}_{c,d}$, such that the Euclidean similarity for dimension $d$ in Eq. (4) can be expressed as $\mathrm{sim}_d(G_q, G_c) \propto \cos(\widehat{\boldsymbol{T}}_{q,d}, \widehat{\boldsymbol{T}}_{c,d})$. Then, the random hyperplane projections $f(G_q) = \mathrm{sign}(\boldsymbol{w}^\top \widehat{\boldsymbol{T}}_{q,d})$ and $h(G_c) = \mathrm{sign}(\boldsymbol{w}^\top \widehat{\boldsymbol{T}}_{c,d})$ with $\boldsymbol{w} \sim \mathcal{N}(0, I)$, will be a valid LSH for $\mathrm{sim}_d$ (Charikar, 2002; Neyshabur et al., 2015a). Since this Euclidean similarity is only a scaled approximation of the transportation-based similarity $\mathrm{sim}(G_c, G_q)$ (Proposition 7), the same random hyperplane projection is a valid LSH for $\mathrm{sim}(G_c, G_q)$. Hence, we now focus on obtaining such vectors $\widehat{\boldsymbol{T}}_{q,d}$ and $\widehat{\boldsymbol{T}}_{c,d}$ whose inner product approximates $\mathrm{sim}_d$.

**GRAPHHASH: Our approach for LSH for graphs**  In their seminal work, Rahimi et al. (2007) showed that kernels of the form $\kappa(\boldsymbol{x} - \boldsymbol{x}')$ can be approximated using a product of finite-dimensional Fourier features. Our approximate similarity $\mathrm{sim}_d(G_c, G_q) = s(\mathrm{SORT}(\boldsymbol{X}^{(q)}[:, d]) - \mathrm{SORT}(\boldsymbol{X}^{(c)}[:, d]))$ has a similar structure. However, $s(\cdot)$ is generally not a kernel, because the underlying distance measure can involve complex asymmetric structure (see examples following Eq. (1)). Hence, their method cannot be directly applied. Roy et al. (2023) extended the approach to hinge-based similarities. We build on their idea and generalize it to arbitrary graph similarity functions. Specifically, we express $\mathrm{sim}_d(G_c, G_q)$ as an integral over dot products of two real vectors.

**Proposition 9.** *For each $u \in [n]$, there exist vectors $\boldsymbol{F}_{q,d}(\iota\omega_u), \boldsymbol{F}_{c,d}(\iota\omega_u) \in \mathbb{R}^4$ with different Fourier frequency $\omega_u$ for each node $u$, such that:* $\mathrm{sim}_d(G_c, G_q)$ *(Eq. (4)) can be expressed as:*

$$\mathrm{sim}_d(G_c, G_q) = \sum_{u \in [n]} \int_{\omega_u \in \mathbb{R}} \boldsymbol{F}_{q,d}(\iota\omega_u)^\top \boldsymbol{F}_{c,d}(\iota\omega_u) \, d\omega_u \qquad (7)$$

To approximate the above integral into finite terms, we design the frequency sampling distribution as $p(\omega_u) \propto |S(\iota\omega_u)|$, where $S(\iota\omega)$ is the Fourier transform of the scoring function $s(\bullet)$ when applied on scalars. Given $\boldsymbol{\omega} = [\omega_1, .., \omega_n]$, we use $\boldsymbol{T}_{\bullet,d}(\boldsymbol{\omega}) = [\boldsymbol{F}_{\bullet,d}(\iota\omega_u)/\sqrt{p(\omega_u)}]_{u \in [n]}$ to obtain an equivalent expression for Eq. (7), as follows:

$$\mathrm{sim}_d(G_c, G_q) = \mathbb{E}_{\omega_1, .., \omega_n \sim p(\bullet)}[\boldsymbol{T}_{q,d}(\boldsymbol{\omega})^\top \boldsymbol{T}_{c,d}(\boldsymbol{\omega})] \qquad (8)$$

We prove it in Appendix E. One can show that $||\boldsymbol{T}_{q,d}(\boldsymbol{\omega})||_2 = ||\boldsymbol{T}_{c,d}(\boldsymbol{\omega})||_2$ for all $G_q$ and $G_c$. Next, we draw $\{\boldsymbol{\omega}^{(m)}\} \overset{iid}{\sim} p(\boldsymbol{\omega})$ to compute $\widehat{\boldsymbol{T}}_{\bullet,d} (\in \mathbb{R}^{4nM}) \triangleq [\boldsymbol{T}_{\bullet,d}(\boldsymbol{\omega}^{(m)})]_{m \in [M]}$, which will give:

$$\mathrm{sim}_d(G_c, G_q) \propto \cos(\widehat{\boldsymbol{T}}_{q,d}, \widehat{\boldsymbol{T}}_{c,d}) \qquad (9)$$

**Overall routine (GRAPHHASH)**  Finally, we use the random hyperplane method to compute hash codes $f(G_q)$ and $h(G_c)$. Given $\dim_T$, the dimension of $\widehat{\boldsymbol{T}}_{\bullet,d}$ and $\dim_h$, the hashcode size, we first draw $\boldsymbol{W} \in \mathbb{R}^{\dim_h \times \dim_T}$ with $\boldsymbol{W}[r, t] \overset{iid}{\sim} \mathcal{N}(0, 1)$ and then set $h^{(d)}(G_c) = \mathrm{sign}(\boldsymbol{W}\widehat{\boldsymbol{T}}_{c,d})$ (Algorithm 1). During query execution, we return top-$b$ corpus graphs $\{G_c\}$ from the hash bucket $f^{(d)}(G_q) = \mathrm{sign}(\boldsymbol{W}\widehat{\boldsymbol{T}}_{q,d})$ (Algorithm 2). The family of these hash functions gives a valid LSH. We call our method as GRAPHHASH. We provide LSH guarantees for GRAPHHASH in Appendix E.

| **Algorithm 1** Indexing phase of GRAPHHASH | **Algorithm 2** Query phase of GRAPHHASH |
|---|---|
| **Require:** Corpus $\{G_c\}$, score function $s(\bullet)$ 
      frequency samples $\{\boldsymbol{\omega}^{(m)}\}$. 
 1: $\boldsymbol{W}[i, j] \sim \mathcal{N}(0, 1), i \in [\dim_h], j \in [\dim_T]$. 
 2: **for all** $G_c$ and $d \in [D]$ **do** 
 3:    Use $s(\cdot)$ to compute $\boldsymbol{F}_{c,d}(\iota\omega_u^{(m)})$ from 
      $\mathrm{SORT}(\boldsymbol{X}^{(c)}[:, d])$ for all $d, m$ 
 4:    Compute $\widehat{\boldsymbol{T}}_{c,d}$ from 
      $\{\boldsymbol{F}_{c,d}(\iota\omega_u^{(m)})\}$ and $\{p_\lambda(\omega_u^{(m)})\}$ 
 5:    $h^{(d)}(G_c) = \mathrm{sign}(\boldsymbol{W}\widehat{\boldsymbol{T}}_{c,d})$ 
 6:    Store $G_c$ in the bucket indexed by $h^{(d)}(G_c)$ 
 7: Store $\boldsymbol{W}$ for use in the query phase | **Require:** Query $G_q$, stored hyperplanes $\boldsymbol{W}$, 
      frequency samples $\{\boldsymbol{\omega}^{(m)}\}_{m=1}^M$ 
 1: $\mathcal{R} \leftarrow \emptyset$ 
 2: **for** $d \in [D]$ **do** 
 3:    Given $s(\cdot)$, compute $\boldsymbol{F}_{q,d}(\iota\omega_u^{(m)})$ from 
      $\mathrm{SORT}(\boldsymbol{X}^{(q)}[:, d])$ for all $d, m$ 
 4:    Compute $\widehat{\boldsymbol{T}}_{q,d}$ from 
      $\{\boldsymbol{F}_{c,d}(\iota\omega_u^{(m)})\}$ and $\{p_\lambda(\omega_u^{(m)})\}$ 
 5:    $f^{(d)}(G_q) = \mathrm{sign}(\boldsymbol{W}\widehat{\boldsymbol{T}}_{q,d})$ 
 6:    $\mathcal{R} \leftarrow \mathcal{R} \cup \{G_c : G_c \in \mathrm{Bucket}(f^{(d)}(G_q))\}$ 
 7: **Return** Top-$b$ graphs from $\mathcal{R}$ |

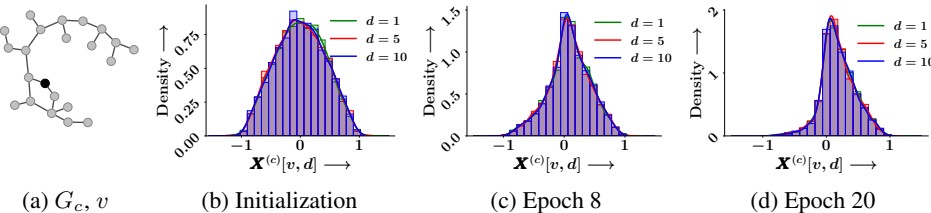

| (a) $G_c$, $v$ | (b) Initialization | (c) Epoch 8 | (d) Epoch 20 |

Figure 1: Empirical probability density of $\boldsymbol{X}^{(c)}[v, d]$ the highlighted node $v$ in the example corpus graph $G_c$ in cox2, obtained using 5000 independently trained instances of the GNN model for Subgraph Matching based graph retrieval. Panels (b)–(d) show the density of $\boldsymbol{X}^{(c)}[v, d]$ after model initialization and different stages of training.

## 5 EXPERIMENTS

We organize our experiments in two parts: first, we empirically validate the exchangeability property of GNN-based graph embeddings (Theorem 5); second, we evaluate the retrieval effectiveness of GRAPHHASH across multiple datasets. Appendix H shows additional experiments.

### 5.1 EMPIRICAL VALIDATION OF EMBEDDING EXCHANGEABILITY

**Validation using marginal distribution** We verify a necessary condition of exchangeability: identical marginal distribution of the embedding elements for a fixed node across independently initialized and trained models. We train 5,000 independently initialized GNN models on a small subset of the cox2 dataset, consisting of 1,024 query-corpus graph pairs. Each model is trained for 20 epochs using the Adam optimizer with an embedding size $D = 10$, by minimizing a ranking loss for a subgraph matching based graph retrieval task. For each trained model, we extract the embedding vector for a fixed, node $v$ from one graph $G_c$ and record the scalar values $\boldsymbol{X}^{(c)}[v, d]$ for $d \in [D]$. This yields an empirical distribution of $\boldsymbol{X}^{(c)}[v, d]$ across model instances for each $d \in [D]$.

Figure 1 shows the empirical probability density of $\boldsymbol{X}^{(c)}[v, d]$ for three representative dimensions $d = 1, 5, 10$, at three points in training: initialization, epoch 8, and epoch 20. We observe that the distributions remain identical across the embedding dimensions throughout training. This validates the necessary condition of our result that the embedding dimensions are exchangeable under random initialization and remain so despite backpropagation, non-convex losses.

**Direct test for exchangeability** The marginal distributions do not capture more complex dependencies between dimensions, which is why we make use of the maximum mean discrepancy to quantify the gap between the distribution of

| | |
|---|---|
| cox2 (GED) | $-3.89 \times 10^{-5}$ $\pm$ $2.69 \times 10^{-5}$ |
| cox2 (SM) | $-1.18 \times 10^{-6}$ $\pm$ $3.28 \times 10^{-5}$ |

Table 2: Estimator for unbiased $\text{MMD}^2$ for $p_{\boldsymbol{X}}$ and $p_{\boldsymbol{X}\boldsymbol{\pi}}$ for cox2 dataset

$\boldsymbol{X}$ and $\boldsymbol{X}\boldsymbol{\pi}$. We sample 100 different permutations and compute the estimator of $\text{MMD}^2$ for each permutation, and report the average over these 100 observations. Note that estimator of $\text{MMD}^2$ can be negative. Table 2 shows that the MMD values are extremely small for cox2dataset for both GED and subgraph matching (SM). These results strongly support that $p_{\boldsymbol{X}}$ and $p_{\boldsymbol{X}\boldsymbol{\pi}}$ are close.

**Rank of** $\mathbb{E}[\boldsymbol{X}]$ Another consequence of exchangeability is that the expectation of the graph embedding matrix $\mathbb{E}[\boldsymbol{X}]$ is rank one. Consequently, we expect the leading singular value of the sample mean graph embedding matrix to be significantly larger than the rest. Figure 3 shows how the ratio $\frac{\sigma_1^2}{\sum_i \sigma_i^2}$ varies over multiple runs, where $\sigma_1, ..\sigma_n$ are the singular values of $\mathbb{E}[\boldsymbol{X}]$, sorted in decreasing order. We observe that this frac-

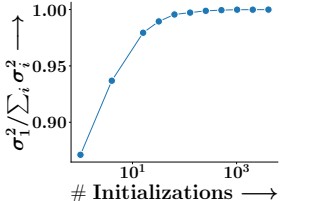
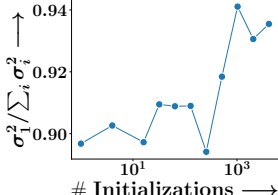

| (a) Graph from cox2 (SM) | (b) Graph from cox2 (GED) |

Figure 3: The relative size of the top singular value of the mean (trained) embedding across model initializations.

tion converges to one, which indicates that the rank of the embedding matrix is 1.

## 5.2 Evaluation of GraphHash's Retrieval Performance

We evaluate GraphHash against existing baselines on four datasets to assess retrieval accuracy-efficiency trade-offs across indexing strategies.

**Setup** We construct retrieval datasets using four real-world benchmarks from the TUDatasets (Morris et al., 2020): `ptc-fr`, `ptc-fm`, `cox2`, and `ptc-mr`. Each dataset consists of 500 query graphs and a corpus of 100,000 graphs, following related work (Roy et al., 2022; Lou et al., 2020). We generate binary relevance labels under two asymmetric graph similarity signals: **(1) Subgraph Matching (SM):** Relevance is determined using the VF2 subgraph matching algorithm (Hagberg et al., 2020). Here, we set binary relevance $\text{rel}(G_c, G_q) = [\![G_q \subset G_c]\!]$, where $[\![\bullet]\!]$ is the indicator function. **(2) GED:** We use the GEDLIB toolkit (Blumenthal et al., 2019) to compute edit distances with asymmetric costs $e_\oplus = 1$ (insertion) and $e_\ominus = 2$ (deletion), followed by thresholding to obtain binary relevance. Here, we set $\text{rel}(G_c, G_q) = [\![\text{GED}(G_c, G_q) \leq \tau]\!]$, where $\tau$ is a threshold. For each supervision type, we train a separate transport-based scoring model using the relevance distances for Subgraph Matching and for GED. The model is trained using a pairwise ranking loss (Roy et al., 2022; Jain et al., 2024) of the form $\sum_q \sum_{c:\text{rel}(G_c, G_q)=1, c':\text{rel}(G_{c'}, G_q)=0} [\Delta(G_c, G_q) - \Delta(G_{c'}, G_q) + \gamma]_+$ where $\gamma$ is a fixed margin, and $\Delta(\cdot, \cdot)$ denotes the transport-based relevance distance (Eq. (1)). We evaluate retrieval performance using both MAP and NDCG. The analysis presented below focuses on MAP, while NDCG results and additional experiments are in Appendix H.

We benchmark GraphHash against five competitive ANN methods adapted to graph retrieval. These include single-vector and multi-vector indexing paradigms. **(I) FourierHashNet (Roy et al., 2023):** It implements an LSH tailored for shift-invariant asymmetric distances by projecting graph emnbeddings into the Fourier space. Each graph $G_\bullet$ is represented as a single vector $z_\bullet = \frac{1}{|V_\bullet|} \sum_{u \in V_\bullet} x(u)$, where $X = [x(u)]_{u \in [n]}$. **(II) Random Hyperplanes (RH) (Charikar, 2002; Indyk et al., 1997):** It serves as a classic LSH baseline, where we directly hash mean pooled graph representations using random linear projections. **(III) IVF (Douze et al., 2024):** It follows the FAISS-based ColBERT-style approach, constructing a dense inverted index over the collection of corpus node embeddings, and probes with individual query node vectors, followed by aggregating the hits at the graph level. **(IV) DiskANN (Simhadri et al., 2023)** follows a similar multi-vector setup but leverages an HNSW index over corpus node embeddings. Lastly, we include a **Random** baseline that retrieves a uniformly

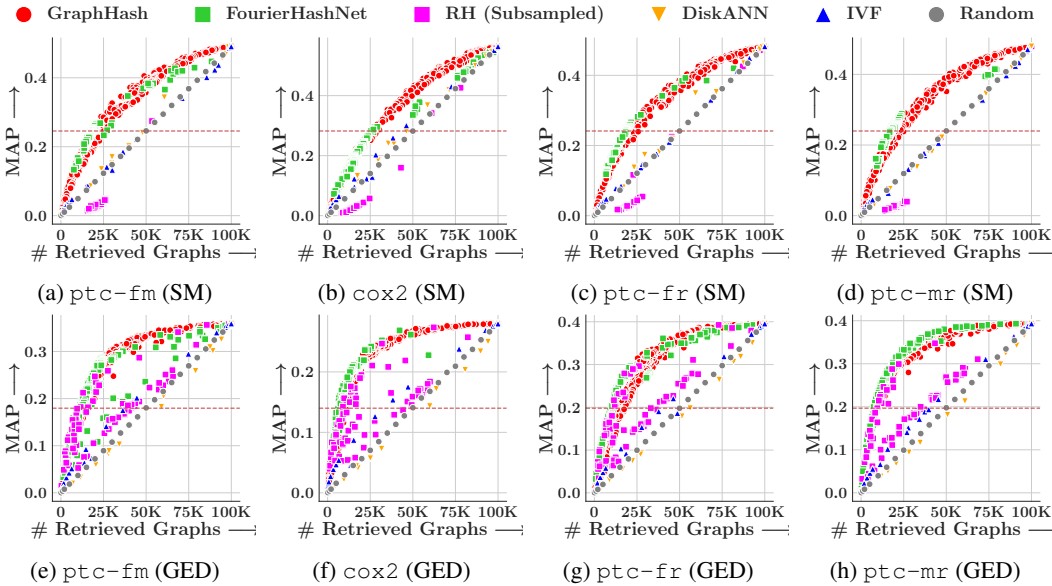

(a) `ptc-fm` (SM)   (b) `cox2` (SM)   (c) `ptc-fr` (SM)   (d) `ptc-mr` (SM)

(e) `ptc-fm` (GED)   (f) `cox2` (GED)   (g) `ptc-fr` (GED)   (h) `ptc-mr` (GED)

Figure 4: Trade-off between mean average precision (MAP) and number of retrieved graphs, for GraphHash, FourierHashNet (Roy et al., 2023), Random Hyperplane (RH) (Charikar, 2002; Indyk et al., 1997), IVF (Douze et al., 2024),DiskANN (Simhadri et al., 2023) and Random, across all datasets. Top row: Retrieval based on Subgraph Matching (SM); Bottom row: Retrieval based on GED. Horizontal red line denotes 50% of exhaustive MAP. Our method shows a better trade-off than others in majority of the cases.

**Results** We vary hyperparameters in each method to produce different retrieval set sizes, yielding MAP vs. # retrieved graphs trade-offs shown in Figure 4. The key observations are as follows. **(1)** GRAPHHASH consistently outperforms all baselines across both Subgraph Matching (SM) and Graph Edit Distance (GED), with FourierHashNet emerging as the next-best method overall. **(2)** FourierHashNet fails to span the full selectivity spectrum, particularly on SM tasks—most notably on `ptc-fr` and `ptc-mr`, where its MAP plateaus below 50% of the exhaustive MAP. **(3)** RH hashing performs reasonably well on GED, occasionally matching GRAPHHASH in MAP. However, it exhibits high variance at fixed selectivity levels, complicating hyperparameter tuning. On SM tasks, RH performs worse than random, which is expected since cosine similarity over pooled vectors is ill-suited to the asymmetric nature of containment queries. **(4)** DiskANN and IVF, despite using multi-vector indexing, perform poorly due to their reliance on symmetric similarity metrics like $L_2$ and cosine, which are incompatible with the asymmetric transport-based supervision. **(5)** Random sampling yields substantially lower MAP compared to both GRAPHHASH and FourierHashNet, highlighting the non-trivial structure captured by learned or LSH-based methods.

Next, we vary $\dim_h$ (number of hash bits) and obtain different trade-off curve between MAP and #no of retrieved graphs. We plot the variation of AUC against $\dim_h$, which shows at around $\dim_h = 10$, we obtain an optimal trade-off.

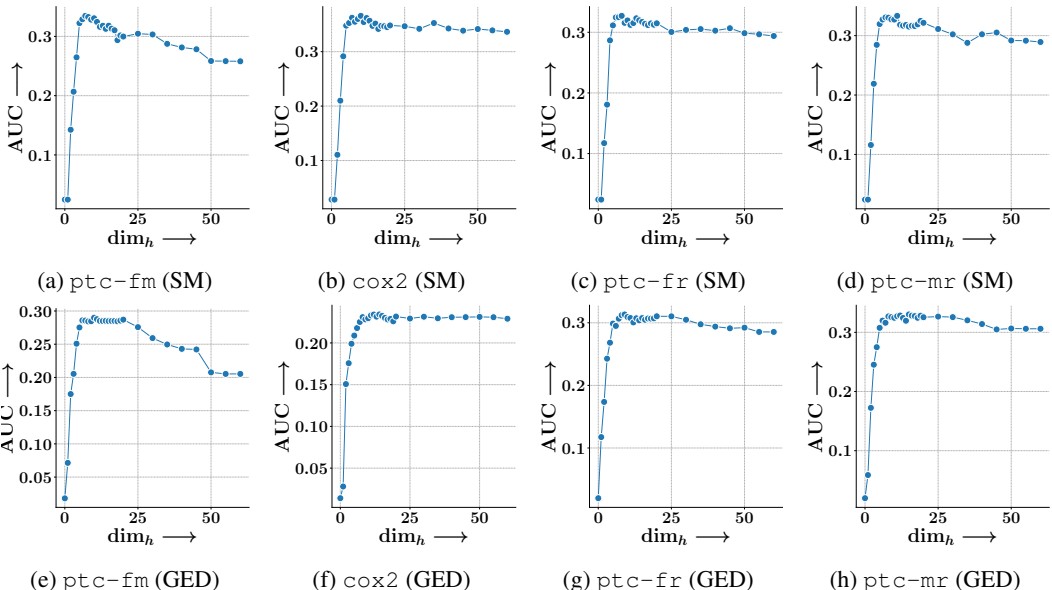

Figure 5: Performance of GRAPHHASH across different choices for $\dim_h$, the size of the hashcode. We summarize the trade-off plot between MAP and the number of retrieved graphs by computing the area under the curve after normalizing the x-axis. We observe that the optimal size is around $\dim_h = 10$ across datasets and tasks.

## 6 CONCLUSIONS

Taking a step beyond existing notions of algebraic symmetries in neural architectures and losses, we introduce the property of exchangeability over neural graph embeddings. We show that this property is exhibited by a broad class of graph neural networks across a broad class of loss functions and optimizers. We utilize this property to obtain a concentration bound for reducing transport problems on node embeddings, culminating in GRAPHHASH, a unified and theoretically grounded framework for approximate graph retrieval using general transport-based distances. We experimentally validate exchangeability, and GRAPHHASH consistently outperforms strong baselines in retrieval performance under both subgraph matching and edit distance supervision. Future work might explore other consequences of the phenomenon on learning and training dynamics. It may be worthwhile to extend the framework to similarities over a richer class of similarity functions between three dimensional molecular structures, 3D objects, *etc*.

ETHICS STATEMENT

This work makes an algorithmic contribution and uses only publicly available, non-proprietary graph datasets under their original licenses. No human subjects or sensitive data are involved. We believe our results advance understanding of graph retrieval without raising additional ethical concerns.

REPRODUCIBILITY STATEMENT

We provide code link in abstract, configuration files, and dataset splits to fully reproduce all experiments. Hyperparameters, training settings, and evaluation protocols are documented, and scripts are included to regenerate the reported figures and tables. In addition, all theorems are stated formally with accompanying proofs in the appendix to allow independent verification of our theoretical claims.

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

# Exchangeability of GNN Representations with Applications to Graph Retrieval (Appendix)

CONTENTS

## A  BROADER IMPACT

Our work is the first of its kind within the space of distributional symmetries in neural architectures, as it moves the focus towards the distribution of embeddings over randomness in initialization. Our work may also be adapted to other classes of neural networks. Probabilistic symmetries may have other consequences to training and learning dynamics, like our concentration bound.

GRAPHHASH also offers an efficient way to retrieve graphs from a large database of graphs. It can help in identifying a subset of molecules which is similar to some other molecule, from a large corpus. It can also help in video or image retrieval by specifically focusing on scene graphs. Thus, our work has the potential to reduce computational cost and carbon footprint of large search systems.

## B  LIMITATIONS

**(1)** We only restrict ourselves to exchangeability as probabilistic symmetry of GNN, which is symmetry induced by permutations in the weight space. In this work, we do not consider how other types of symmetry can affect the probability density function of the embeddings. However, our work can be seen as a stepping stone to characterize such cases. **(2)** It is well known that the exchangeable sequence $(Y_1, ..., Y_D)$ tends to become an i.i.d. sequence as $D \to \infty$. However, this does not apply to our setting because the values of the embedding elements also depend on $D$. It would be interesting to discover asymptotic characterization of embedding values. **(3)** Exact graph distance involves solving a quadratic assignment problem, whereas its surrogate used in Eq. (1) approximates graphs using sets. This gives a first order approximation, which allows us to leverage exchangeability to approximate transportation distance between two embedding sets using Euclidean distance. One can provide more accurate approximation using distance between edge embeddings. We did not provide this formulation in our paper. However, our work can be easily extended to such setting, by considering joint distribution between node pairs.

## C  LLM USAGE

We used an LLM primarily for correction of grammar and polishing text. Very occasionally, we used it to supplement bibliographic search. No LLM was used to generate ideas, design experiments, analyze data, implement algorithms, or produce results. We carefully reviewed and revised any response provided by LLM.

## D  RELATED WORK

**Representation learning**   Representation using dense embeddings of structured objects has been a much-studied area of research, e.g. for, sets (Lee et al., 2019; Zaheer et al., 2017), sequences (Palangi et al., 2016; Zhou et al., 2024), and graphs (Cai et al., 2018; Wang et al., 2017). Relatively fewer results focus on the question of retrieval using these embeddings (Li et al., 2024; Duong, 2022; Gerritse et al., 2020). Prior works on graph retrieval predominantly aggregate node embeddings from each graph into a single, pre-computable embedding vector (Li et al., 2019; Bai et al., 2019; Ranjan et al., 2022). This allows for the use of standard indexing methods for vector similarity search. However, this reduces accuracy due to compressing the entire graph into one embedding. Yi et al. (2025b) discuss exchangeability in the context of hyperdimensional vectors.

**Transportation distance in graphs**   More recent techniques for graph embedding employ node-based vectors and then define relevance scores of the corpus graphs with respect to the query by using transportation distance between the two sets of vectors (Roy et al., 2022; Zhuo et al., 2022; Fey et al., 2020). The cost within the transportation framework models various notions of relevance measure, including asymmetric measures for subgraph matching, graph edit distance with non-uniform costs, *etc.*, which results in enhanced accuracy, as compared to aggregation to single vectors.

**Locality sensitive hashing**   After obtaining the embedding (or set of embeddings), there still remains the question of finding out the most relevant object using this representation. For traditional vector databases, locality sensitive hashing (LSH), Indyk et al. (1998) pioneered a celebrated method for approximate near neighbor search. The benefit of LSH over comparable techniques, e.g., IVF, and graph-based techniques, e.g., HNSW, is the faster indexing time while giving comparable or slightly worse recall times.

**LSH for transportation distance**    A key contribution of the current work is to propose an LSH for transportation distance, in context of GNN. Nearest neighbor methods has been studied extensively in the theory community  (Andoni et al., 2009; Chen et al., 2022; 2020; Indyk, 2004; Andoni et al., 2008; Jayaram et al., 2024). They first embed a set similarity into Euclidean space with some distortion factor, and then use this reduction to design an LSH. However, the similarity measure in these existing works is always symmetric, whereas in graph retrieval, it is often asymmetric, such as in subgraph matching or Graph Edit Distance (GED) with non-uniform costs.

**Sliced Wasserstein distance**    While transportation distance is computationally expensive, recent studies have explored approximations that are cheaper (Kolouri et al., 2019; Deshpande et al., 2018; Vayer et al., 2019). The most well-known one, perhaps, is the *sliced Wasserstein* (SW) distance, which is the average of the Wasserstein distance over multiple 1D random projections. Deshpande et al. (2018) show the efficacy of the SW distance for GAN training. Kolouri et al. (2019) demonstrate the connection of SW distance to the Radon transform, and Vayer et al. (2019) propose *sliced Gromov Wasserstein*, a similar approximation for the Gromov-Wasserstein distance, also used for optimal transport. However, none of them study the question of efficient retrieval under such distances, or the connection with dimension exchangeability of representations produced by common neural networks.

Transportation distance has also been studied in the average case: Jayaram et al. (2024) give a $O(\log n)$ approximate data-dependent LSH in the distributional case. In our setting, this problem is tackled by showing the exchangeability of embedding dimensions of GNNS. Our result is incomparable to (Jayaram et al., 2024), since their posited distribution is not exchangeable, and our set of exchangeable distributions is broader than what (Jayaram et al., 2024) assumed. The notion of exchangeability has been studied before for neural networks, but in different contexts and toward different goals. Set transformers famously utilized permutation invariance to give set embeddings, exchangeable networks for set-to-set matching were described by Saito et al. (2020), while Bloem-Reddy et al. (2020) characterized invariant network architectures for a particular symmetry property, including exchangeability, of the input. However, none of these results have characterized the exchangeability property of the embedding dimensions, as is done in our work. In Introduction, we have already mentioned works that recognized various symmetries of loss surfaces with respect to hidden units of some standard networks. In those works, such symmetry is usually an impediment to fast optimization, remedied by advanced optimization techniques. In contrast, we use such symmetries to establish exchangeability, in the service of efficient LSH indexes.

# E   PROOFS AND OTHER TECHNICAL DETAILS

In this section, we present the proofs of the technical results presented in Section 3 and Section 4.

## E.1   PROOFS OF THE RESULTS OF EXCHANGEABILITY PRESENTED IN SECTION 3

Here, we prove Lemma 2, Lemma 3, Lemma 4, Theorem 5 and Proposition 6. To achieve this goal, we first restate the setting:

**(1) Broad class of GNN architectures**   We consider the a wide variety of GNN architectures, which are enlisted in Appendix F. This list encompasses a wide range of GNN architectures, including gated GNN (Gilmer et al., 2017), GIN (Xu et al., 2019), GAT (Veličković et al., 2018), GCN (Kipf et al., 2017). Note that, our analysis is likely to extend beyond these cases, and can also be applied in Graph transformers, as shown in Appendix F

**(2) IID intialization of the parameters within a layer**   The entries of the parameter matrix $\Theta^{(\ell)}$ in each layer of are initialized in an i.i.d manner. Parameters across different layers are initialized independently, but not necessarily identically. This covers standard model initialization schemes, such as Kaiming initialization (He et al., 2015) and Xavier initialization (Glorot et al., 2010), both of which yield i.i.d. initialization of the parameters within a layer.

**(3) Permutation invariance of loss function**   We consider the loss function is invariant to the permutations of elements in the node embeddings. This holds naturally in several settings including our graph retrieval. Here, the loss, whether binary cross-entropy or pairwise ranking, depends on the similarity between $(G_q, G_c)$ via the transportation plan between $X^{(q)}$ and $X^{(c)}$ (Roy et al., 2022; Zhuo et al., 2022). Since this similarity is invariant under permutations of embedding elements, the loss is likewise permutation-invariant. In link prediction, the similarity between two nodes $u$ and $v$ is often computed as the dot product $x(u)^\top x(v)$, which is invariant to permutations of the elements of $x$. Consequently, the associated loss is also permutation-invariant.

**(4) Broad class of optimizers**   The optimizer for training can be SGD (Zhang, 2004), Adam (Kingma et al., 2015), *etc*. This pertains to standard optimizers, which are routinely employed across learning settings.

**Additional Notation**   We further introduce supplementary notation.

**(1)** We use $\Theta_t^{(\ell)}$ to denote the parameter matrix of the $\ell$-th layer at the $t^{\text{th}}$ update step. We shall index our weights using the set $[\ell_{\max}] = \{0, 1, \ldots, \ell_{\max}\}$, which shall implicitly cover each of the components (embedding initialization, message passing and update step). We will typically use $\ell$ to denote the layer index.

**(2)** $\Theta_{<t}^{(\ell)}$ denotes the *collection* of parameters $\Theta_{\text{iter}}^{(\ell)}$ for iter $= 0, 1, \ldots, t - 1$.

**(3)** $\theta_{<t}$ denotes the collection of all parameters $\theta_{\text{iter}}$ for iter $= 0, 1, \ldots, t - 1$.

**(4)** $\Gamma_{\boldsymbol{\pi}}^{(\ell)}$ is a transformation on the parameters of the $\ell$-th layer. $\Gamma_{\boldsymbol{\pi}}$ is a global transformation on all parameters. We take $\Gamma_{\boldsymbol{\pi}}$ to be separable across layers (this holds for the permutation-based transformations considered by us). That is, $\Gamma_{\boldsymbol{\pi}}$ may be written as $\Gamma_{\boldsymbol{\pi}} = \bigoplus_{\ell \in [\ell_{\max}]} \Gamma_{\boldsymbol{\pi}}^{(\ell)}$. This means that $\Gamma_{\boldsymbol{\pi}}(\theta) = \left( \Gamma_{\boldsymbol{\pi}}^{(\ell)}(\Theta^{(\ell)}) \mid \ell \in [\ell_{\max}] \right)$.

**(5)** $\mathcal{I}_2$ refers to the domain of the parameters, which is $\mathbb{R}^p$ where $p$ is the number of parameters in the network.

**(6)** We refer to the loss function at the $t^{\text{th}}$ update step as $\text{loss}_t$, which a function of the parameters of the network, i.e., $\text{loss}_t(\theta)$; thus the index $t$ encodes the batching/data used for that update step. When it is clear from context, we may write $\text{loss}_t(\theta_t)$ simply as $\text{loss}_t$.

**(7)** $\boldsymbol{\delta}_{\Delta,(k,l)}$ is defined as the matrix of appropriate dimensions with all zeros except for a $\Delta$ at the $(k, l)$-th position. Note that this is different from Dirac delta function $\delta(\bullet)$ — we will alert the reader if we use $\delta$ as Dirac delta function.

**(8)** We denote the gradient of the loss function with respect to the parameters $\theta_t$ as the collection $\text{grad}_t \triangleq (\mathbf{grad}_t^{(\ell)} | \ell \in [\ell_{\max}])$, where $\ell$ is the layer index. Here, $\mathbf{grad}_t^{(\ell)}$ is a matrix of the same dimensions as $\Theta_t^{(\ell)}$ which has the corresponding gradients. As set by earlier convention, $\mathbf{grad}_{<t}^{(\ell)}$ denotes the collection of gradients $\mathbf{grad}_{\text{iter}}^{(\ell)}$ for iter $= 0, 1, \ldots, t - 1$, and $\text{grad}_{<t}$ denotes the collection of all gradients $\text{grad}_{\text{iter}}$ for iter $= 0, 1, \ldots, t - 1$.

### E.1.1 Proof of Lemma 2

**Lemma 2.** *Given a graph $G$ and a GNN architecture $\mathrm{GNN}_\theta$ enlisted in Appendix F, let the node embedding matrix of $G$ be $\boldsymbol{X} = \mathrm{GNN}_\theta(G) \in \mathbb{R}^{n \times D}$. Then, for any permutation matrix $\boldsymbol{\pi} \in \mathcal{P}_D$, there exists a bijective transformation $\Gamma_{\boldsymbol{\pi}}$ with $|Det\left(\partial \Gamma_{\boldsymbol{\pi}}(\theta)/\partial \theta\right)| = 1$ such that $\boldsymbol{X}\boldsymbol{\pi} = \mathrm{GNN}_{\Gamma_{\boldsymbol{\pi}}(\theta)}(G)$. We call $\Gamma_{\boldsymbol{\pi}}$ as a permutation induced transformation, for $\boldsymbol{\pi}$.*

**Proof:** *Overview.* In this section, we focus on two architectures, which covers the intricacy involved in designing the permutation inducing transformation. For other GNN architectures, we provide the reader with building blocks for transformations involving other common GNN layers in Appendix F.

In this proof, we consider the GNN in the form of gated GNN used by Li et al. (2016); Gilmer et al. (2017).

*Architecture.* Given integers $K$ and $D$, a graph neural network ($\mathrm{GNN}_\theta$) computes node embeddings $\boldsymbol{x}_k(u) \in \mathbb{R}^D$ for $u \in V$ using $K$ message passing steps. Here, we initialize $\boldsymbol{x}_0(u)$ using node features $\mathbf{feat}(u)$ and keep updating $\boldsymbol{x}_k$ using two neural networks $\mathrm{upd}_\theta$ and $\mathrm{msg}_\theta$ having parameters $\theta$.

$$\boldsymbol{x}_0(u) = \mathrm{init}_\theta(\mathbf{feat}(u)), \tag{10}$$

$$\boldsymbol{x}_{k+1}(u) = \mathrm{upd}_\theta\left(\boldsymbol{x}_k(u), \sum_{v:(u,v)\in E} \mathrm{msg}_\theta(\boldsymbol{x}_k(u), \boldsymbol{x}_k(v))\right), \qquad \text{for } k < K. \tag{11}$$

In the above: $\mathrm{init}_\theta, \mathrm{msg}_\theta$ are multilayer perceptron (MLP) networks of the form of $\mathrm{Linear}^{(\ell_{\max})} \circ \sigma^{(\ell_{\max}-1)} \circ \cdots \circ \sigma^{(1)} \circ \mathrm{Linear}^{(1)}$, where $\mathrm{Linear}^{(\ell)}$ is a linear layer and $\sigma^{(\ell)}$ is an activation function that applies pointwise. $\mathrm{upd}_\theta$ can be (a) an MLP network or, (b) one layer of GRU (Gilmer et al., 2017). In the current analysis, we omit step index $t$, since we are focusing on only one step.

**Gated GNN with MLP based $\mathrm{upd}_\theta$:** *Proof Sketch.* In particular, we assume that each of

$\mathrm{init}_\theta, \mathrm{msg}_\theta, \mathrm{upd}_\theta$ is a simple MLP with 1, 2, and 2 layers, respectively. The figure shows initialization and recursive propagation from layer $k$ to $k+1$. To induce the transformation $\boldsymbol{x}_K(u) \mapsto \boldsymbol{x}_K(u)\boldsymbol{\pi}$, we modify the final layer of $\mathrm{upd}_\theta$ as $\Theta^{(4)} \mapsto \Theta^{(4)}\boldsymbol{\pi}$, which also changes all intermediate outputs of $\mathrm{upd}_\theta$: $\boldsymbol{x}_k(u) \mapsto \boldsymbol{x}_k(u)\boldsymbol{\pi}$. This change affects $\mathrm{msg}_\theta$ inputs. We undo the "side-effect" by transforming $\Theta^{(1)}$ to $\mathrm{Diag}(\boldsymbol{\pi}^\top, \boldsymbol{\pi}^\top)\Theta^{(1)}$. Finally, we update $\Theta^{(0)} \mapsto \Theta^{(0)}\boldsymbol{\pi}$ to ensure that the initial input to $\mathrm{msg}_\theta$, namely $\boldsymbol{x}_0(u)\boldsymbol{\pi}$, aligns with the transformed flow. Since the rest of the network remains unchanged, this transformation is agnostic to the depths of init, msg, and upd, affecting only the last layers of init and upd and the first layer of msg.

*Detailed Proof.* Firstly, we re-index the network weights for readability, as — **(I)** init: Let the last weight of init be $\Theta^{(\ell_0)}$. **(II)** msg: Given $(u, v) \in E$, and the propagation layer $k$, let $\bar{\boldsymbol{X}}_k^{(0)} = [\boldsymbol{x}_k^\top(u), \ \boldsymbol{x}_k^\top(v)]$ be the input to the message propagation layer after the node embeddings are concatenated according to the edges in the graph. The weight matrix in the first propagation layer of msg is $\Theta^{(\ell_1)}$. Let $\bar{\boldsymbol{X}}_k^{(\ell_1)}$ be the output of $\Theta^{(\ell_1)}$, *i.e.*, $\bar{\boldsymbol{X}}_k^{(\ell_1)} = \bar{\boldsymbol{X}}_k^{(0)}\Theta^{(\ell_1)}$ **(III)** upd: Let the final layer of upd be $\Theta^{(\ell_2)}$. The transformation is defined as follows:

$$\Gamma_{\boldsymbol{\pi}}^{(\ell_0)}(\boldsymbol{\Theta}^{(\ell_0)}) = \boldsymbol{\Theta}^{(\ell_0)}\boldsymbol{\pi}, \tag{12}$$

$$\Gamma_{\boldsymbol{\pi}}^{(\ell_1)}(\boldsymbol{\Theta}^{(\ell_1)}) = \begin{bmatrix} \boldsymbol{\pi}^\top & \boldsymbol{0} \\ \boldsymbol{0} & \boldsymbol{\pi}^\top \end{bmatrix} \boldsymbol{\Theta}^{(\ell_1)}, \tag{13}$$

$$\Gamma_{\boldsymbol{\pi}}^{(\ell_2)}(\boldsymbol{\Theta}^{(\ell_2)}) = \boldsymbol{\Theta}^{(\ell_2)}\boldsymbol{\pi} \tag{14}$$

While the remaining transformations are identity, *i.e.*, $\Gamma_{\boldsymbol{\pi}}^{(\ell)} = \boldsymbol{I}_{\dim(\boldsymbol{\Theta}^{(\ell)})}$ for all $\ell \notin \{\ell_0, \ell_1, \ell_2\}$. We shall show that the output of the network is permuted in columns by $\boldsymbol{\pi}$, by tracing the effect of the transformation from the input to the output. We show this inductively on the number of propagation steps.

*Base case.* For $k = 0$. As $\boldsymbol{\Theta}^{(\ell_0)} \mapsto \boldsymbol{\Theta}^{(\ell_0)}\boldsymbol{\pi}$, we have: $\boldsymbol{X}_0 \mapsto \boldsymbol{X}_0\boldsymbol{\pi}$.

*Inductive Step.* Suppose that $\boldsymbol{X}_k \mapsto \boldsymbol{X}_k\boldsymbol{\pi}$ for some $k$. Then $\bar{\boldsymbol{X}}_k^{(0)} = [\boldsymbol{x}_k^{\top}(u), \; \boldsymbol{x}_k^{\top}(v)] \mapsto \bar{\boldsymbol{X}}_k^{(0)} \begin{bmatrix} \boldsymbol{\pi} & \boldsymbol{0} \\ \boldsymbol{0} & \boldsymbol{\pi} \end{bmatrix}$ under $\theta \mapsto \Gamma_{\boldsymbol{\pi}}(\theta)$.

Since we transform $\boldsymbol{\Theta}^{(\ell_1)} \mapsto \begin{bmatrix} \boldsymbol{\pi}^{\top} & \boldsymbol{0} \\ \boldsymbol{0} & \boldsymbol{\pi}^{\top} \end{bmatrix} \boldsymbol{\Theta}^{(\ell_1)}$ and $\bar{\boldsymbol{X}}_k^{(0)} \mapsto \bar{\boldsymbol{X}}_k^{(0)} \begin{bmatrix} \boldsymbol{\pi} & \boldsymbol{0} \\ \boldsymbol{0} & \boldsymbol{\pi} \end{bmatrix}$, the quantity $\bar{\boldsymbol{X}}_k^{(\ell_1)} \mapsto \bar{\boldsymbol{X}}_k^{(0)} \begin{bmatrix} \boldsymbol{\pi} & \boldsymbol{0} \\ \boldsymbol{0} & \boldsymbol{\pi} \end{bmatrix} \begin{bmatrix} \boldsymbol{\pi}^{\top} & \boldsymbol{0} \\ \boldsymbol{0} & \boldsymbol{\pi}^{\top} \end{bmatrix} \boldsymbol{\Theta}^{(\ell_1)} = \bar{\boldsymbol{X}}_k^{(0)} \boldsymbol{\Theta}^{(\ell_1)}$ remains unchanged as $\boldsymbol{\pi}\boldsymbol{\pi}^{\top} = \boldsymbol{I}$.

Due to this, $\bar{\boldsymbol{X}}_k^{(\ell_1)}$ remains invariant to $\Gamma_{\boldsymbol{\pi}}$. Until the final layer of updates, all transformations $\Gamma_{\boldsymbol{\pi}}^{(\ell)}$ are identity and therefore, the resultant intermediate embeddings also remain invariant. At the final layer, we have $\boldsymbol{\Theta}^{(\ell_2)} \mapsto \boldsymbol{\Theta}^{(\ell_2)}\boldsymbol{\pi}$ (from Eq. (14)). This will give: $\boldsymbol{X}_{k+1} \mapsto \boldsymbol{X}_{k+1}\boldsymbol{\pi}$.

**Gated GNN with** GRU **based** $\mathrm{upd}_\theta$**:** **(I)** Let $\boldsymbol{\Theta}^{(\ell_0)}, \boldsymbol{\Theta}^{(\ell_1)}, \bar{\boldsymbol{X}}_k^{(0)}, \bar{\boldsymbol{X}}_k^{(\ell_1)}$ bear the same meaning as before. **(II)** $\mathrm{upd}_\theta$: We introduce the hidden state encoding of the GRU: $\bar{\boldsymbol{X}}_k^{(\mathrm{reset})}, \bar{\boldsymbol{X}}_k^{(\mathrm{update})}, \bar{\boldsymbol{X}}_k^{(\mathrm{hidden})}$. The corresponding weights are indexed by $\ell_{\mathrm{inp},\bullet}$ or $\ell_{\mathrm{hid},\bullet}$, Here, the update steps considered in the GRU at the $k^{th}$ round of propagation are:

$$\bar{\boldsymbol{X}}_k^{(\mathrm{reset})} = \sigma\left(\boldsymbol{X}_k\boldsymbol{\Theta}^{(\ell_{\mathrm{inp},1})} + \bar{\boldsymbol{X}}_k^{(\ell_1)}\boldsymbol{\Theta}^{(\ell_{\mathrm{hid},1})}\right) \tag{15}$$

$$\bar{\boldsymbol{X}}_k^{(\mathrm{update})} = \sigma\left(\boldsymbol{X}_k\boldsymbol{\Theta}^{(\ell_{\mathrm{inp},2})} + \bar{\boldsymbol{X}}_k^{(\ell_1)}\boldsymbol{\Theta}^{(\ell_{\mathrm{hid},2})}\right) \tag{16}$$

$$\bar{\boldsymbol{X}}^{(\mathrm{hidden})} = \tanh\left(\boldsymbol{X}_k\boldsymbol{\Theta}^{(\ell_{\mathrm{inp},3})} + (\bar{\boldsymbol{X}}_k^{(\ell_1)} \odot \bar{\boldsymbol{X}}_k^{(\mathrm{update})})\boldsymbol{\Theta}^{(\ell_{\mathrm{hid},3})}\right) \tag{17}$$

$$\boldsymbol{X}_{k+1} = (1 - \bar{\boldsymbol{X}}_k^{(\mathrm{reset})}) \odot \boldsymbol{X}_k + \bar{\boldsymbol{X}}_k^{(\mathrm{reset})} \odot \bar{\boldsymbol{X}}_k^{(\mathrm{hidden})} \tag{18}$$

We define our transformation as

$$\Gamma_{\boldsymbol{\pi}}^{(0)}(\boldsymbol{\Theta}^{(\ell_0)}) = \boldsymbol{\Theta}^{(\ell_0)}\boldsymbol{\pi} \qquad \Gamma_{\boldsymbol{\pi}}^{(\ell_1)}(\boldsymbol{\Theta}^{(\ell_1)}) = \begin{bmatrix} \boldsymbol{\pi}^{\top} & \boldsymbol{0} \\ \boldsymbol{0} & \boldsymbol{\pi}^{\top} \end{bmatrix} \boldsymbol{\Theta}^{(\ell_1)} \tag{19}$$

$$\Gamma_{\boldsymbol{\pi}}^{(\ell_{\mathrm{inp},\bullet})}(\boldsymbol{\Theta}^{(\ell_{\mathrm{inp},\bullet})}) = \boldsymbol{\pi}^{\top}\boldsymbol{\Theta}^{(\ell_{\mathrm{inp},\bullet})}\boldsymbol{\pi} \qquad \Gamma_{\boldsymbol{\pi}}^{(\ell_{\mathrm{hid},\bullet})}(\boldsymbol{\Theta}^{\ell_{\mathrm{hid},\bullet}}) = \boldsymbol{\Theta}^{(\ell_{\mathrm{hid},\bullet})}\boldsymbol{\pi} \tag{20}$$

While the remaining transformations are identity.

Like the previous proof, we trace the computations in the network inductively over the propagation rounds.

*Base case.* For $k = 0$, this is true just like the previous case. $\boldsymbol{X}_0 \mapsto \boldsymbol{X}_0\boldsymbol{\pi}$ as $\boldsymbol{\Theta}^{(\ell_0)} \mapsto \boldsymbol{\Theta}^{(\ell_0)}\boldsymbol{\pi}$.

*Inductive Step.* Suppose $\boldsymbol{X}_k \mapsto \boldsymbol{X}_k\boldsymbol{\pi}$ for a value of $k$. Then $\bar{\boldsymbol{X}}_k^{(0)} = [\boldsymbol{x}_k^{\top}(u), \; \boldsymbol{x}_k^{\top}(v)] \mapsto \bar{\boldsymbol{X}}_k^{(0)} \begin{bmatrix} \boldsymbol{\pi} & \boldsymbol{0} \\ \boldsymbol{0} & \boldsymbol{\pi} \end{bmatrix}$ under $\theta \mapsto \Gamma_{\boldsymbol{\pi}}(\theta)$. Since, we transform $\boldsymbol{\Theta}^{(\ell_1)} \mapsto \begin{bmatrix} \boldsymbol{\pi}^{\top} & \boldsymbol{0} \\ \boldsymbol{0} & \boldsymbol{\pi}^{\top} \end{bmatrix} \boldsymbol{\Theta}^{(\ell_1)}$ and $\bar{\boldsymbol{X}}_k^{(0)} \mapsto \bar{\boldsymbol{X}}_k^{(0)} \begin{bmatrix} \boldsymbol{\pi} & \boldsymbol{0} \\ \boldsymbol{0} & \boldsymbol{\pi} \end{bmatrix}$, the quantity $\bar{\boldsymbol{X}}_k^{(\ell_1)} \mapsto \bar{\boldsymbol{X}}_k^{(0)} \begin{bmatrix} \boldsymbol{\pi} & \boldsymbol{0} \\ \boldsymbol{0} & \boldsymbol{\pi} \end{bmatrix} \begin{bmatrix} \boldsymbol{\pi}^{\top} & \boldsymbol{0} \\ \boldsymbol{0} & \boldsymbol{\pi}^{\top} \end{bmatrix} \boldsymbol{\Theta}^{(\ell_1)} = \bar{\boldsymbol{X}}_k^{(0)}\boldsymbol{\Theta}^{(\ell_1)}$ remains unchanged as $\boldsymbol{\pi}\boldsymbol{\pi}^{\top} = \boldsymbol{I}$.

Due to the transformations in Eq. (20), we have: (1) $\boldsymbol{X}_k\boldsymbol{\Theta}^{(\ell_{\mathrm{inp},i})} \mapsto \boldsymbol{X}_k\boldsymbol{\pi}\boldsymbol{\pi}^{\top}\boldsymbol{\Theta}^{(\ell_{\mathrm{inp},i})}\boldsymbol{\pi} = \boldsymbol{X}_k\boldsymbol{\Theta}^{(\ell_{\mathrm{inp},i})}\boldsymbol{\pi}$, for each $i = 1, 2, 3$; and, (2) $\bar{\boldsymbol{X}}^{(\ell_i)}\boldsymbol{\Theta}^{(\ell_{\mathrm{hid},i})} \mapsto \bar{\boldsymbol{X}}^{(\ell)}\boldsymbol{\Theta}^{(\ell_{\mathrm{hid},i})}\boldsymbol{\pi}$ for each $i = 1, 2$.

Consequently $\bar{X}^{(\text{reset})}, \bar{X}^{(\text{update})}, \bar{X}^{(\text{hidden})} \mapsto \bar{X}^{(\text{reset})}\pi, \bar{X}^{(\text{update})}\pi, \bar{X}^{(\text{hidden})}\pi$, resulting in $X_{k+1} \mapsto X_{k+1}\pi$ as follows:

$$\bar{X}^{(\text{reset})} \mapsto \sigma\left(X_k\Theta^{(\ell_{\text{inp},1})}\pi + \bar{X}^{(\ell_1)}\Theta^{(\ell_{\text{hid},1})}\pi\right) \tag{21}$$

$$= \sigma\left(X_k\Theta^{(\ell_{\text{inp},1})} + \bar{X}^{(\ell_1)}\Theta^{(\ell_{\text{hid},1})}\right)\pi = \bar{X}^{(\text{reset})}\pi \tag{22}$$

$$\bar{X}^{(\text{update})} \mapsto \sigma\left(X_k\Theta^{(\ell_{\text{inp},2})}\pi + \bar{X}^{(\ell_1)}\Theta^{(\ell_{\text{hid},2})}\pi\right) \tag{23}$$

$$= \sigma\left(X_k\Theta^{(\ell_{\text{inp},2})} + \bar{X}^{(\ell_1)}\Theta^{(\ell_{\text{hid},2})}\right)\pi = \bar{X}^{(\text{update})}\pi \tag{24}$$

$$\bar{X}^{(\text{hidden})} \mapsto \tanh\left(X_k\Theta^{(\ell_{\text{inp},3})}\pi + (\bar{X}^{(\ell_1)} \odot \bar{X}^{(\text{update})}\pi)\Theta^{(\ell_{\text{hid},3})}\pi\right) \tag{25}$$

$$= \tanh\left(X_k\Theta^{(\ell_{\text{inp},3})} + (\bar{X}^{(\ell_1)} \odot \bar{X}^{(\text{update})})\Theta^{(\ell_{\text{hid},3})}\right)\pi = \bar{X}^{(\text{hidden})}\pi \tag{26}$$

Therefore we will have:

$$X_{k+1} \mapsto (1 - \bar{X}^{(\text{reset})}\pi) \odot X_k\pi + \bar{X}^{(\text{reset})}\pi \odot \bar{X}^{(\text{hidden})}\pi \tag{27}$$

$$= \left((1 - \bar{X}^{(\text{reset})}) \odot X_k + \bar{X}^{(\text{reset})} \odot \bar{X}^{(\text{hidden})}\right)\pi = X_{k+1}\pi \tag{28}$$

∎

### E.1.2 PROOF OF LEMMA 3

**Lemma 3.** *Given the setting described in Section 3.1. Let $\Gamma_{\pi}$ be the transformation on the GNN parameters $\theta$, induced by a permutation $\pi \in \mathbb{R}^D$, as introduced in Lemma 2. Then the gradient of the loss is equivariant under transformation $\Gamma_{\pi}$ of the parameters.*

**Proof:** *Outline.* We assume that the loss is differentiable with respect to each parameter. We shall work with a finite difference of $\Delta$ as a proxy for the gradient. We show that that equivariance holds for this setup. Thus, the equivariance holds in the limiting case $\Delta \to 0$, hence in the case of gradients.

We shall make the following observation in order to prove the lemma: **For every layer, the transformation consists of a permutation of its entries.** This also makes $\Gamma_{\pi}$ linear.

*Additional Notation to Facilitate the Proof.* Corresponding to each layer $\ell$ and each *scalar* parameter $\Theta_t^{(\ell)}[j,k]$, we shall consider a perturbation of the parameter by $\Delta \in \mathbb{R} - \{0\}$. Within this proof, $\Delta$ is a perturbation and *not* relevance distance. Finally, $\delta_{\Delta,(k,l)}$ is defined as the matrix of appropriate dimensions with all zeros except for a $\Delta$ at the $(k,l)$-th position.

We write $\theta_t +_{\ell} \delta_{\Delta,(j,k)} = \left(\Theta_t^{(\ell')} + \delta_{\Delta,(j,k)}[\![\ell' = \ell]\!]\right)_{\ell' \in [\ell_{\max}]}$. This indicates the perturbation only at $(j,k)$-th entry of $\Theta_t^{(\ell')}$ at $\ell' = \ell$. We define the matrix of discrete differences as $\mathcal{L}_{t,\Delta}^{(\ell)}$ as

$$\mathcal{L}_{t,\Delta}^{(\ell)}[j,k] = \frac{1}{\Delta}\left[\text{loss}_t(\theta_t +_{\ell} \delta_{\Delta,(j,k)}) - \text{loss}_t(\theta_t)\right]. \tag{29}$$

First, we show that when $\theta_t \mapsto \Gamma_{\pi}(\theta_t)$, the transformation $\mathcal{L}_t \mapsto \Gamma_{\pi}(\mathcal{L}_t)$ will hold true. To show this, we derive that for a general $\ell \in [\ell_{\max}]$, $\mathcal{L}_{t,\Delta}^{(\ell)} \mapsto \Gamma_{\pi}^{(\ell)}(\mathcal{L}_{t,\Delta}^{(\ell)})$. Let us characterize the permutation on the entries of the parameter corresponding to $\Gamma_{\pi}^{(\ell)}$ by introducing a permutation map $\widehat{\pi} : [m] \times [n] \to [m] \times [n]$. For any $\Theta_t^{(\ell)}$, there exists $\widehat{\pi}$ defined as above, such that: $\Gamma_{\pi}^{(\ell)}(\Theta_t^{(\ell)})[\widehat{\pi}(j,k)] = \Theta_t^{(\ell)}[j,k]$. Here, $\widehat{\pi}$ depends on $\ell$. However, we omit this for the sake of readability.

*Proof.* Note the following identities that hold as a consequence:

- For all $j, k$, we have:

$$\Gamma_{\pi}^{(\ell)}(\Theta^{(\ell)})[j,k] = \Theta^{(\ell)}[\widehat{\pi}^{-1}(j,k)] \tag{30}$$

- Consider the $(a,b)^{\text{th}}$ entry of the following matrix: $\Gamma_{\pi}^{(\ell)}\delta_{\Delta,(j,k)}[a,b] = \delta_{\Delta,(j,k)}[\widehat{\pi}^{-1}(a,b)]$, which is $\Delta$ if $a, b = \widehat{\pi}(j,k)$ and 0, otherwise. Then, by definition of $\delta_{\Delta,(\bullet,\bullet)}$, we have:

$$\Gamma_{\pi}^{(\ell)}\delta_{\Delta,(j,k)} = \delta_{\Delta,(\widehat{\pi}(j,k))} \tag{31}$$

The transformation $\Gamma_{\boldsymbol{\pi}}$ is linear, which implies that $\Gamma_{\boldsymbol{\pi}}(\theta +_\ell \boldsymbol{\delta}_{\Delta,(\bullet)}) = \Gamma_{\boldsymbol{\pi}}(\theta) +_\ell \Gamma_{\boldsymbol{\pi}}^{(\ell)}(\boldsymbol{\delta}_{\Delta,(\bullet)})$. Consider the $(a,b)$-th entry of $\widehat{\boldsymbol{\mathcal{L}}}_{t,\Delta}^{(\ell)} = \boldsymbol{\mathcal{L}}_{t,\Delta}^{(\ell)}\big|_{\theta_t \mapsto \Gamma_{\boldsymbol{\pi}}(\theta_t)}$ which is the loss:

$$\widehat{\boldsymbol{\mathcal{L}}}_{t,\Delta}^{(\ell)}[a,b] = \frac{1}{\Delta}\left[\text{loss}_t(\Gamma_{\boldsymbol{\pi}}(\theta_t) +_\ell \boldsymbol{\delta}_{\Delta,(a,b)}) - \text{loss}_t(\Gamma_{\boldsymbol{\pi}}(\theta_t))\right] \tag{32}$$

$$= \frac{1}{\Delta}\left[\text{loss}_t(\Gamma_{\boldsymbol{\pi}}(\theta_t) +_\ell \Gamma_{\boldsymbol{\pi}}^{(\ell)} \circ \Gamma_{\boldsymbol{\pi}}^{(\ell)^{-1}}\boldsymbol{\delta}_{\Delta,(a,b)}) - \text{loss}_t(\Gamma_{\boldsymbol{\pi}}(\theta_t))\right] \tag{33}$$

$$= \frac{1}{\Delta}\left[\text{loss}_t(\Gamma_{\boldsymbol{\pi}}(\theta_t +_\ell \Gamma_{\boldsymbol{\pi}}^{(\ell)^{-1}}(\boldsymbol{\delta}_{\Delta,(a,b)}))) - \text{loss}_t(\Gamma_{\boldsymbol{\pi}}\theta_t)\right] \tag{34}$$

$$= \frac{1}{\Delta}\left[\text{loss}_t(\theta_t +_\ell \Gamma_{\boldsymbol{\pi}}^{(\ell)^{-1}}(\boldsymbol{\delta}_{\Delta,(a,b)})) - \text{loss}_t(\theta_t)\right] \quad \text{(as the loss is invariant of } \Gamma_{\boldsymbol{\pi}}) \tag{35}$$

$$= \frac{1}{\Delta}\left[\text{loss}_t(\theta_t +_\ell \boldsymbol{\delta}_{\Delta,(\widehat{\pi}^{-1}(a,b))}) - \text{loss}_t(\theta_t)\right] \quad \text{from Eq. (31)} \tag{36}$$

$$= \boldsymbol{\mathcal{L}}_{t,\Delta}^{(\ell)}[\widehat{\pi}^{-1}(a,b)] = \Gamma_{\boldsymbol{\pi}}^{(\ell)}(\boldsymbol{\mathcal{L}}_{t,\Delta}^{(\ell)})[a,b] \quad \text{from Eq. (30)} \tag{37}$$

Thus, $\widehat{\boldsymbol{\mathcal{L}}}_{t,\Delta}^{(\ell)} = \Gamma_{\boldsymbol{\pi}}^{(\ell)}(\boldsymbol{\mathcal{L}}_{t,\Delta}^{(\ell)})$. Now, $\lim_{\Delta \to 0} \boldsymbol{\mathcal{L}}_{t,\Delta}^{(\ell)} = \mathbf{grad}_t^{(\ell)}$. Hence, we have:

$$\lim_{\Delta \to 0}\widehat{\boldsymbol{\mathcal{L}}}_{t,\Delta}^{(\ell)} = \lim_{\Delta \to 0}\Gamma_{\boldsymbol{\pi}}^{(\ell)}(\boldsymbol{\mathcal{L}}_{t,\Delta}^{(\ell)}) \tag{38}$$

$$= \Gamma_{\boldsymbol{\pi}}^{(\ell)}\left(\lim_{\Delta \to 0}\boldsymbol{\mathcal{L}}_{t,\Delta}^{(\ell)}\right) \quad (\Gamma_{\boldsymbol{\pi}}^{(\ell)} \text{ is a smooth map}) \tag{39}$$

$$= \Gamma_{\boldsymbol{\pi}}^{(\ell)}(\mathbf{grad}_t^\ell) \tag{40}$$

Therefore as $\boldsymbol{\Theta}_t^{(\ell)} \mapsto \Gamma_{\boldsymbol{\pi}}^{(\ell)}(\boldsymbol{\Theta}_t^\ell)$, we have $\mathbf{grad}_t^\ell \mapsto \Gamma_{\boldsymbol{\pi}}^{(\ell)}(\mathbf{grad}_t^\ell)$. Hence, $\mathbf{grad}_t = [\mathbf{grad}_t^\ell]_\ell \mapsto [\Gamma_{\boldsymbol{\pi}}^{(\ell)}(\mathbf{grad}_t^\ell)]_\ell = \Gamma_{\boldsymbol{\pi}}([\mathbf{grad}_t^\ell]_\ell) = \Gamma_{\boldsymbol{\pi}}(\mathbf{grad}_t)$. ∎

### E.1.3 PROOF OF LEMMA 4

**Lemma 4.** *Given the setting described in Section 3.1. Let $\{\theta_t \,|\, t \geq 0\}$ be the trajectory of the parameter $\theta$ of a GNN across different training epochs $t \geq 0$. Then, we have: $p(\theta_t) = p(\Gamma_{\boldsymbol{\pi}}(\theta_t))$ for all $t \geq 0$.*

**Proof:** For iter $= 0$, we have $p(\theta_0) = p(\Gamma_{\boldsymbol{\pi}}(\theta_0))$ by the i.i.d. initialization of parameters. For iter $> 0$, we use two key conditions: (1) The loss function is invariant under $\Gamma_{\boldsymbol{\pi}}$ (which holds, as our loss is permutation invariant in the GNN output). (2) The gradient and update steps are equivariant under $\Gamma_{\boldsymbol{\pi}}$. We first note that:

$$p(\theta_t) = \int_{\underbrace{\mathcal{J} \times \ldots \times \mathcal{J}}_{t \text{ times}}} \prod_{\text{iter}=1}^{t} p(\theta_{\text{iter}} \,|\, \theta_{<\text{iter}})\, d\theta_{<t} \tag{41}$$

First, to build up intuition, consider a simpler setup which, instead of using an advanced optimizer like Adam/SGD, uses simple full batch gradient descent. Assuming the learning rate is 1, we will have:

$$\boldsymbol{\Theta}_{\text{iter}}^{(\ell)} = \boldsymbol{\Theta}_{\text{iter}-1}^{(\ell)} - \mathbf{grad}^\ell\big|_{\boldsymbol{\Theta}=\boldsymbol{\Theta}_{\text{iter}-1}^{(\ell)}} \tag{42}$$

Hence, $p(\theta_{\text{iter}} \,|\, \theta_{<\text{iter}})$ is given by:

$$p(\theta_{\text{iter}} \,|\, \theta_{<\text{iter}}) = \delta(\theta_{\text{iter}} - \theta_{\text{iter}-1} + \text{grad}_{\text{iter}-1}) \tag{43}$$

Since $\Gamma_{\boldsymbol{\pi}}^{(\ell)}$ is a linear homeomorphism, we have

$$\Gamma_{\boldsymbol{\pi}}^{(\ell)}(\boldsymbol{\Theta}_{\text{iter}}^{(\ell)}) = \Gamma_{\boldsymbol{\pi}}^{(\ell)}(\boldsymbol{\Theta}_{\text{iter}-1}^{(\ell)}) - \Gamma_{\boldsymbol{\pi}}^{(\ell)}\left(\mathbf{grad}^\ell\big|_{\boldsymbol{\Theta}=\boldsymbol{\Theta}_{\text{iter}-1}^{(\ell)}}\right) \tag{44}$$

$$= \Gamma_{\boldsymbol{\pi}}^{(\ell)}(\boldsymbol{\Theta}_{\text{iter}-1}^{(\ell)}) - \mathbf{grad}^\ell\big|_{\boldsymbol{\Theta}=\Gamma_{\boldsymbol{\pi}}^{(\ell)}(\boldsymbol{\Theta}_{\text{iter}-1}^{(\ell)})} \quad \text{(Lemma 3)} \tag{45}$$

Given $\Gamma_{\boldsymbol{\pi}}(\theta) = \bigoplus_\ell \Gamma_{\boldsymbol{\pi}}^{(\ell)}(\boldsymbol{\Theta}^{(\ell)})$

$$\Gamma_{\boldsymbol{\pi}}(\theta_{\text{iter}}) = \Gamma_{\boldsymbol{\pi}}(\theta_{\text{iter}-1}) - \text{grad}\big|_{\theta=\Gamma_{\boldsymbol{\pi}}(\theta_{\text{iter}-1})} \tag{46}$$

This allows us to write:

$$p(\Gamma_{\boldsymbol{\pi}}(\theta_{\text{iter}}) \,|\, \Gamma_{\boldsymbol{\pi}}(\theta_{<\text{iter}})) = \delta(\Gamma_{\boldsymbol{\pi}}(\theta_{\text{iter}}) - \Gamma_{\boldsymbol{\pi}}(\theta_{\text{iter}-1}) + \Gamma_{\boldsymbol{\pi}}(\text{grad}_{\text{iter}-1})) \tag{47}$$

Now, since Eq. (42) and Eq. (46) are equivalent, we have

$$p(\theta_{\text{iter}} \,|\, \theta_{<\text{iter}}) = p(\Gamma_{\boldsymbol{\pi}}(\theta_{\text{iter}}) \,|\, \Gamma_{\boldsymbol{\pi}}(\theta_{<\text{iter}})) \tag{48}$$

The above relationship suggests Eq. (41) is equivalent to

$$p(\theta_t) = \int_{\underbrace{\mathcal{J} \times \ldots \times \mathcal{J}}_{t \text{ times}}} \prod_{\text{iter}=1}^{t} p(\Gamma_{\boldsymbol{\pi}}(\theta_{\text{iter}}) \,|\, \Gamma_{\boldsymbol{\pi}}(\theta_{<\text{iter}})) \, d\theta_{\text{iter}} \tag{49}$$

$$= \int_{(\Gamma_{\boldsymbol{\pi}} \circ \mathcal{J})^t} \prod_{\text{iter}=0}^{t} p(\Gamma_{\boldsymbol{\pi}}(\theta_{\text{iter}}) \,|\, \Gamma_{\boldsymbol{\pi}}(\theta_{<\text{iter}})) \, d(\Gamma_{\boldsymbol{\pi}}(\theta_{\text{iter}})) \left| \text{Det} \left( \frac{\partial \theta_{\text{iter}}}{\partial \Gamma_{\boldsymbol{\pi}}(\theta_{\text{iter}})} \right) \right|^{=1} \tag{50}$$

$$= p(\Gamma_{\boldsymbol{\pi}}(\theta_t)) \tag{51}$$

$\left| \text{Det} \left( \frac{\partial \theta_{\text{iter}}}{\partial \Gamma_{\boldsymbol{\pi}}(\theta_{\text{iter}})} \right) \right| = 1$ because $\Gamma_{\boldsymbol{\pi}}$ consists only of permutation matrices. Here, we proved that Eq. (42) and Eq. (46) are equivalent for full batch gradient descent. This relationship also holds for other standard optimizers (such as listed in E.1.5), which is shown below. We may abstract the update step as follows –

$$\boldsymbol{\Theta}_{\text{iter}}^{(\ell)} = \text{Update}_{\ell, \text{iter}} \left( \left( \boldsymbol{\Theta}_b^{(\ell)} \,|\, b < \text{iter} \right), \left( \mathbf{grad}_b^{(\ell)} \,|\, b < \text{iter} \right) \right) \tag{52}$$

This gives: $p(\theta_{\text{iter}} \,|\, \theta_{<\text{iter}}) = \prod_{\ell} \delta \left( \left[ \boldsymbol{\Theta}_{\text{iter}}^{(\ell)} - \text{Update}_{\ell, \text{iter}} \left( \left( \boldsymbol{\Theta}_b^{(\ell)} \,|\, b < \text{iter} \right), \left( \mathbf{grad}_b^{(\ell)} \,|\, b < \text{iter} \right) \right) \right] \right)$

$$\tag{53}$$

According to Lemma 10, Eq. (52) is equivalent to:

$$\Gamma_{\boldsymbol{\pi}}^{(\ell)}(\boldsymbol{\Theta}_{\text{iter}}^{(\ell)}) = \text{Update}_{\ell, \text{iter}} \left( \left( \Gamma_{\boldsymbol{\pi}}^{(\ell)}(\boldsymbol{\Theta}_b^{(\ell)}) \,|\, b < \text{iter} \right), \left( \Gamma_{\boldsymbol{\pi}}^{(\ell)}(\mathbf{grad}_b^{(\ell)}) \,|\, b < \text{iter} \right) \right) \tag{54}$$

as long as $\Gamma_{\boldsymbol{\pi}}^{(\ell)}$ is a permutation matrix (which is the case according to Lemma 2). This implies that $p(\theta_{\text{iter}} \,|\, \theta_{<\text{iter}})$ (53) is the same as:

$$p(\Gamma_{\boldsymbol{\pi}}(\theta_{\text{iter}}) \,|\, \Gamma_{\boldsymbol{\pi}}(\theta_{<\text{iter}}))$$
$$= \prod_{\ell} \delta \left( \Gamma_{\boldsymbol{\pi}}^{(\ell)}(\boldsymbol{\Theta}_{\text{iter}}^{(\ell)}) - \text{Update}_{\ell, \text{iter}} \left( \left( \Gamma_{\boldsymbol{\pi}}^{(\ell)}(\boldsymbol{\Theta}_b^{(\ell)}) \,|\, b < \text{iter} \right), \left( \Gamma_{\boldsymbol{\pi}}^{(\ell)}(\mathbf{grad}_b^{(\ell)}) \,|\, b < \text{iter} \right) \right) \right)$$
$$\tag{55}$$

### E.1.4 PROOF OF THEOREM 5 AND PROPOSITION 6

We state both the results.

**Theorem 5.** *Given the setting described in Section 3.1. Then, $\boldsymbol{X} = \text{GNN}_\theta(G)$ are exchangeable random variables, where the randomness is induced by the model initialization prior to training. That is, $p(\boldsymbol{X}) = p(\boldsymbol{X}\boldsymbol{\pi})$.*

**Proposition 6.** *Given two graphs $G_q, G_c$, let the settings in Section 3.1 hold true. Specifically, let us assume that the loss function be invariant to simultaneous permutations of the embeddings $\boldsymbol{X}^{(q)} = \text{GNN}_\theta(G_q)$ and $\boldsymbol{X}^{(c)} = \text{GNN}_\theta(G_c)$. Then, $\boldsymbol{Y} = [\boldsymbol{X}^{(q)}; \boldsymbol{X}^{(c)}] \in \mathbb{R}^{2n \times D}$ satisfies $p(\boldsymbol{Y}) = p(\boldsymbol{Y}\boldsymbol{\pi})$.*

We shall prove both of these in one go, as the latter implies the former.

**Proof:** Let $\boldsymbol{Y}$ denote the concatenation of the query and corpus embeddings, i.e., $\boldsymbol{Y} = \begin{bmatrix} \boldsymbol{X}^{(q)} \\ \boldsymbol{X}^{(c)} \end{bmatrix}$, where $\boldsymbol{X}^{(\bullet)} \in \mathbb{R}^{m \times D}$. We need to show that:

$$p(\boldsymbol{Y}) = p(\boldsymbol{Y}\boldsymbol{\pi}). \tag{56}$$

This is precisely the condition for exchangeability as stated in Definition 1. We first observe that:

$$p(\boldsymbol{Y}) = \int_{\mathcal{J}} p(\boldsymbol{Y} \,|\, \theta_t) \, p(\theta_t) \, d\theta_t \quad \text{(marginalization)} \tag{57}$$

$$= \int_{\mathcal{J}} p(\boldsymbol{Y}\boldsymbol{\pi} \,|\, \Gamma_{\boldsymbol{\pi}}(\theta_t)) \, p(\theta_t) d\theta_t \quad \text{(using } p(\boldsymbol{Y} \,|\, \theta_t) = p(\boldsymbol{Y}\boldsymbol{\pi} \,|\, \Gamma_{\boldsymbol{\pi}}(\theta_t))) \tag{58}$$

$$= \int_{\mathcal{J}} p(\boldsymbol{Y}\boldsymbol{\pi} \,|\, \Gamma_{\boldsymbol{\pi}}(\theta_t)) p(\Gamma_{\boldsymbol{\pi}}(\theta_t)) d\theta_t \quad \text{(using } p(\theta_t) = p(\Gamma_{\boldsymbol{\pi}}(\theta_t))) \tag{59}$$

$$= \int_{\Gamma_{\boldsymbol{\pi}} \circ \mathcal{J} = \mathcal{J}} p(\boldsymbol{Y}\boldsymbol{\pi} \,|\, \Gamma_{\boldsymbol{\pi}}(\theta_t)) p(\Gamma_{\boldsymbol{\pi}}(\theta_t)) d(\Gamma_{\boldsymbol{\pi}}(\theta_t)) \left| \frac{\partial \theta_t}{\partial \Gamma_{\boldsymbol{\pi}}(\theta_t)} \right|$$
$$\text{(Random variable transform } \theta_t \mapsto \Gamma_{\boldsymbol{\pi}}\theta_t) \tag{60}$$

$$= \int_{\mathcal{J}} p(\boldsymbol{Y}\boldsymbol{\pi} \,|\, \Gamma_{\boldsymbol{\pi}}(\theta_t)) p(\Gamma_{\boldsymbol{\pi}}(\theta_t)) d(\Gamma_{\boldsymbol{\pi}}(\theta_t)) \cdot 1 = p(\boldsymbol{Y}\boldsymbol{\pi}) \quad \text{(marginalization)} \tag{61}$$

Justifications of Eqs (57), (61) are trivial. We now provide justifications for the claims in Eq. (58) and Eq. (59) are as follows.

**Justification for** $p(\boldsymbol{Y} \,|\, \theta_t) = p(\boldsymbol{Y}\boldsymbol{\pi} \,|\, \Gamma_{\boldsymbol{\pi}}(\theta_t))$ **used in Eq. (58):** As the network output is deterministic, $p(\boldsymbol{Y} \,|\, \theta_t)$ can be written in terms of the network output $\text{GNN}_\theta$ and the Dirac delta function as follows:

$$p(\boldsymbol{Y} \,|\, \theta_t) = \delta \left( \boldsymbol{Y} - \begin{bmatrix} \text{GNN}_{\theta_t}(G_q) \\ \text{GNN}_{\theta_t}(G_c) \end{bmatrix} \right) \tag{62}$$

Here $\delta(\bullet)$ is the Diract delta functional. $\delta(\bullet) = \begin{cases} \infty & \text{if } \boldsymbol{Z} = \boldsymbol{0} \\ 0 & \text{otherwise} \end{cases}$ and $\int_{\mathcal{J}} \delta(\boldsymbol{Z}) d\boldsymbol{Z} = 1$.

Since the following relation holds: $\boldsymbol{Y} = \begin{bmatrix} \text{GNN}_{\theta_t}(G_q) \\ \text{GNN}_{\theta_t}(G_c) \end{bmatrix}$ iff $\boldsymbol{Y}\boldsymbol{\pi} = \begin{bmatrix} \text{GNN}_{\Gamma_{\boldsymbol{\pi}}(\theta_t)}(G_q) \\ \text{GNN}_{\Gamma_{\boldsymbol{\pi}}(\theta_t)}(G_c) \end{bmatrix}$, we have $p(\boldsymbol{Y} \,|\, \theta_t) = p(\boldsymbol{Y}\boldsymbol{\pi} \,|\, \Gamma_{\boldsymbol{\pi}}(\theta_t))$. Justification for $p(\theta_t) = p(\Gamma_{\boldsymbol{\pi}}(\theta_t))$ in Eq. (59) occurs due to Lemma 4.

Here, we note that our result holds even in the presence of additional sources of randomness in the training process, such as data shuffling or batching. Since these sources are independent of parameter initialization, the proof extends by conditioning on the training randomness and then marginalizing, yielding the same conclusion.

### E.1.5 EQUIVARIANCE OF THE UPDATE STEP

We shall present a general lemma that states the precise update step equivariance property. Later, we will prove it for optimizers such as Adam, SGD, AdaGrad, RMSProp, followed by a more general general formulation.

**Lemma 10** (Equivariance of update step). *The update steps of the optimizer follow the functional form and equivariance property. Specifically Eq. (63) holds true iff Eq. (64) holds true.*

$$\boldsymbol{\Theta}_t^{(\ell)} \triangleq \mathrm{Update}_{\ell,\mathrm{t}} \left( \left( \boldsymbol{\Theta}_{\mathrm{iter}}^{(\ell)} \,|\, \mathrm{iter} < t \right), \left( \mathbf{grad}_{\mathrm{iter}}^{(\ell)} \,|\, \mathrm{iter} < t \right) \right) \tag{63}$$

$$\boldsymbol{\pi}_1 \boldsymbol{\Theta}_t^{(\ell)} \boldsymbol{\pi}_2 = \mathrm{Update}_{\ell,\mathrm{t}} \left( \left( \boldsymbol{\pi}_1 \boldsymbol{\Theta}_{\mathrm{iter}}^{(\ell)} \boldsymbol{\pi}_2 \,|\, \mathrm{iter} < t \right), \left( \boldsymbol{\pi}_1 \mathbf{grad}_{\mathrm{iter}}^{(\ell)} \boldsymbol{\pi}_2 \,|\, \mathrm{iter} < t \right) \right) \tag{64}$$

Note that this means that the update step is equivariant with respect to a transformation that permutes the rows and columns of each parameter matrix. The transformation $\boldsymbol{\pi}_1$ permutes the rows of the parameter matrix, while $\boldsymbol{\pi}_2$ permutes the columns.

**Proof for Adam (Kingma et al., 2015)** We first describe the Adam update steps — For layer $\ell$ at time $t$, we refer to the momentum of the gradients $\boldsymbol{m}_t^\ell$, and the squared gradients $\boldsymbol{v}_t^\ell$. The corresponding bias-corrected terms which used by Adam are denoted by $\widehat{\boldsymbol{m}}_t^\ell$ and $\widehat{\boldsymbol{v}}_t^\ell$ respectively.

The hyperparameters for Adam are defined as follows: $\beta_1$ and $\beta_2$ are scalar coefficients that control the exponential moving averages of the gradient and its square. $\alpha$ denotes the learning rate. $\epsilon$ is a small positive constant added for numerical stability. $\lambda$ is the weight decay parameter.

The Adam optimizer (Kingma et al., 2014) updates each parameter as follows:

$$\boldsymbol{\Theta}_t^{(\ell)} = \boldsymbol{\Theta}_{t-1}^{(\ell)} - \alpha \frac{\widehat{\boldsymbol{m}}_t^\ell}{\sqrt{\widehat{\boldsymbol{v}}_t^\ell} + \epsilon} \tag{65}$$

$$\boldsymbol{g}_t^{(\ell)} = \mathbf{grad}_t^{(\ell)} + \lambda \boldsymbol{\Theta}_{t-1}^{(\ell)} \tag{66}$$

$$\widehat{\boldsymbol{m}}_t^\ell = \frac{\boldsymbol{m}_t^\ell}{1 - n^\top} \tag{67}$$

$$\boldsymbol{m}_t^\ell = \beta_1 \boldsymbol{m}_{t-1}^\ell + (1 - \beta_1) \boldsymbol{g}_t^{(\ell)} \tag{68}$$

$$\widehat{\boldsymbol{v}}_t^\ell = \frac{\boldsymbol{v}_t^\ell}{1 - \beta_2^\top} \tag{69}$$

$$\boldsymbol{v}_t^\ell = \beta_2 \boldsymbol{v}_{t-1}^\ell + (1 - \beta_2)(\boldsymbol{g}_t^{(\ell)} \odot \boldsymbol{g}_t^{(\ell)}) \tag{70}$$

Where $\boldsymbol{m}_0^\ell = \boldsymbol{v}_0^\ell = 0$.

Eq (63) can be represented by simply inductively writing out the update steps in terms of the previous steps using $\boldsymbol{\Theta}_{<t}^{(\ell)}$ and $\mathbf{grad}_{<t}^{(\ell)}$. Similarly for Eq. (64), we can show that each $\boldsymbol{v}_{\mathrm{iter}}^\ell$ and $\boldsymbol{m}_{\mathrm{iter}}^\ell$ are permutation equivariant with respect to the gradients, and consequently even $\widehat{\boldsymbol{m}}_{\mathrm{iter}}^\ell$ and $\widehat{\boldsymbol{v}}_{\mathrm{iter}}^\ell$. We shall work this out here–

Consider the transformation $\boldsymbol{\Theta}_{<t}^{(\ell)} \mapsto \boldsymbol{\pi}_1^\top (\boldsymbol{\Theta}_{<t}^{(\ell)}) \boldsymbol{\pi}_2$,

$$\mathbf{grad}_{<t}^{(\ell)} \mapsto \boldsymbol{\pi}_1^\top \mathbf{grad}_{<t}^{(\ell)} \boldsymbol{\pi}_2 \quad \text{(assumption, shown in Lemma 3)} \tag{71}$$

We show equivariance for $\boldsymbol{v}_t^\ell$ and $\boldsymbol{m}_t^\ell$ by induction–

$$\boldsymbol{g}_t^{(\ell)} \mapsto \boldsymbol{\pi}_1^\top (\boldsymbol{g}_t^{(\ell)}) \boldsymbol{\pi}_2 \tag{72}$$

$$\boldsymbol{v}_0^\ell = (1 - \beta_2)(\boldsymbol{g}_0^{(\ell)} \odot \boldsymbol{g}_0^{(\ell)}) \mapsto (1 - \beta_2)(\boldsymbol{\pi}_1^\top \boldsymbol{g}_0^{(\ell)} \boldsymbol{\pi}_2 \odot \boldsymbol{\pi}_1^\top \boldsymbol{g}_0^{(\ell)} \boldsymbol{\pi}_2) \tag{73}$$

$$= \boldsymbol{\pi}_1^\top (1 - \beta_2)(\boldsymbol{g}_0^{(\ell)} \odot \boldsymbol{g}_0^{(\ell)}) \boldsymbol{\pi}_2 = \boldsymbol{\pi}_1^\top \boldsymbol{v}_0^\ell \boldsymbol{\pi}_2 \tag{74}$$

$$\boldsymbol{v}_t^\ell = \beta_2 \boldsymbol{v}_{t-1}^\ell + (1 - \beta_2)(\boldsymbol{g}_t^{(\ell)} \odot \boldsymbol{g}_t^{(\ell)}) \mapsto \beta_2 \boldsymbol{\pi}_1^\top \boldsymbol{v}_{t-1}^\ell \boldsymbol{\pi}_2 + \boldsymbol{\pi}_1^\top (1 - \beta_2)(\boldsymbol{g}_t^{(\ell)} \odot \boldsymbol{g}_t^{(\ell)}) \boldsymbol{\pi}_2 \tag{75}$$

$$= \boldsymbol{\pi}_1^\top \boldsymbol{v}_t^\ell \boldsymbol{\pi}_2 \tag{76}$$

$$\boldsymbol{m}_0^\ell = (1 - \beta_1)(\boldsymbol{g}_0^{(\ell)}) \mapsto (1 - \beta_1)(\boldsymbol{\pi}_1^\top \boldsymbol{g}_0^{(\ell)} \boldsymbol{\pi}_2) \tag{77}$$

$$= \boldsymbol{\pi}_1^\top (1 - \beta_1)(\boldsymbol{g}_0^{(\ell)}) \boldsymbol{\pi}_2 = \boldsymbol{\pi}_1^\top \boldsymbol{m}_0^\ell \boldsymbol{\pi}_2 \tag{78}$$

$$\boldsymbol{m}_t^\ell = \beta_1 \boldsymbol{m}_{t-1}^\ell + (1 - \beta_1)(\boldsymbol{g}_t^{(\ell)}) \mapsto \beta_1 \boldsymbol{\pi}_1^\top \boldsymbol{m}_{t-1}^\ell \boldsymbol{\pi}_2 + \boldsymbol{\pi}_1^\top (1 - \beta_1)(\boldsymbol{g}_t^{(\ell)}) \boldsymbol{\pi}_2 \tag{79}$$

$$= \boldsymbol{\pi}_1^\top \boldsymbol{m}_t^\ell \boldsymbol{\pi}_2 \tag{80}$$

$$\widehat{\boldsymbol{v}}_t^\ell = \frac{\boldsymbol{v}_t^\ell}{1 - \beta_2^\top} \mapsto \frac{\boldsymbol{\pi}_1^\top \boldsymbol{v}_t^\ell \boldsymbol{\pi}_2}{1 - \beta_2^\top} = \boldsymbol{\pi}_1^\top \widehat{\boldsymbol{v}}_t^\ell \boldsymbol{\pi}_2 \tag{81}$$

$$\widehat{\boldsymbol{m}}_t^\ell = \frac{\boldsymbol{m}_t^\ell}{1 - \beta_1^\top} \mapsto \frac{\boldsymbol{\pi}_1^\top \boldsymbol{m}_t^\ell \boldsymbol{\pi}_2}{1 - \beta_1^\top} = \boldsymbol{\pi}_1^\top \widehat{\boldsymbol{m}}_t^\ell \boldsymbol{\pi}_2 \tag{82}$$

Finally, from (65), $\boldsymbol{\Theta}_t^{(\ell)}$ is permutation equivariant with respect to $\boldsymbol{\Theta}_{t-1}^{(\ell)}$ and the gradients.

$$\boldsymbol{\Theta}_t^{(\ell)} = \boldsymbol{\Theta}_{t-1}^{(\ell)} - \alpha \frac{\widehat{\boldsymbol{m}}_t^\ell}{\sqrt{\widehat{\boldsymbol{v}}_t^\ell} + \epsilon} \mapsto \boldsymbol{\pi}_1^\top \boldsymbol{\Theta}_{t-1}^{(\ell)} \boldsymbol{\pi}_2 - \alpha \frac{\boldsymbol{\pi}_1^\top \widehat{\boldsymbol{m}}_t^\ell \boldsymbol{\pi}_2}{\sqrt{\boldsymbol{\pi}_1^\top \widehat{\boldsymbol{v}}_t^\ell \boldsymbol{\pi}_2} + \epsilon} \tag{83}$$

$$= \boldsymbol{\pi}_1^\top \left( \boldsymbol{\Theta}_{t-1}^{(\ell)} - \alpha \frac{\widehat{\boldsymbol{m}}_t^\ell}{\sqrt{\widehat{\boldsymbol{v}}_t^\ell} + \epsilon} \right) \boldsymbol{\pi}_2 = \boldsymbol{\pi}_1^\top \boldsymbol{\Theta}_t^{(\ell)} \boldsymbol{\pi}_2 \tag{84}$$

∎

**Proof for SGD**    SGD has hyperparameters for learning rate $\alpha$, and weight decay $\lambda$. For layer $\ell$ at time $t$, the update step of SGD with weight decay is given by:

$$\boldsymbol{\Theta}_t^{(\ell)} = \boldsymbol{\Theta}_{t-1}^{(\ell)} - \alpha \boldsymbol{g}_t^{(\ell)} \tag{85}$$

$$\boldsymbol{g}_t^{(\ell)} = \mathbf{grad}_t^{(\ell)} + \lambda \boldsymbol{\Theta}_{t-1}^{(\ell)} \tag{86}$$

Where $\lambda$ is the weight decay term and $\alpha$ is the learning rate.

Here, the gradient is computed over a point/mini-batch of points sampled at time $t$. We can fix the randomness of the sampling by conditioning on the "trajectory" of sampled points(or mini-batches). Thus, we can treat $\mathbf{grad}_t^{(\ell)}$ as a deterministic function of $\boldsymbol{\Theta}_{<t}^{(\ell)}$.

Furthermore, this gradient also follows the gradient equivariance property from Lemma 3.

Consider the transformation $\boldsymbol{\Theta}_{t-1}^{(\ell)} \mapsto \boldsymbol{\pi}_1^\top (\boldsymbol{\Theta}_{t-1}^{(\ell)}) \boldsymbol{\pi}_2$ and $\mathbf{grad}_t^{(\ell)} \mapsto \boldsymbol{\pi}_1^\top (\mathbf{grad}_t^{(\ell)}) \boldsymbol{\pi}_2$. Then:

$$\boldsymbol{g}_t^{(\ell)} \mapsto \boldsymbol{\pi}_1^\top (\mathbf{grad}_t^{(\ell)}) \boldsymbol{\pi}_2 + \lambda \boldsymbol{\pi}_1^\top (\boldsymbol{\Theta}_{t-1}^{(\ell)}) \boldsymbol{\pi}_2 = \boldsymbol{\pi}_1^\top \boldsymbol{g}_t^{(\ell)} \boldsymbol{\pi}_2 \tag{87}$$

$$\boldsymbol{\Theta}_t^{(\ell)} \mapsto \boldsymbol{\pi}_1^\top \boldsymbol{\Theta}_{t-1}^{(\ell)} \boldsymbol{\pi}_2 - \alpha \boldsymbol{\pi}_1^\top \boldsymbol{g}_t^{(\ell)} \boldsymbol{\pi}_2 \tag{88}$$

$$= \boldsymbol{\pi}_1^\top (\boldsymbol{\Theta}_{t-1}^{(\ell)} - \alpha \boldsymbol{g}_t^{(\ell)}) \boldsymbol{\pi}_2 = \boldsymbol{\pi}_1^\top \boldsymbol{\Theta}_t^{(\ell)} \boldsymbol{\pi}_2 \tag{89}$$

Thus, the SGD update is equivariant with respect to the transformation. By conditioning on the trajectory, we actually show a stronger result for equivariance. We may show the equivariance without conditioning on the trajectory, by considering the expectation of the above result over the randomness of the sampling. □

**Proof for AdaGrad (Duchi et al., 2011)**    AdaGrad has hyperparameters for (time dependent) learning rate $\alpha_t$, weight decay $\lambda$, and a small constant $\epsilon$ for stability. For layer $\ell$ at time $t$, we refer to the accumulated squared gradients as $\boldsymbol{G}_t^{(\ell)}$ (which is defined below). The update steps for AdaGrad

are given by:

$$\Theta_t^{(\ell)} = \Theta_{t-1}^{(\ell)} - \frac{\alpha_t}{\sqrt{G_t^{(\ell)}} + \epsilon} \odot g_t^{(\ell)} \tag{90}$$

$$g_t^{(\ell)} = \mathbf{grad}_t^{(\ell)} + \lambda \Theta_{t-1}^{(\ell)} \tag{91}$$

$$G_t^{(\ell)} = G_{t-1}^{(\ell)} + (g_t^{(\ell)} \odot g_t^{(\ell)}) \tag{92}$$

Where $G_0^{(\ell)} = 0$.

Consider the transformation $\Theta_{<t}^{(\ell)} \mapsto \pi_1^\top(\Theta_{<t}^{(\ell)})\pi_2$ and $\mathbf{grad}_{<t}^{(\ell)} \mapsto \pi_1^\top(\mathbf{grad}_{<t}^{(\ell)})\pi_2$. We show that $G_t^{(\ell)}$ is equivariant by induction:

$$g_t^{(\ell)} \mapsto \pi_1^\top(g_t^{(\ell)})\pi_2 \tag{93}$$

$$G_0^{(\ell)} = 0 \mapsto \pi_1^\top 0 \pi_2 = 0 = \pi_1^\top G_0^{(\ell)}\pi_2 \tag{94}$$

$$G_t^{(\ell)} = G_{t-1}^{(\ell)} + (g_t^{(\ell)} \odot g_t^{(\ell)}) \tag{95}$$

$$\mapsto \pi_1^\top G_{t-1}^{(\ell)}\pi_2 + (\pi_1^\top g_t^{(\ell)}\pi_2 \odot \pi_1^\top g_t^{(\ell)}\pi_2) \tag{96}$$

$$= \pi_1^\top G_{t-1}^{(\ell)}\pi_2 + \pi_1^\top(g_t^{(\ell)} \odot g_t^{(\ell)})\pi_2 \tag{97}$$

$$= \pi_1^\top(G_{t-1}^{(\ell)} + g_t^{(\ell)} \odot g_t^{(\ell)})\pi_2 = \pi_1^\top G_t^{(\ell)}\pi_2 \tag{98}$$

Finally, for the weight update:

$$\Theta_t^{(\ell)} = \Theta_{t-1}^{(\ell)} - \frac{\alpha_t}{\sqrt{G_t^{(\ell)}} + \epsilon} \odot g_t^{(\ell)} \tag{99}$$

$$\mapsto \pi_1^\top \Theta_{t-1}^{(\ell)}\pi_2 - \frac{\alpha_t}{\sqrt{\pi_1^\top G_t^{(\ell)}\pi_2} + \epsilon} \odot \pi_1^\top g_t^{(\ell)}\pi_2 \tag{100}$$

$$= \pi_1^\top \Theta_{t-1}^{(\ell)}\pi_2 - \pi_1^\top\left(\frac{\alpha_t}{\sqrt{G_t^{(\ell)}} + \epsilon} \odot g_t^{(\ell)}\right)\pi_2 \tag{101}$$

$$= \pi_1^\top\left(\Theta_{t-1}^{(\ell)} - \frac{\alpha_t}{\sqrt{G_t^{(\ell)}} + \epsilon} \odot g_t^{(\ell)}\right)\pi_2 = \pi_1^\top \Theta_t^{(\ell)}\pi_2 \tag{102}$$

Thus, the AdaGrad update is equivariant with respect to the transformation. $\qquad\square$

**Proof for RMSProp (Tieleman et al., 2012)**   RMSProp has hyperparameters for learning rate $\alpha$, weight decay $\lambda$, momentum $\beta$, and a small constant $\epsilon$ for stability, and an additional mode if the square averages are centered. For layer $\ell$ at time $t$, we refer to the moving average of squared gradients as $v_t^\ell$, the "average" gradient as $g_t^{\text{ave}(\ell)}$ (which is required if the square averages are centered), and the buffer $b_t^{(\ell)}$, which are all defined below. The update steps for RMSProp are given by:

$$\Theta_t^{(\ell)} = \Theta_{t-1}^{(\ell)} - \alpha b_t^\ell \tag{103}$$

$$b_t^{(\ell)} = \mu b_{t-1}^{(\ell)} + \frac{g_t^{(\ell)}}{\sqrt{v_t^\ell} + \epsilon} \tag{104}$$

$$g_t^{(\ell)} = \mathbf{grad}_t^{(\ell)} + \lambda \Theta_{t-1}^{(\ell)} \tag{105}$$

$$v_t^\ell = \beta v_{t-1}^\ell + (1-\beta)(g_t^{(\ell)} \odot g_t^{(\ell)}) \quad \text{(if not centered)} \tag{106}$$

$$v_t^\ell = \beta v_{t-1}^\ell + (1-\beta)(g_t^{(\ell)} \odot g_t^{(\ell)}) - g_t^{\text{ave}(\ell)} \odot g_t^{\text{ave}(\ell)} \quad \text{(if centered)} \tag{107}$$

$$g_t^{\text{ave}(\ell)} = \beta g_{t-1}^{\text{ave}(\ell)} + (1-\beta)g_t^{(\ell)} \quad \text{(if centered)} \tag{108}$$

Where $g_0^{\text{ave}(\ell)} = 0, v_0^\ell = 0, b_0^{(\ell)} = 0$. Note that in the absense of momentum ($\mu = 0$), the buffer $b_t^{(\ell)}$ is not required, and the update step will simplify to $\Theta_t^{(\ell)} = \Theta_{t-1}^{(\ell)} - \alpha\frac{g_t^{(\ell)}}{\sqrt{v_t^\ell} + \epsilon}$.

Consider the transformation $\mathbf{\Theta}_{<t}^{(\ell)} \mapsto \boldsymbol{\pi}_1^\top(\mathbf{\Theta}_{<t}^{(\ell)})\boldsymbol{\pi}_2$ and $\mathbf{grad}_{<t}^{(\ell)} \mapsto \boldsymbol{\pi}_1^\top(\mathbf{grad}_{<t}^{(\ell)})\boldsymbol{\pi}_2$. We show that the other variables are equivariant by induction:

$$\boldsymbol{g}_t^{(\ell)} \mapsto \boldsymbol{\pi}_1^\top(\boldsymbol{g}_t^{(\ell)})\boldsymbol{\pi}_2 \tag{109}$$

$$\boldsymbol{v}_0^\ell = \mathbf{0} \mapsto \boldsymbol{\pi}_1^\top \mathbf{0} \boldsymbol{\pi}_2 = \mathbf{0} = \boldsymbol{\pi}_1^\top \boldsymbol{v}_0^\ell \boldsymbol{\pi}_2 \tag{110}$$

$$\boldsymbol{v}_t^\ell = \beta \boldsymbol{v}_{t-1}^\ell + (1-\beta)(\boldsymbol{g}_t^{(\ell)} \odot \boldsymbol{g}_t^{(\ell)}) \tag{111}$$

$$\mapsto \beta \boldsymbol{\pi}_1^\top \boldsymbol{v}_{t-1}^\ell \boldsymbol{\pi}_2 + (1-\beta)(\boldsymbol{\pi}_1^\top \boldsymbol{g}_t^{(\ell)} \boldsymbol{\pi}_2 \odot \boldsymbol{\pi}_1^\top \boldsymbol{g}_t^{(\ell)} \boldsymbol{\pi}_2) \tag{112}$$

$$= \beta \boldsymbol{\pi}_1^\top \boldsymbol{v}_{t-1}^\ell \boldsymbol{\pi}_2 + (1-\beta)\boldsymbol{\pi}_1^\top(\boldsymbol{g}_t^{(\ell)} \odot \boldsymbol{g}_t^{(\ell)})\boldsymbol{\pi}_2 \tag{113}$$

$$= \boldsymbol{\pi}_1^\top(\beta \boldsymbol{v}_{t-1}^\ell + (1-\beta)(\boldsymbol{g}_t^{(\ell)} \odot \boldsymbol{g}_t^{(\ell)}))\boldsymbol{\pi}_2 = \boldsymbol{\pi}_1^\top \boldsymbol{v}_t^\ell \boldsymbol{\pi}_2 \tag{114}$$

$$\boldsymbol{g}_0^{\mathrm{ave}(\ell)} = \mathbf{0} \mapsto \boldsymbol{\pi}_1^\top \mathbf{0} \boldsymbol{\pi}_2 = \mathbf{0} = \boldsymbol{\pi}_1^\top \boldsymbol{g}_0^{\mathrm{ave}(\ell)} \boldsymbol{\pi}_2 \tag{115}$$

$$\boldsymbol{g}_t^{\mathrm{ave}(\ell)} = \beta \boldsymbol{g}_{t-1}^{\mathrm{ave}(\ell)} + (1-\beta)\boldsymbol{g}_t^{(\ell)} \tag{116}$$

$$\mapsto \beta \boldsymbol{\pi}_1^\top \boldsymbol{g}_{t-1}^{\mathrm{ave}(\ell)} \boldsymbol{\pi}_2 + (1-\beta)\boldsymbol{\pi}_1^\top \boldsymbol{g}_t^{(\ell)} \boldsymbol{\pi}_2 \tag{117}$$

$$= \boldsymbol{\pi}_1^\top(\beta \boldsymbol{g}_{t-1}^{\mathrm{ave}(\ell)} + (1-\beta)\boldsymbol{g}_t^{(\ell)})\boldsymbol{\pi}_2 = \boldsymbol{\pi}_1^\top \boldsymbol{g}_t^{\mathrm{ave}(\ell)} \boldsymbol{\pi}_2 \tag{118}$$

$$\boldsymbol{v}_t^\ell = \beta \boldsymbol{v}_{t-1}^\ell + (1-\beta)(\boldsymbol{g}_t^{(\ell)} \odot \boldsymbol{g}_t^{(\ell)}) - \boldsymbol{g}_t^{\mathrm{ave}(\ell)} \odot \boldsymbol{g}_t^{\mathrm{ave}(\ell)} \quad \text{(if centered)} \tag{119}$$

$$\mapsto \beta \boldsymbol{\pi}_1^\top \boldsymbol{v}_{t-1}^\ell \boldsymbol{\pi}_2 + (1-\beta)(\boldsymbol{\pi}_1^\top \boldsymbol{g}_t^{(\ell)} \boldsymbol{\pi}_2 \odot \boldsymbol{\pi}_1^\top \boldsymbol{g}_t^{(\ell)} \boldsymbol{\pi}_2) - \boldsymbol{\pi}_1^\top \boldsymbol{g}_t^{\mathrm{ave}(\ell)} \boldsymbol{\pi}_2 \odot \boldsymbol{\pi}_1^\top \boldsymbol{g}_t^{\mathrm{ave}(\ell)} \boldsymbol{\pi}_2 \tag{120}$$

$$= \beta \boldsymbol{\pi}_1^\top \boldsymbol{v}_{t-1}^\ell \boldsymbol{\pi}_2 + (1-\beta)\boldsymbol{\pi}_1^\top(\boldsymbol{g}_t^{(\ell)} \odot \boldsymbol{g}_t^{(\ell)})\boldsymbol{\pi}_2 - \boldsymbol{\pi}_1^\top(\boldsymbol{g}_t^{\mathrm{ave}(\ell)} \odot \boldsymbol{g}_t^{\mathrm{ave}(\ell)})\boldsymbol{\pi}_2 \tag{121}$$

$$= \boldsymbol{\pi}_1^\top(\beta \boldsymbol{v}_{t-1}^\ell + (1-\beta)(\boldsymbol{g}_t^{(\ell)} \odot \boldsymbol{g}_t^{(\ell)}) - \boldsymbol{g}_t^{\mathrm{ave}(\ell)} \odot \boldsymbol{g}_t^{\mathrm{ave}(\ell)})\boldsymbol{\pi}_2 = \boldsymbol{\pi}_1^\top \boldsymbol{v}_t^\ell \boldsymbol{\pi}_2 \tag{122}$$

$$\boldsymbol{b}_0^{(\ell)} = \mathbf{0} \mapsto \boldsymbol{\pi}_1^\top \mathbf{0} \boldsymbol{\pi}_2 = \mathbf{0} = \boldsymbol{\pi}_1^\top \boldsymbol{b}_0^{(\ell)} \boldsymbol{\pi}_2 \tag{123}$$

$$\boldsymbol{b}_t^{(\ell)} = \mu \boldsymbol{b}_{t-1}^{(\ell)} + \frac{\boldsymbol{g}_t^{(\ell)}}{\sqrt{\boldsymbol{v}_t^\ell} + \epsilon} \tag{124}$$

$$\mapsto \mu \boldsymbol{\pi}_1^\top \boldsymbol{b}_{t-1}^{(\ell)} \boldsymbol{\pi}_2 + \frac{\boldsymbol{\pi}_1^\top \boldsymbol{g}_t^{(\ell)} \boldsymbol{\pi}_2}{\sqrt{\boldsymbol{\pi}_1^\top \boldsymbol{v}_t^\ell \boldsymbol{\pi}_2} + \epsilon} \tag{125}$$

$$= \mu \boldsymbol{\pi}_1^\top \boldsymbol{b}_{t-1}^{(\ell)} \boldsymbol{\pi}_2 + \boldsymbol{\pi}_1^\top \left( \frac{\boldsymbol{g}_t^{(\ell)}}{\sqrt{\boldsymbol{v}_t^\ell} + \epsilon} \right) \boldsymbol{\pi}_2 \tag{126}$$

$$= \boldsymbol{\pi}_1^\top \left( \mu \boldsymbol{b}_{t-1}^{(\ell)} + \frac{\boldsymbol{g}_t^{(\ell)}}{\sqrt{\boldsymbol{v}_t^\ell} + \epsilon} \right) \boldsymbol{\pi}_2 = \boldsymbol{\pi}_1^\top \boldsymbol{b}_t^{(\ell)} \boldsymbol{\pi}_2 \tag{127}$$

Finally, for the weight update:

$$\mathbf{\Theta}_t^{(\ell)} = \mathbf{\Theta}_{t-1}^{(\ell)} - \alpha \frac{\boldsymbol{g}_t^{(\ell)}}{\sqrt{\boldsymbol{v}_t^\ell} + \epsilon} \tag{128}$$

$$\mapsto \boldsymbol{\pi}_1^\top \mathbf{\Theta}_{t-1}^{(\ell)} \boldsymbol{\pi}_2 - \alpha \frac{\boldsymbol{\pi}_1^\top \boldsymbol{g}_t^{(\ell)} \boldsymbol{\pi}_2}{\sqrt{\boldsymbol{\pi}_1^\top \boldsymbol{v}_t^\ell \boldsymbol{\pi}_2} + \epsilon} \tag{129}$$

$$= \boldsymbol{\pi}_1^\top \mathbf{\Theta}_{t-1}^{(\ell)} \boldsymbol{\pi}_2 - \boldsymbol{\pi}_1^\top \left( \alpha \frac{\boldsymbol{g}_t^{(\ell)}}{\sqrt{\boldsymbol{v}_t^\ell} + \epsilon} \right) \boldsymbol{\pi}_2 \tag{130}$$

$$= \boldsymbol{\pi}_1^\top \left( \mathbf{\Theta}_{t-1}^{(\ell)} - \alpha \frac{\boldsymbol{g}_t^{(\ell)}}{\sqrt{\boldsymbol{v}_t^\ell} + \epsilon} \right) \boldsymbol{\pi}_2 = \boldsymbol{\pi}_1^\top \mathbf{\Theta}_t^{(\ell)} \boldsymbol{\pi}_2 \tag{131}$$

Thus, the RMSProp update is equivariant with respect to the transformation. $\qquad\square$

**Proof for a general case** We can show that a general optimizer leads to equivariance under the transformation if the update step can be separated for each scalar entry of the parameters.

**Lemma 11** (Update Equivariance of a separable optimizer). *Let the parameters be updated by the function $f$, such that for any step $t$,*

$$\theta_{t+1} = f(\{\theta_{\text{iter}} : \text{iter} \leq t\}, \{g_{\text{iter}} : \text{iter} \leq t\}, \eta_t, Z_t) \tag{132}$$

*where, $g_{\text{iter}}$ based on the optimizer may be the gradient (which may also be clipped and/or normalized gradient) w.r.t. the parameters $\theta_{\text{iter}}$ which is equivariant under $\Gamma_{\boldsymbol{\pi}}$.*

*Let $\eta_t$ be the set of hyperparameters of the optimizer (this may include learning rate, momentum, etc.) at update step $t$, and $Z_t$ be a latent random variable representing any stochasticity in the update step (such as data selection for SGD/mini-batch).*

*We call $f$ to be separable over each scalar, if we can write for any parameter $\boldsymbol{\Theta}^{(\ell)}$, for all of its entries entries $i, j$,*

$$\boldsymbol{\Theta}_{t+1}^{(\ell)}[i,j] = f^{(\ell)}(\{\boldsymbol{\Theta}_{\text{iter}}^{(\ell)}[i,j] : \text{iter} \leq t\}, \{\boldsymbol{g}_{\text{iter}}^{(\ell)}[i,j] : \text{iter} \leq t\}, \eta_t, Z_t) \tag{133}$$

*where $f^{(\ell)}$ is an appropriate function which may be different for each layer $\ell \in d$.*

*Then, the update step is equivariant (conditioned on $(Z_i : i \leq t)$) to any transformation $\Gamma_{\boldsymbol{\pi}}$ applied jointly to each of $\{\theta_{\text{iter}}, g_{\text{iter}}\}$ for $\text{iter} \leq t$.*

Note that this functional form is quite general despite the separability condition, as it subsumes commonly used optimizers - GD,SGD, Momentum, RMSProp, Adam, AdamW, Adagrad, etc. The conditioning on the latent random variables implies that the equivariance also holds in expectation over the randomness.

*Proof*:

The proof follows from the fact that the transformation $\Gamma_{\boldsymbol{\pi}}$ is composed of permutations in each of the weights. Consider a layer $\ell$ with parameters $\theta^{(\ell)}$, of size $d_1 \times d_2$. We may find a permutation $\hat{\pi} : [d_1] \times [d_2] \mapsto [d_1] \times [d_2]$ such that for any entry $(i,j)$ of a matrix $\boldsymbol{A}$, $\Gamma_{\boldsymbol{\pi}}^{(\ell)}(\boldsymbol{A})[i,j] = \boldsymbol{A}[\hat{\pi}(i,j)]$. To reiterate, under the transformation $\Gamma_{\boldsymbol{\pi}}$, $\forall t \forall (i,j) \in [d_1] \times [d_2]$, $\boldsymbol{\Theta}^{(\ell)}[i,j] \mapsto \boldsymbol{\Theta}^{(\ell)}[\hat{\pi}(i,j)]$ and $\boldsymbol{g}^{(\ell)}[i,j] \mapsto \boldsymbol{g}^{(\ell)}[\hat{\pi}(i,j)]$.

Then, for any step $t$, under the action of $\Gamma_{\boldsymbol{\pi}}$ on $\{\theta_{\text{iter}}, g_{\text{iter}}\}$ for $\text{iter} \leq t$,

$$f^{(\ell)}(\{\boldsymbol{\Theta}_{\text{iter}}^{(\ell)}[i,j] : \text{iter} \leq t\}, \{\boldsymbol{g}_{\text{iter}}^{(\ell)}[i,j] : \text{iter} \leq t\}, \eta_t, Z_t)$$
$$\mapsto f^{(\ell)}(\{\boldsymbol{\Theta}_{\text{iter}}^{(\ell)}[\hat{\pi}(i,j)] : \text{iter} \leq t\}, \{\boldsymbol{g}_{\text{iter}}^{(\ell)}[\hat{\pi}(i,j)] : \text{iter} \leq t\}, \eta_t, Z_t) \tag{134}$$

$$= \boldsymbol{\Theta}_{t+1}^{(\ell)}[\hat{\pi}(i,j)] = \Gamma_{\boldsymbol{\pi}}^{(\ell)}(\boldsymbol{\Theta}_{t+1}^{(\ell)})[i,j] \tag{135}$$

Thus $\boldsymbol{\Theta}_{t+1}^{(\ell)}[i,j] \mapsto \Gamma_{\boldsymbol{\pi}}^{(\ell)}(\boldsymbol{\Theta}_{t+1}^{(\ell)})[i,j]$. Since this holds for all entries $(i,j)$, we have $\boldsymbol{\Theta}_{t+1}^{(\ell)} \mapsto \Gamma_{\boldsymbol{\pi}}^{(\ell)}(\boldsymbol{\Theta}_{t+1}^{(\ell)})$. Finally, since this holds for all layers $\ell$, we have $\theta_{t+1} \mapsto \Gamma_{\boldsymbol{\pi}}(\theta_{t+1})$.

∎

### E.1.6 ADDITIONAL RESULTS ON EXCHANGEABILITY

**Loss functions without permutation equivariance** In this paper, we take the loss to be a direct function of the embeddings, which necessitates that the loss function be permutation invariant.

When we consider settings where the loss is not permutation invariant, for example a classification task, the 'representations' exist within the middle of the network rather than at the end. Moreover, such representations can be shown to be exchangeable.

For this analysis, we may partition the network into two, which could be referred to as the 'embedding' network and the 'classifier head'. e may write $\mathbf{X} = \text{NN}(G)$ where we refer to $\mathbf{X}$ as the embeddings and $\hat{\mathbf{y}} = \text{Clf}(\mathbf{X})$ where $\hat{\mathbf{y}}$ is the prediction label vector across nodes. We can characterize and prove the exchangeability of $\mathbf{X}$ for this setting.

Let the parameters of the entire network at $t$ timesteps be represented by $\theta = (\theta_{\text{NN}}, \theta_{\text{Clf}})$, coresponding to the parameters of either network. Let us also define the permutation inducing transformation as $\Gamma_{\pi} = \Gamma_{\text{NN},\pi} \otimes \Gamma_{\text{Clf},\pi}$, i.e. $\Gamma_{\pi}(\theta) = (\Gamma_{\text{NN},\pi}(\theta_{\text{NN}}), \Gamma_{\text{Clf},\pi}(\theta_{\text{Clf}}))$.

Given the dataset, we may reparameterise the loss function as $\mathcal{L}(\mathbf{X}, \text{Clf})$, or equivalently, $\mathcal{L}(\mathbf{X}, \theta_{\text{Clf}})$.

The new condition for the transformation boils down to

- $\mathbf{X} \mapsto \mathbf{X}\pi$ under $\Gamma_{\mathrm{NN},\pi}$
- the loss is invariant under $(\pi, \Gamma_{\mathrm{Clf},\pi})$, i.e.

$$\mathcal{L}(\mathbf{X}, \theta_{\mathrm{Clf}}) = \mathcal{L}(\mathbf{X}\pi, \Gamma_{\mathrm{Clf},\pi}(\theta_{\mathrm{Clf}})) \tag{136}$$

Under these conditions, exchangeability follows with the same steps - exchangeability at initialisation, equivariance of gradient, equivariance of update step.

To illustrate this, consider a three class classification task with a single layer for both NN and Clf. Let the input feature be **feat**. Let us focus on one channel/node of $\mathbf{X}$ denoted as $\mathbf{x} = \mathbf{X}[:, \bullet]$ and $\hat{\mathbf{y}}[\bullet] = y$. We have: $\mathbf{x} = \mathrm{NN}(\mathbf{feat}) = \sigma(\mathbf{feat}\Theta_{\mathrm{NN}})$. Hence, we will have: $\hat{y} = \mathrm{Softmax}([(\mathbf{x} \cdot \mathbf{w}_1), (\mathbf{x} \cdot \mathbf{w}_2), (\mathbf{x} \cdot \mathbf{w}_3)])$.

The transformation $\Gamma_{\mathrm{NN},\pi}$ can then represented as, $\Theta_{\mathrm{NN}} \mapsto \Theta_{\mathrm{NN}}\pi$ and $[\mathbf{w}_1, \mathbf{w}_2, \mathbf{w}_3] \mapsto [\pi^\top \mathbf{w}_1, \pi^\top \mathbf{w}_2, \pi^\top \mathbf{w}_3]$. Under this transformation $\mathbf{x} \mapsto \mathbf{x}\pi$ but $\hat{y}$ remains invariant—therefore, the loss is invariant.

**Effect of normalization**  Batch norm, layer norm, etc. do not break exchangeability condition. If the network without the norm layers can be shown to give exchangeable embeddings, the same will hold for the embeddings for the network with batch norm or layer norm.

We denote a normalization layer as $NL_{\gamma,\beta}$, where $\gamma$ and $\beta$ are parameters. Such layers allow us to extend permutation inducing transformation $\gamma_\pi$ to $\gamma'_\pi$. For simplicity, assume that the normalization layer $NL_{\gamma,\beta}$ is applied on one layer $\ell$. Suppose, $\theta \to \gamma_\pi(\theta)$ gives $\mathbf{Z} \to \mathbf{Z}\pi$ in that $\ell$ layer (where $\mathbf{Z} \in \mathbb{R}^{n \times dim_z}$). Then we can obtain a transformation $\gamma'_\pi$ such that $\theta \cup \{\gamma, \beta\} \to \gamma'_\pi(\theta \cup \{\gamma, \beta\})$ will also give $\mathbf{Z} \to \mathbf{Z}\pi$.

Let the batch of inputs be $G_1, G_2, \cdots, G_B$ and a single batch norm layer, with the corresponding inputs $\mathbf{Y}_1, \mathbf{Y}_2, \cdots, \mathbf{Y}_B$ to the layer. Then, we have: $\mathbf{Z}_1, \mathbf{Z}_2, \cdots, \mathbf{Z}_B = \mathrm{BatchNorm}(\mathbf{Y}_1, \mathbf{Y}_2, \cdots, \mathbf{Y}_B; \gamma, \beta)$. Suppose: $\widehat{\mathbf{Y}} = \frac{[\mathbf{Y}_1, \mathbf{Y}_2, \cdots, \mathbf{Y}_B] - \overline{\mathbf{Y}}}{\sqrt{\mathrm{Var}(\mathbf{Y}_1, \mathbf{Y}_2, \cdots, \mathbf{Y}_B) + \epsilon}}$ where $\overline{\mathbf{Y}}$ is the batch mean. Then, we have: $\mathbf{Z}_1, \mathbf{Z}_2, \cdots, \mathbf{Z}_B = \widehat{\mathbf{Y}} \odot \gamma + \beta$. Now, suppose $\theta \to \gamma_\pi(\theta)$ gives $\mathbf{Y} \to \mathbf{Y}\pi$. This would give $\widehat{\mathbf{Y}} \to \widehat{\mathbf{Y}}\pi$. Suppose, we now transform $\gamma \to \gamma\pi$ and $\beta \to \beta\pi$. Then, $\mathbf{Z}_1, \mathbf{Z}_2, \cdots, \mathbf{Z}_B \to \widehat{\mathbf{Y}}\pi \odot (\gamma\pi) + \beta\pi = (\widehat{\mathbf{Y}} \odot \gamma + \beta)\pi = \mathbf{Z}_1\pi, \mathbf{Z}_2\pi, \cdots, \mathbf{Z}_B\pi$.

Consider layer norm. Assume the corresponding input is $\mathbf{y}$ and output in one channel is $\mathbf{z} = \mathrm{LayerNorm}(\mathbf{y}; \gamma, \beta)$. Suppose: $\widehat{\mathbf{y}} = \frac{\mathbf{y} - y\mathbf{1}}{\sqrt{\mathrm{Var}(\mathbf{y}) + \epsilon}}$ where $y$ is the feature mean. Then, we have: $\mathbf{z} = \widehat{\mathbf{y}} \odot \gamma + \beta$. Now, suppose $\theta \to \gamma_\pi(\theta)$ gives $\mathbf{y} \to \mathbf{y}\pi$. This would give $\widehat{\mathbf{y}} \to \widehat{\mathbf{y}}\pi$. Suppose, we now transform $\gamma \to \gamma\pi$ and $\beta \to \beta\pi$. Then, $\mathbf{z} \to \widehat{\mathbf{y}} \odot (\gamma\pi) + \beta\pi = \mathbf{z}\pi$.

Hence, $\gamma'_\pi(\theta \cup \{\gamma, \beta\}) = (\gamma_\pi(\theta), \gamma\pi, \beta\pi)$. Therefore, Lemma 2 holds true even when we apply Batch norm or Layer norm on each layer/feature. Since Lemma 2 is used to prove Lemma 3, 4 and these lemmas are used to prove the final result in Theorem 5, our results of exchangeability remain the same, regardless of normalization layer.

### E.2 PROOFS OF THE TECHNICAL RESULTS IN SECTION 4

Here, we first prove Proposition 7, and then derive the equivalence of Eqs. (3) and (4).

### E.2.1 PROOF OF PROPOSITION 7

**Proposition 7.** *For any $\epsilon > 0, \delta > 0$, setting $D > \frac{1}{\epsilon^2 \delta}$ ensures that, for some $\beta_0 = O_D(1)$, we have:*

$$\Pr \left( \left| \frac{1}{D} \text{sim}(G_c, G_q) - \text{sim}_d(G_c, G_q) \right| \le \epsilon \right) \ge 1 - \beta_0 \delta \tag{137}$$

**Proof:** For the purposes of the proof, we introduce a new similarity measure $\overline{\text{sim}}(G_c, G_q)$,

$$\overline{\text{sim}}(G_c, G_q) = \max_{\boldsymbol{P} \in \mathcal{P}_n} \sum_{u, u' \in [n] \times [n]} \mathbb{E}[s(\boldsymbol{x}^{(q)}(u)[d] - \boldsymbol{x}^{(c)}(u')[d])] \boldsymbol{P}[u, u']. \tag{138}$$

We use the above to prove two results:

$$\Pr \left( \left| \frac{1}{D} \text{sim}(G_c, G_q) - \overline{\text{sim}}(G_c, G_q) \right| \le \epsilon \right) \ge 1 - \beta \delta \tag{139}$$

$$\Pr \left( \left| \text{sim}_d(G_c, G_q) - \overline{\text{sim}}(G_c, G_q) \right| \le \epsilon \right) \ge 1 - \beta \delta \tag{140}$$

where $\beta = O_D(1)$. Finally, we will use the union bound to get the desired result. In addition to $\overline{\text{sim}}(G_c, G_q)$, we also introduce additional notation to facilitate the proofs:

**(1)** $\boldsymbol{Z}$ is a matrix indexed by the pair of nodes, and the embedding dimension. In particular,

$$\boldsymbol{Z}[(u, u'), d] \triangleq s(\boldsymbol{x}^{(q)}(u)[d] - \boldsymbol{x}^{(c)}(u')[d]) \tag{141}$$

**(2)** We define the vector $\boldsymbol{Z}_d$ by fixing the value at dimension $d$.

$$\boldsymbol{Z} \triangleq [\boldsymbol{Z}[(u, u'), d]]_{(u,u'),d} \tag{142}$$

$$\boldsymbol{Z}_d \triangleq [\boldsymbol{Z}[(u, u'), d]]_{(u,u')} \tag{143}$$

**(3)** $\overline{\boldsymbol{Z}}$ is the expectation value of $\boldsymbol{Z}_d$ with respect to the initialization of the embedding model. As it follows from the exchangeability of the dimensions in Theorem 15, we have: $\mathbb{E}[\boldsymbol{Z}_1] = \mathbb{E}[\boldsymbol{Z}_2] = \cdots = \mathbb{E}[\boldsymbol{Z}_D]$.

$$\overline{\boldsymbol{Z}} = \mathbb{E}[\boldsymbol{Z}_d] \tag{144}$$

**(4)** Our estimator is denoted by the vector $\widehat{\boldsymbol{Z}}$.

$$\widehat{\boldsymbol{Z}} \triangleq \frac{1}{D} \sum_{i=1}^{D} \boldsymbol{Z}_d \tag{145}$$

Thus, our similarity can be written as

$$\frac{1}{D} \text{sim}(G_c, G_q) = \max_{P \in \mathcal{P}_n} \sum_{u, u' \in [n] \times [n]} \widehat{\boldsymbol{Z}}[(u, u')] \boldsymbol{P}[u, u'] \tag{146}$$

Suppose $\boldsymbol{R}$ is any matrix in $\mathbb{R}^{n \times n}$. Then, we define the following quantities:

$$\Lambda(\boldsymbol{R}, \boldsymbol{P}) \triangleq \sum_{u, u' \in [n] \times [n]} \boldsymbol{R}[(u, u')] \boldsymbol{P}[u, u'] \tag{147}$$

$$\boldsymbol{P}^*(\boldsymbol{R}) \triangleq \arg\max_{\boldsymbol{P} \in \mathcal{P}_n} \Lambda(\boldsymbol{R}, \boldsymbol{P}) \tag{148}$$

$$\Lambda^*(\boldsymbol{R}) \triangleq \max_{\boldsymbol{P} \in \mathcal{P}_n} \Lambda(\boldsymbol{R}, \boldsymbol{P}) = \Lambda(\boldsymbol{R}, \boldsymbol{P}^*(\boldsymbol{Z})) \tag{149}$$

Thus we have: $\frac{1}{D} \text{sim}(G_c, G_q) = \Lambda^*(\widehat{\boldsymbol{Z}})$ and $\text{sim}_d(G_c, G_q) = \Lambda^*(\boldsymbol{Z}_d)$. Therefore, we first establish that if $D > \frac{1}{\epsilon^2 \delta}$, then

$$\Pr \left( \left| \Lambda^*(\widehat{\boldsymbol{Z}}) - \Lambda^*(\overline{\boldsymbol{Z}}) \right| \ge \epsilon \right) \le \beta \delta. \tag{150}$$

We begin by showing that $\Lambda^*$ is $\sqrt{n}$-Lipschitz. Convexity of $\Lambda^*$ follows from the convexity of $\Lambda(\cdot, \boldsymbol{P})$ and Danskin's Theorem (Theorem 13). By Danskin's theorem, the semi-derivative of $\Lambda^*$ with respect to $\boldsymbol{R}$ in the direction of $\boldsymbol{v}$ is given by

$$\partial_{\boldsymbol{v}} \Lambda^*(\boldsymbol{R}) = \max_{\boldsymbol{P}: \boldsymbol{P} = \boldsymbol{P}^*(\boldsymbol{R})} \Lambda(\boldsymbol{R}, \boldsymbol{P}) \le \Lambda^*(\boldsymbol{v}) \tag{151}$$

From Eq. (147), we have: $|\partial_{\boldsymbol{R}} \Lambda^*(\boldsymbol{R})| \leq \|\text{vec}(\boldsymbol{P})\|_2 = \sqrt{n}$. This gives us:

$$|\Lambda^*(\boldsymbol{R}_1) - \Lambda^*(\boldsymbol{R}_2)| \leq \sqrt{n} \|\boldsymbol{R}_1 - \boldsymbol{R}_2\|_2 \tag{152}$$

Note that this is the vector $2-$norm, not the matrix $2-$norm. This proves that $\Lambda^*$ is Lipschitz, from which it follows that for any $\epsilon$, $|\Lambda^*(\boldsymbol{R}_1) - \Lambda^*(\boldsymbol{R}_2)| \geq \epsilon \implies \sqrt{n} \|\boldsymbol{R}_1 - \boldsymbol{R}_2\|_2 \geq \epsilon$. This gives us: We now use this fact in proving Eq. (139).

$$\Pr\left(\left|\Lambda^*(\widehat{\boldsymbol{Z}}) - \Lambda^*(\overline{\boldsymbol{Z}})\right| \geq \epsilon\right) \leq \Pr\left(\|\widehat{\boldsymbol{Z}} - \overline{\boldsymbol{Z}}\|_2 \geq \frac{1}{\sqrt{n}}\epsilon\right) \qquad \text{(Eq. (152))} \tag{153}$$

$$\leq \frac{\sum_{u,u' \in [n] \times [n]} \text{Var}(\widehat{\boldsymbol{Z}}[(u, u')])}{\left(\frac{\epsilon}{\sqrt{n}}\right)^2} \quad \text{(Chebyshev's Inequality)}$$

$$= \frac{n}{D^2 \epsilon^2} \sum_{u,u' \in [n] \times [n]} \text{Var}\left(\sum_{d \in [D]} \boldsymbol{Z}_d[(u, u')]\right) \tag{154}$$

$$= \frac{\beta}{D\epsilon^2} \tag{155}$$

Here, $\beta$ is computed using the variance bound computed by Lemma 16: $\beta = n \cdot 4L_s^2 B^2 \cdot n^2$. To prove Eq. (140), we directly invoke the Lipschitz condition for $\Lambda^*$ from Eq. (152).

$$\Pr\left(\left|\Lambda^*(\boldsymbol{Z}_i) - \Lambda^*(\overline{\boldsymbol{Z}})\right| \geq \epsilon\right) \leq \Pr\left(\|\boldsymbol{Z}_i - \overline{\boldsymbol{Z}}\|_2 \geq \frac{\epsilon}{\sqrt{n}}\right) \qquad \text{(Eq. (152))} \tag{156}$$

$$\leq \frac{\sum_{u,u' \in [n] \times [n]} \text{Var}(\boldsymbol{Z}_i[u, u'])}{\left(\frac{\epsilon}{\sqrt{n}}\right)^2} \quad \text{(Chebyshev's Inequality)}$$

$$\leq \sum_{u,u' \in [n] \times [n]} \frac{n}{\epsilon^2} \cdot \frac{4L_s^2 B^2}{D} \quad \text{(From variance bound, Lemma 17)}$$

$$\tag{157}$$

$$= \frac{\beta}{D\epsilon^2}, \quad \text{where } \beta = n \cdot 4L_s^2 B^2 \cdot n^2. \tag{158}$$

Using the results in Eqs. (139) and (140), we now prove the main result (5), using the union bound

$$\Pr(|\text{sim}(G_c, G_q) - \text{sim}_d(G_c, G_q)| \geq \epsilon)$$

$$\leq \Pr(|\text{sim}(G_c, G_q) - \overline{\text{sim}}(G_c, G_q)| \geq \frac{\epsilon}{2})$$

$$+ \Pr(|\text{sim}_d(G_c, G_q) - \overline{\text{sim}}(G_c, G_q)| \geq \frac{\epsilon}{2}) \tag{159}$$

$$\leq \frac{4\beta}{D\epsilon^2} + \frac{4\beta}{D\epsilon^2} = \frac{8\beta}{D\epsilon^2}$$

$$=: \frac{\beta_0}{D\epsilon^2} \tag{160}$$

$$\blacksquare$$

### E.2.2 PROOF OF THE FACT THAT EQ. (3) AND EQ. (4) ARE EQUIVALENT

Here, we will show that if we have:

$$\text{sim}_d(G_c, G_q) = \max_{\boldsymbol{P} \in \mathcal{P}_n} \sum_{u,u'} s(\boldsymbol{x}^{(q)}(u)[d] - \boldsymbol{x}^{(c)}(u')[d])\boldsymbol{P}[u, u'], \tag{161}$$

then $\text{sim}_d$ can also be written as:

$$\text{sim}_d(G_c, G_q) = s\big(\text{SORT}(\boldsymbol{X}^{(q)}[:, d]) - \text{SORT}(\boldsymbol{X}^{(c)}[:, d])\big) \tag{162}$$

In the following, we provide this result, in terms of any two vectors $\boldsymbol{x}$ and $\boldsymbol{y}$.

**Theorem 12** (Rearrangement for $s$). *Given a convex function $\rho : \mathbb{R}^D \to [0, \infty)$, which is not necessarily symmetric and satisfies $\rho(\boldsymbol{x}) = \sum_i \rho(\boldsymbol{x}[i])$, and a score function $s$ that is of the form $s(\cdot) = \rho_{\max} - \rho(\cdot)$[1], for all $\boldsymbol{x}, \boldsymbol{y}$ with $\|\boldsymbol{x}\|_\infty, \|\boldsymbol{y}\|_\infty \leq x_{\max}$, we have:*

$$\max_{\boldsymbol{P} \in \mathcal{P}_n} \sum_{u,u'} s\left(\boldsymbol{x}[u] - \boldsymbol{y}[u']\right) \boldsymbol{P}[u, u'] = s\left(\text{SORT}(\boldsymbol{x}) - \text{SORT}(\boldsymbol{y})\right) \tag{163}$$

**Proof** This is a well known result for $L_p$ metric. For optimal transport between distributions, such result exists for convex distances (Santambrogio, 2015, Proposition 2.17). We still provide the proof for self containment. Here, we will apply Lemma 14. But that requires some conditions on $s(\bullet - \bullet)$ (stated as $\mu(\bullet, \bullet)$ therein). We will prove that as long as $\rho$ is convex, $s$ satisfies those conditions required to apply Lemma 14.

Those conditions requires us to show the following: For $a_1, a_2, b_1, b_2 \in \mathbb{R}$ with $a_1 \geq a_2, b_1 \geq b_2$,

$$\rho\left(a_1 - b_2\right) + \rho\left(a_2 - b_1\right) \geq \rho\left(a_1 - b_1\right) + \rho\left(a_2 - b_2\right) \tag{164}$$

To show this, we invoke the convexity of $\rho\left(\cdot\right)$. For any $x, y, z \in \mathbb{R}$ with $x \geq y$ and $z \geq 0$, consider the case $x \geq y$, then $x + z \geq x \geq y$, $x + z \geq y + z \geq y$. Convexity of $\rho$ gives us:

$$\frac{(x - y)\rho(x + z) + z\rho(y)}{x + z - y} \geq \rho(x) \tag{165}$$

$$\frac{z\rho(x + z) + (x - y)\rho(y)}{x + z - y} \geq \rho(y + z) \tag{166}$$

Summing both inequalities, we have: $\rho(x + z) + \rho(y) \geq \rho(x) + \rho(y + z)$. W.l.o.g. consider $a_1, a_2, b_1, b_2 \in \mathbb{R}$ with $a_1 \geq a_2, b_1 \geq b_2$, of the following form:

$$a_1 = b_1 + x$$
$$a_2 = b_1 + y$$
$$b_2 = b_1 - z$$

This gives us Eq. (164).

To finish proving the theorem, we notice that: due to $\max_{\boldsymbol{P} \in \mathcal{P}_n} \sum_{u,u'} s\left(\boldsymbol{x}[u] - \boldsymbol{y}[u']\right) \boldsymbol{P}[u, u'] = \max_{\boldsymbol{P} \in \mathcal{P}_n} \sum_{u,u'} s\left((\boldsymbol{P'x})[u] - \boldsymbol{y}[u']\right) \boldsymbol{P}[u, u']$ for any permutation $\boldsymbol{P'}$, we have:

$$\max_{\boldsymbol{P} \in \mathcal{P}_n} \sum_{u,u'} s\left(\boldsymbol{x}[u] - \boldsymbol{y}[u']\right) \boldsymbol{P}[u, u'] = \max_{\boldsymbol{P} \in \mathcal{P}_n} \sum_{u,u'} s\left(\text{SORT}(\boldsymbol{x})[u] - \boldsymbol{y}[u']\right) \boldsymbol{P}[u, u'] \tag{167}$$

Now, thanks to Eq. (164), $s(\bullet)$ satisfies the conditions in Lemma 14 with $\mu(x, y)$ in that Lemma satisfies $\mu(x, y) = s(x - y)$. This gives us: $\max_{\boldsymbol{P} \in \mathcal{P}_n} \sum_{u,u'} s\left(\boldsymbol{x}[u] - \boldsymbol{y}[u']\right) \boldsymbol{P}[u, u'] = s\left(\text{SORT}(\boldsymbol{x}) - \text{SORT}(\boldsymbol{y})\right)$. ∎

---

[1]as designed before introducing Eq. 2.

### E.2.3 AUXILIARY RESULTS USED TO PROVE LEMMAS IN APPENDIX E.2

**Lemma 13** (Danskin's Theorem (Danskin, 1967)). *Let $g : \mathbb{R}^m \times Z \to \mathbb{R}$ be a continuous function of two arguments where $Z \subset \mathbb{R}^l$ is a compact set. Let $f(x) = \max_{z \in Z} g(x, z)$, then*

- *$f$ is convex if $g(\cdot, z)$ is convex for any $z \in Z$.*
- *$f$ is differentiable at $x$ if the $\arg\max_z$ is a single possible element.*
- *The semi-differential of $f$ in the direction of $\boldsymbol{v}$ is given by*

$$\partial_{\boldsymbol{v}} f(x) = \max_{z \in Z^*} g'(x, z | \boldsymbol{v}) \tag{168}$$

  *where $g'(x, z | \boldsymbol{v})$ is the derivative of $g$ in the direction $\boldsymbol{v}$, and $Z^*$ is the set of maximising points of $g(\cdot, z)$*
- *If $f$ is differentiable at $x$, then the gradient of $f$ is given by $\nabla_{\boldsymbol{x}} f(\boldsymbol{x}) = \nabla_{\boldsymbol{x}} g(\boldsymbol{x}, z^*) = \nabla_1 g(\boldsymbol{x}, z^*)$ (gradient in the first argument).*

**Lemma 14** (Rearrangement Inequality). *(Wu, 2022, Theorem 7) Let $\mu$ be a real-valued function of 2 variables defined on $I_a \times I_b$. If*

$$\mu(x_2, y_2) - \mu(x_2, y_1) - \mu(x_1, y_2) + \mu(x_1, y_1) \geq 0$$

*for all $x_1 \leq x_2$ in $I_a$ and $y_1 \leq y_2$ in $I_b$, then*

$$\sum_{i \in [n]} \mu(a_i, b_{n-i+1}) \leq \sum_{i \in [n]} \mu(a_i, b_{\pi(i)}) \leq \sum_{i \in [n]} \mu(a_i, b_i) \tag{169}$$

*for all sequences $a_1 \leq a_2 \leq \cdots \leq a_n$ in $I_a$, $b_1 \leq b_2 \leq \cdots \leq b_n$ in $I_b$, and all permutations $\pi$ of $[n]$.*

**Theorem 15.** *If the columns of $\boldsymbol{X}$ are distrbuted exchangeably, then for any $d, d' \in [D]$ and $u, v \in [n]$*

$$\mathbb{E}_{\boldsymbol{x}_u[d], \boldsymbol{x}_v[d]} s\left(\boldsymbol{x}_u[d] - \boldsymbol{x}_v[d]\right) = \mathbb{E}_{\boldsymbol{x}_u[d'], \boldsymbol{x}_v[d']} s\left(\boldsymbol{x}_u[d'] - \boldsymbol{x}_v[d']\right) \tag{170}$$

**Proof** As columns of $\boldsymbol{X}$ are distributed exchangeably, the joint distribution of $(\boldsymbol{x}_u, \boldsymbol{x}_v)$ is also exchangeable. Thus the marginals are also the same, $p_{\boldsymbol{x}_u[d], \boldsymbol{x}_v[d]} = p_{\boldsymbol{x}_u[d'], \boldsymbol{x}_v[d']}$. Therefore,

$$\mathbb{E}_{\boldsymbol{x}_u[d], \boldsymbol{x}_v[d]} s\left(\boldsymbol{x}_u[d] - \boldsymbol{x}_v[d]\right) = \int_{\mathbb{R}^2} s\left(x, y\right) p_{\boldsymbol{x}_u[d], \boldsymbol{x}_v[d]}\left(x, y\right) dx \, dy \tag{171}$$

$$= \int_{\mathbb{R}^2} s\left(x, y\right) p_{\boldsymbol{x}_u[d'], \boldsymbol{x}_v[d']}\left(x, y\right) dx \, dy \tag{172}$$

$$= \mathbb{E}_{\boldsymbol{x}_u[d'], \boldsymbol{x}_v[d']} s\left(\boldsymbol{x}_u[d'] - \boldsymbol{x}_v[d']\right). \tag{173}$$

$\blacksquare$

**Lemma 16** (Variance Bound for $\sum_{d \in [D]} \boldsymbol{Z}_d$). *Let $\boldsymbol{Z}_d$ be defined as in Eq. (143). Given that $\|\boldsymbol{x}^{(c)}(u')\|_2, \|\boldsymbol{x}^{(q)}(u)\|_2 \leq B$, then we can bound*

$$\mathrm{Var}\left(\sum_{d \in [D]} \boldsymbol{Z}_d[(u, u')]\right) \leq 4 L_s^2 D B^2. \tag{174}$$

**Proof** We write the variance as follows:

$$\mathrm{Var}\left(\sum_{d \in [D]} \boldsymbol{Z}_d[(u, u')]\right)$$

$$= \sum_{d, d' \in [D] \times [D]} \mathrm{Cov}(\boldsymbol{Z}_d[(u, u')], \boldsymbol{Z}_{d'}[(u, u')]) \tag{175}$$

$$= \sum_{d, d' \in [D] \times [D]} \mathbb{E}\Big[ \Big( s(\boldsymbol{x}^{(q)}(u)[i] - \boldsymbol{x}^{(c)}(u')[i]) - \mathbb{E}[s(\boldsymbol{x}^{(q)}(u)[i] - \boldsymbol{x}^{(c)}(u')[i])] \Big)$$

$$\cdot \Big( s(\boldsymbol{x}^{(q)}(u)[j] - \boldsymbol{x}^{(c)}(u')[j]) - \mathbb{E}[s(\boldsymbol{x}^{(q)}(u)[j] - \boldsymbol{x}^{(c)}(u')[j])] \Big) \Big] \tag{176}$$

We refer to $\boldsymbol{x}^{(q)}(u)[i] - \boldsymbol{x}^{(c)}(u')[i]$ as $\delta_d$ so that Eq. (176) can be rewritten as

$$= \sum_{d,d' \in [D] \times [D]} \mathbb{E}\left[(s(\delta_d) - \mathbb{E}[s(\delta_d)])(s(\delta_{d'}) - \mathbb{E}[s(\delta_{d'})])\right] \tag{177}$$

$$= \sum_{d,d' \in [D] \times [D]} \mathbb{E}\left[(s(\delta_d) - s(0) - \mathbb{E}[s(\delta_d) - s(0)])(s(\delta_{d'}) - s(0) - \mathbb{E}[s(\delta_{d'}) - s(0)])\right] \tag{178}$$

$$= \sum_{d,d' \in [D] \times [D]} \mathbb{E}[(s(\delta_d) - s(0))(s(\delta_{d'}) - s(0))] - \mathbb{E}[(s(\delta_d) - s(0))]\mathbb{E}[(s(\delta_{d'}) - s(0))] \tag{179}$$

$$= \sum_{d,d' \in [D] \times [D]} \mathbb{E}[(s(\delta_d) - s(0))(s(\delta_{d'}) - s(0))] - \left(\sum_{d \in [D]} \mathbb{E}[(s(\delta_d) - s(0))]\right)^2. \tag{180}$$

We can write $|s(\delta_d) - s(0)| \le \left|\frac{\partial s}{\partial \delta}\right|_{\max_{(-2B,2B)}} |\delta_d| = L_s|\delta_d|$. Thus Eq. (180) can be reduced to

$$\sum_{d,d' \in [D] \times [D]} \mathbb{E}[(s(\delta_d) - s(0))(s(\delta_{d'}) - s(0))] - \left(\sum_{d \in [D]} \mathbb{E}[(s(\delta_d) - s(0))]\right)^2$$

$$\le \sum_{d,d' \in [D] \times [D]} \mathbb{E}[(s(\delta_d) - s(0))(s(\delta_{d'}) - s(0))] \tag{181}$$

$$\le L_s^2 \sum_{d,d' \in [D] \times [D]} \mathbb{E}[|\delta_d||\delta_{d'}|] = L_s^2 \mathbb{E}[\|\delta\|_1^2] \tag{182}$$

$$\le L_s^2 \cdot \mathbb{E}[D\|\delta\|_2^2] \le 4L_s^2 \cdot D \cdot B^2 \tag{183}$$

Where the final bound in Eq. (183) uses the bound on $\boldsymbol{x}^{(\bullet)}(u)$. ∎

**Lemma 17** (Variance Bound for $\boldsymbol{Z}_d$). *Let $\boldsymbol{Z}_d$ be defined as in Eq. (143). Given that $\|\boldsymbol{x}^{(c)}(u')\|_2, \|\boldsymbol{x}^{(q)}(u)\|_2 \le B$, then we can bound*

$$\mathrm{Var}(\boldsymbol{Z}_d[(u,u')]) \le \frac{4L_s^2 B^2}{D} \tag{184}$$

*Proof for the Variance Bound* We follow similar steps as the proof for Lemma 16.

$$\mathrm{Var}(\boldsymbol{Z}_d[(u,u')]) \le \mathbb{E}\left[(s(\delta_d) - \mathbb{E}[s(\delta_d)])(s(\delta_d) - \mathbb{E}[s(\delta_d)])\right] \tag{185}$$

$$\le \mathbb{E}\left[(s(\delta_d) - s(0))^2\right] - \mathbb{E}[s(\delta_d) - s(0)]^2 \tag{186}$$

$$\le \mathbb{E}\left[(s(\delta_d) - s(0))^2\right] \tag{187}$$

$$\le L_s^2 \mathbb{E}[\delta_d^2] = L_s^2 \left(\frac{1}{D} \sum_{d \in [D]} \mathbb{E}[\delta_d^2]\right) \quad \text{as } \mathbb{E}[\delta_1^2] = \mathbb{E}[\delta_2^2] = \cdots = \mathbb{E}[\delta_D^2] \tag{188}$$

$$= L_s^2 \left(\frac{1}{D} \mathbb{E}[\|\delta\|_2^2]\right) \le \frac{L_s^2}{D} \cdot 4B^2. \tag{189}$$

Here, the final bound in Eq. (189) uses the bound on $\boldsymbol{x}^{(\bullet)}(u)$. ∎

### E.2.4 PROOFS OF LSH RESULTS

We show that our random hyperplane hashing on $\widehat{T}_{q,d}$ and $\widehat{T}_{c,d}$ used in Eq. (9) gives us produce a valid LSH for the similarity measure $\text{sim}_d(G_c, G_q)$ and $\text{sim}(G_c, G_q)$. We first establish some key details of our procedure.

**Augmentation of Low Pass Filter with scoring function** $s(\cdot)$  Since $s(\cdot)$ is bounded and absolutely convergent, its Fourier transform $S(\iota\omega) = \frac{1}{2\pi}\int_{x\in\mathbb{R}} s(x)\exp(-\iota\omega x)dx$ is finite. This allows us to write $s(x) = \int_{\omega\in\mathbb{R}} S(\iota\omega)\exp(\iota\omega x)d\omega$. However, for simple scoring functions, $S(\iota\omega)$ imparts significant amount of high frequency signals, which leads to divergence of the integral of $|S(\iota\omega)|$. To tackle this problem, we multiply a smooth low pass filter $\text{LPF}_\lambda(\omega) = \frac{1}{2\pi}\frac{\lambda}{\lambda+\iota\omega}$ with $S(\iota\omega)$ to obtain $S_\lambda(\iota\omega) = \text{LPF}_\lambda(\omega)S(\iota\omega)$ which is absolutely integrable, *i.e.*, $\int_{\omega\in\mathbb{R}} |S_\lambda(\iota\omega)|d\omega < \infty$.

We first demonstrate that the integral $\int_{\omega\in\mathbb{R}} |\text{Re}(S(\iota\omega))| + |\text{Im}(S(\iota\omega))|\, d\omega$ may diverge in the absence of smoothing. Consider $\rho$ as the hinge function, $\rho(x) = [x]_+$. Applying the construction, we obtain $s(\bullet)$ and $S(\bullet)$ similar to the formulation in (Roy et al., 2023).

$$s(x) = \begin{cases} x_{\max} & -x_{\max} \le x \le 0 \\ x_{\max} - x & 0 < x \le x_{\max} \\ 0 & \text{otherwise} \end{cases} \tag{190}$$

$$S(\iota\omega) = \left[ x_{\max}\frac{\sin\omega x_{\max}}{2\pi\omega} + 2\frac{\sin^2(\frac{\omega x_{\max}}{2})}{2\pi\omega^2} \right] + \iota\left[ \frac{\sin\omega x_{\max}}{2\pi\omega^2} - \frac{x_{\max}\cos\omega x_{\max}}{2\pi\omega} \right] \tag{191}$$

In order to show that the integral diverges, it suffices to show that the +ve tail diverges–

$$\int_{\omega_0}^{\infty} |\text{Re}(S(\iota\omega))| + |\text{Im}(S(\iota\omega))|\, d\omega \ge \int_{\omega_0}^{\infty} |\text{Re}(S(\iota\omega)) + \text{Im}(S(\iota\omega))|\, d\omega \quad \text{using } |a+b| \le |a| + |b| \tag{192}$$

$$= \int_{\omega_0}^{\infty} \left| x_{\max}\frac{\sin\omega x_{\max}}{2\pi\omega} + 2\frac{\sin^2(\frac{\omega x_{\max}}{2})}{2\pi\omega^2} + \frac{\sin\omega x_{\max}}{2\pi\omega^2} - \frac{x_{\max}\cos\omega x_{\max}}{2\pi\omega} \right| d\omega \tag{193}$$

$$= \int_{\omega_0}^{\infty} \left| \left( x_{\max}\frac{\sin\omega x_{\max}}{2\pi\omega} - \frac{x_{\max}\cos\omega x_{\max}}{2\pi\omega} \right) + \left( 2\frac{\sin^2(\frac{\omega x_{\max}}{2})}{2\pi\omega^2} + \frac{\sin\omega x_{\max}}{2\pi\omega^2} \right) \right| d\omega \tag{194}$$

$$\ge \int_{\omega_0}^{\infty} \left| x_{\max}\frac{\sin\omega x_{\max}}{2\pi\omega} - \frac{x_{\max}\cos\omega x_{\max}}{2\pi\omega} \right| d\omega - \int_{\omega_0}^{\infty} \left| 2\frac{\sin^2(\frac{\omega x_{\max}}{2})}{2\pi\omega^2} + \frac{\sin\omega x_{\max}}{2\pi\omega^2} \right| d\omega \tag{195}$$

The second term is finite; hence we focus on the first term. Choose $\omega_0 x_{\max} = 2\pi n_0 + \frac{\pi}{4}$ for a natural number $n_0$. This allows us to write

$$\int_{\omega_0}^{\infty} \left| x_{\max}\frac{\sin\omega x_{\max}}{2\pi\omega} - \frac{x_{\max}\cos\omega x_{\max}}{2\pi\omega} \right| d\omega = \int_{\omega_0}^{\infty} \frac{x_{\max}\sqrt{2}}{2\pi\omega} \left| \sin(\omega x_{\max} - \frac{\pi}{4}) \right| d\omega \tag{196}$$

$$= \int_{2\pi n_0 + \frac{\pi}{4}}^{\infty} \frac{\sqrt{2}}{2\pi\omega} \left| \sin(t - \frac{\pi}{4}) \right| dt \quad \text{substituting } t = \omega x_{\max}. \tag{197}$$

$$= \sum_{n=2n_0}^{\infty} \int_{\pi n + \frac{\pi}{4}}^{\pi(n+1)+\frac{\pi}{4}} \frac{\sqrt{2}}{2\pi\omega} \left| \sin(t - \frac{\pi}{4}) \right| dt \tag{198}$$

$$\ge \sum_{n=2n_0}^{\infty} \frac{\sqrt{2}}{2\pi(\pi(n+1)+\frac{\pi}{4})} \int_{\pi n+\frac{\pi}{4}}^{\pi(n+1)+\frac{\pi}{4}} \left| \sin(t - \frac{\pi}{4}) \right| dt \tag{199}$$

$$= \sum_{n=2n_0}^{\infty} \frac{\sqrt{2}}{2\pi(\pi(n+1)+\frac{\pi}{4})} \cdot 2 > \sum_{n=2n_0}^{\infty} \frac{\sqrt{2}}{\pi^2(n+2)} = \infty \tag{200}$$

Finally, we show that that after the low pass filter is applied, the resultant integral is $\int_{\omega\in\mathbb{R}} |\text{Re}(S_\lambda(\iota\omega))| + |\text{Im}(S_\lambda(\iota\omega))|\, d\omega < \infty$ integrable for the general $s$ function considered in

this paper.

$$|\text{Re}(S_\lambda(\iota\omega))| + |\text{Im}(S_\lambda(\iota\omega))| \le \sqrt{2}|S_\lambda(\iota\omega)| \quad \text{Modulus of the complex number} \tag{201}$$

$$= \sqrt{2}|S(\iota\omega)| \cdot |\text{LPF}_\lambda(\omega)| \tag{202}$$

As $s(\bullet)$ is a measurable, bounded, absolutely integrable function, we know that $\lim_{\omega \to \pm\infty} |S(\iota\omega)| = 0$ by the Riemann-Lebesgue Lemma (Bochner et al., 1949).

Thus, $|S(\iota\omega)|$ is $o(1)$. $|\text{LPF}_\lambda(\omega)| = \frac{1}{2\pi} \frac{\lambda}{\sqrt{\lambda^2+\omega^2}} \sim \frac{1}{|\omega|}$. Thus, $|S_\lambda(\iota\omega)| = o(\frac{1}{|\omega|})$, and thus, $\int_{-\infty}^{\infty} |S_\lambda(\iota\omega)| d\omega < \infty$.

$$\int_{\omega \in \mathbb{R}} |\text{Re}(S_\lambda(\iota\omega))| + |\text{Im}(S_\lambda(\iota\omega))| \, d\omega \le \int_{\omega \in \mathbb{R}} \sqrt{2}|S_\lambda(\iota\omega)| d\omega < \infty \tag{203}$$

**Proof that RH on the approximate Fourier vectors $\widehat{T}_{q,d}$ and $\widehat{T}_{c,d}$ give LSH**    Finally, we show our results which shows that the above Algorithms result in valid LSH.

**Theorem 18.** *Let $\mathrm{sim}(\bullet, \bullet)$ and $\mathrm{sim}_d(\bullet, \bullet)$ be defined as in Eq. (2) and Eq. (3) respectively. We compute $h^{(d)}(G_c) = \mathrm{sign}(\boldsymbol{w}^\top \widehat{\boldsymbol{T}}_{q,d})$ with $\boldsymbol{w} \in \mathcal{N}(0, \mathbb{I})$. Then we have the following results:*

1. *(LSH for $\mathrm{sim}_d(\bullet, \bullet)$) For $\epsilon > 0$, there exist $p, p', \lambda_{\min}(\epsilon) > 0$ and $M_{\min}(\epsilon) > 0$ such that the above random hyperplane hashing will give a $(S_0, \gamma S_0, p, p')$-ALSH for $\mathrm{sim}_d(\bullet, \bullet)$ when $\lambda > \lambda_{\min}(\epsilon)$, $M > M_{\min}(\epsilon)$.*
2. *(LSH for $\mathrm{sim}(\bullet, \bullet)$) For $\epsilon, \epsilon' > 0$, there exists $\widehat{p}, \widehat{p}', \lambda_{\min}(\epsilon, \epsilon') > 0$ and $M_{\min}(\epsilon, \epsilon') > 0$ such that the above random hyperplane hashing will give a $(S_1, \gamma S_1, \widehat{p}, \widehat{p}')$-ALSH for $\mathrm{sim}(\bullet, \bullet)$ when $\lambda > \lambda_{\min}(\epsilon, \epsilon')$, $M > M_{\min}(\epsilon, \epsilon')$ and $D > 1/\epsilon^2\epsilon'$.*

**Proof of (1)**    Assume $L_s$ is the Lipschitz constant for $s(\bullet)$ and $L_{\cos}$ is Lipschitz constant for $\cos^{-1}$; $\delta_{\max} \triangleq \max_{c,q} ||\mathrm{SORT}(\boldsymbol{x}^{(q)}) - \mathrm{SORT}(\boldsymbol{x}^{(c)})||_\infty$ and $x_{\max} = \max\{||\boldsymbol{X}^{(q)}||_{\infty,\infty}, ||\boldsymbol{X}^{(c)}||_{\infty,\infty}\}$. Our random projection hashing is finally based on the similarity measure $\widehat{\mathrm{sim}}_d$ from Section 4, which is the Monte Carlo estimate of $\mathrm{sim}_d^{\mathrm{LPF}}$, which is the low-pass filtered version of $\mathrm{sim}_d$, as defined in Eq. (7).:

$$\widehat{\mathrm{sim}}_d(G_c, G_q) \triangleq \frac{1}{M} \widehat{\boldsymbol{T}}_{q,d}^\top \widehat{\boldsymbol{T}}_{c,d} \tag{204}$$

In the following proofs, we shall trace back the approximations from sim leading up to $\widehat{\mathrm{sim}}_d$, and appropriately bound the differences. Let $I_\lambda \triangleq \int_{\mathbb{R}} |\mathrm{Re}(S_\lambda(\iota\omega))| + |\mathrm{Im}(S_\lambda(\iota\omega))| d\omega$. Then,

$$||\boldsymbol{T}_{\bullet,d}(\omega)||_2^2 = \frac{|\mathrm{Re}(S_\lambda(\iota\omega))| + |\mathrm{Im}(S_\lambda(\iota\omega))|}{\frac{|\mathrm{Re}(S_\lambda(\iota\omega))| + |\mathrm{Im}(S_\lambda(\iota\omega))|}{I_\lambda}} = I_\lambda \tag{205}$$

We also observe that $||\boldsymbol{T}_{\bullet,d}(\boldsymbol{\omega})||_2^2 = nI_\lambda$ and $||\widehat{\boldsymbol{T}}_{\bullet,d}||_2^2 = MnI_\lambda$. From now on we drop $d$ from $f^{(d)}(G_q)$ and $h^{(d)}(G_c)$.

$$\mathrm{Pr}_{f,h}\left(f(G_q) = h(G_c)|\boldsymbol{\omega}\right) = 1 - \frac{1}{\pi}\cos^{-1}\left(\frac{\widehat{\boldsymbol{T}}_{q,d}^\top \widehat{\boldsymbol{T}}_{c,d}}{||\widehat{\boldsymbol{T}}_{q,d}||_2 \cdot ||\widehat{\boldsymbol{T}}_{c,d}||_2}\right) \tag{206}$$

$$= 1 - \frac{1}{\pi}\cos^{-1}\left(\frac{\widehat{\boldsymbol{T}}_{q,d}^\top \widehat{\boldsymbol{T}}_{c,d}}{||\widehat{\boldsymbol{T}}_{q,d}||_2 \cdot ||\widehat{\boldsymbol{T}}_{c,d}||_2}\right)$$

$$+ \frac{1}{\pi}\cos^{-1}\left(\frac{\mathrm{sim}_d^{\mathrm{LPF}}(G_c, G_q)}{\int_{\mathbb{R}} ||\boldsymbol{T}_{q,d}(\boldsymbol{\omega})||_2 \cdot ||\boldsymbol{T}_{c,d}(\boldsymbol{\omega})||_2 p_\lambda(\boldsymbol{\omega})d\boldsymbol{\omega}}\right) \tag{207}$$

$$- \frac{1}{\pi}\cos^{-1}\left(\frac{\mathrm{sim}_d^{\mathrm{LPF}}(G_c, G_q)}{\int_{\mathbb{R}} ||\boldsymbol{T}_{q,d}(\boldsymbol{\omega})||_2 \cdot ||\boldsymbol{T}_{c,d}(\boldsymbol{\omega})||_2 p_\lambda(\boldsymbol{\omega})d\boldsymbol{\omega}}\right) \tag{208}$$

$$= 1 \underbrace{- \frac{1}{\pi}\cos^{-1}\left(\frac{\widehat{\mathrm{sim}}_d(G_c, G_q)}{nI_\lambda}\right) + \frac{1}{\pi}\cos^{-1}\left(\frac{\mathrm{sim}_d^{\mathrm{LPF}}(G_c, G_q)}{nI_\lambda}\right)}_{\mathcal{I}_1}$$

$$\underbrace{- \frac{1}{\pi}\cos^{-1}\left(\frac{\mathrm{sim}_d^{\mathrm{LPF}}(G_c, G_q)}{nI_\lambda}\right) + \frac{1}{\pi}\cos^{-1}\left(\frac{\mathrm{sim}_d(G_c, G_q)}{nI_\lambda}\right)}_{\mathcal{I}_2}$$

$$- \frac{1}{\pi}\cos^{-1}\left(\frac{\mathrm{sim}_d(G_c, G_q)}{nI_\lambda}\right) \tag{209}$$

Note that the argument $\mathrm{sim}_d(G_c, G_q)/nI_\lambda$ in the final term must reside within the domain of $\cos^{-1}$. Since $I_\lambda$ is monotonically increasing in $\lambda$, it suffices to require $\lambda > \inf_\lambda\{\lambda : I_\lambda > s_{\max}/n\}$.

We shall now bound each of the terms in Eq. (209)

$$|\mathcal{I}_1| \leq \frac{1}{\pi} L_{\cos} \frac{1}{nI_\lambda} \left| \widehat{\text{sim}}_d(G_c, G_q) - \text{sim}_d^{\text{LPF}}(G_c, G_q) \right| \tag{210}$$

$$\mathbb{E}_{\boldsymbol{\omega}}[|\mathcal{I}_1|] \leq \frac{L_{\cos}}{\pi nI_\lambda} \mathbb{E} \left| \widehat{\text{sim}}_d(G_c, G_q) - \text{sim}_d^{\text{LPF}}(G_c, G_q) \right| \tag{211}$$

$$\leq \frac{L_{\cos}}{\pi nI_\lambda} \sqrt{\frac{n}{M} \mathbb{E} \left( ||\boldsymbol{T}_{q,d}(\omega_u)||_2^2 ||\boldsymbol{T}_{c,d}(\omega_u)||_2^2 \right)} \quad \text{(Lemma 19)} \tag{212}$$

$$= \frac{L_{\cos}}{\pi nI_\lambda} \sqrt{\frac{nI_\lambda^2}{M}} = \frac{L_{\cos}}{\pi \sqrt{Mn}} \tag{213}$$

As $\cos^{-1}$ is monotonically decreasing, and Lipschitz in our context, we can use the bound in Lemma 20, *i.e.*,

$$-\frac{L_{\cos}}{\pi I_\lambda} \left( \frac{L_s}{\lambda} + \frac{s_{\max}}{\lambda} \frac{e^{-1}}{2x_{\max} - \delta_{\max}} \right) \leq \mathcal{I}_2 \leq \frac{L_{\cos}}{\pi I_\lambda} \frac{L_s}{\lambda} \tag{214}$$

Thus,

$$\Pr_{f,h}(f(G_q) = h(G_c)) \leq 1 - \frac{1}{\pi} \cos^{-1} \left( \frac{\text{sim}_d(G_c, G_q)}{nI_\lambda} \right) + \frac{L_{\cos}}{\pi \sqrt{Mn}} + \frac{L_{\cos}}{\pi I_\lambda} \frac{L_s}{\lambda} \tag{215}$$

$$\Pr_{f,h}(f(G_q) = h(G_c)) \geq 1 - \frac{1}{\pi} \cos^{-1} \left( \frac{\text{sim}_d(G_c, G_q)}{nI_\lambda} \right) - \frac{L_{\cos}}{\pi \sqrt{Mn}} \tag{216}$$

$$- \frac{L_{\cos}}{\pi \lambda I_\lambda} \left( L_s + s_{\max} \frac{e^{-1}}{2x_{\max} - \delta_{\max}} \right) \tag{217}$$

Using Lagrange's mean value theorem, we have:

$$\frac{1}{\pi} \left[ \cos^{-1} \left( \frac{\gamma S_0}{nI_\lambda} \right) - \cos^{-1} \left( \frac{S_0}{nI_\lambda} \right) \right] = \frac{1}{\pi} \left( \frac{(\gamma - 1)S_0}{nI_\lambda} \right) \left[ (\cos^{-1})'(t) \right] \quad t \in \left( \frac{\gamma S_0}{nI_\lambda}, \frac{S_0}{nI_\lambda} \right) \tag{218}$$

$$\geq \frac{(1 - \gamma)S_0}{\pi nI_\lambda} \quad \text{as } (\cos^{-1})'(t) \leq -1 \tag{219}$$

From the bounds obtained in Eq. (215) and Eq. (217), we have

$$p' = 1 - \frac{1}{\pi} \cos^{-1} \left( \frac{\gamma S_0}{nI_\lambda} \right) + \frac{L_{\cos}}{\pi \sqrt{Mn}} + \frac{L_{\cos}}{\pi I_\lambda} \frac{L_s}{\lambda} \tag{220}$$

$$p = 1 - \frac{1}{\pi} \cos^{-1} \left( \frac{S_0}{nI_\lambda} \right) - \frac{L_{\cos}}{\pi \sqrt{Mn}} - \frac{L_{\cos}}{\pi \lambda I_\lambda} \left( L_s + s_{\max} \frac{e^{-1}}{2x_{\max} - \delta_{\max}} \right) \tag{221}$$

We have $p > p'$ if

$$\frac{1}{\pi} \left[ \cos^{-1} \left( \frac{\gamma S_0}{nI_\lambda} \right) - \cos^{-1} \left( \frac{S_0}{nI_\lambda} \right) \right] > \frac{2L_{\cos}}{\pi \sqrt{Mn}} + \frac{L_{\cos}}{\pi \lambda I_\lambda} \left( 2L_s + s_{\max} \frac{e^{-1}}{2x_{\max} - \delta_{\max}} \right) \tag{222}$$

Using Eq. (219), the sufficient conditions for the above equation are:

$$\frac{2}{3} \frac{(1 - \gamma)S_0}{\pi nI_\lambda} > \frac{2L_{\cos}}{\pi \sqrt{Mn}} \tag{223}$$

$$\frac{1}{3} \frac{(1 - \gamma)S_0}{\pi nI_\lambda} > \frac{L_{\cos}}{\pi \lambda I_\lambda} \left( 2L_s + s_{\max} \frac{e^{-1}}{2x_{\max} - \delta_{\max}} \right) \tag{224}$$

We obtain

$$\lambda > \frac{3L_{\cos}n \left( 2L_s + s_{\max} \frac{e^{-1}}{2x_{\max} - \delta_{\max}} \right)}{(1 - \gamma)S_0} \qquad M > \frac{9L_{\cos}^2 nI_\lambda^2}{(1 - \gamma)^2 S_0^2} \tag{225}$$

∎

This is a sufficient condition for the LSH to hold that denotes the existence of appropriate $n_{\min}, \lambda_{\min}$ such that the LSH holds. We can also choose other bounds on $M$ and $\lambda$ such that the above conditions are satisfied, and the LSH is valid. We now show the second part of the theorem.

**Proof for (2)** Now that we have shown that we have a $(S_0, \gamma S_0, p, p')$-ALSH for $\text{sim}_d$, we show that it is a hash for $\text{sim}$. We shall use the concentration result in Proposition 7. Given $\left| \frac{1}{D} \text{sim}(G_c, G_q) - \right.$

$\operatorname{sim}_d(G_c, G_q)| \leq \epsilon$ with probability $1 - \beta_0\delta$, we can express this as:

$$-\epsilon \leq \frac{1}{D}\operatorname{sim}(G_c, G_q) - \operatorname{sim}_d(G_c, G_q) \leq \epsilon \tag{226}$$

with probability $1 - \beta_0\delta$. Here, the randomness arises from $\operatorname{sim}_d$. This can be rewritten as:

$$-\epsilon \leq \frac{1}{D}\operatorname{sim}(G_c, G_q) - \operatorname{sim}_d(G_c, G_q) \leq \epsilon \tag{227}$$

$$\implies \begin{cases} \operatorname{sim}_d(G_c, G_q) \leq \frac{1}{D}\operatorname{sim}(G_c, G_q) + \epsilon & \text{(condition 1)}, \\ \operatorname{sim}_d(G_c, G_q) \geq \frac{1}{D}\operatorname{sim}(G_c, G_q) - \epsilon & \text{(condition 2)}. \end{cases} \tag{228}$$

Both condition 1 and condition 2 have probability $\geq 1 - \beta_0\delta$. Here, $p$ and $p'$ are computed in the proof of (1).

1. Condition 1 implies that if $\frac{1}{D}\operatorname{sim}(G_c, G_q) \leq \gamma S_0 - \epsilon$, then $\operatorname{sim}_d(G_c, G_q) \leq \gamma S_0$ with probability $\geq 1 - \beta_0\delta$. Therefore, when $\frac{1}{D}\operatorname{sim}(G_c, G_q) \leq \gamma S_0 - \epsilon$

$$\Pr{}_{f,h}(f(G_q) = h(G_c))$$
$$= \Pr(f(G_q) = h(G_c) \mid \operatorname{sim}_d(G_c, G_q) \leq \gamma S_0) \cdot \Pr(\operatorname{sim}_d(G_c, G_q) \leq \gamma S_0)$$
$$+ \Pr(f(G_q) = h(G_c) \mid \operatorname{sim}_d(G_c, G_q) > \gamma S_0) \cdot \Pr(\operatorname{sim}_d(G_c, G_q) > \gamma S_0) \tag{229}$$

$$\leq p'(1 - \beta_0\delta) + 1 \cdot \beta_0\delta \tag{230}$$

2. Condition 2 implies that if $\frac{1}{D}\operatorname{sim}(G_c, G_q) \geq S_0 + \epsilon$, then $\operatorname{sim}_d(G_c, G_q) \geq S_0$ with probability $\geq 1 - \beta_0\delta$. Therefore, when $\frac{1}{D}\operatorname{sim}(G_c, G_q) \geq S_0 + \epsilon$

$$\Pr{}_{f,h}(f(G_q) = h(G_c))$$
$$= \Pr(f(G_q) = h(G_c) \mid \operatorname{sim}_d(G_c, G_q) \geq S_0) \cdot \Pr(\operatorname{sim}_d(G_c, G_q) \geq S_0)$$
$$+ \Pr(f(G_q) = h(G_c) \mid \operatorname{sim}_d(G_c, G_q) < S_0) \cdot \Pr(\operatorname{sim}_d(G_c, G_q) < S_0) \tag{231}$$
$$\geq \Pr(f(G_q) = h(G_c) \mid \operatorname{sim}_d(G_c, G_q) \geq S_0) \Pr(\operatorname{sim}_d(G_c, G_q) \geq S_0) \tag{232}$$
$$\geq p(1 - \beta_0\delta) \tag{233}$$

Then, we have a $(D(S_0 + \epsilon), D(\gamma S_0 - \epsilon), p(1 - \beta_0\delta), p'(1 - \beta_0\delta) + \beta_0\delta)$-ALSH if

$$p(1 - \beta_0\delta) > p'(1 - \beta_0\delta) + \beta_0\delta \tag{234}$$

$$p > p' + \frac{\beta_0\delta}{1 - \beta_0\delta} \tag{235}$$

We shall find a sufficient condition for Eq. (235) to hold. We use the expressions in the previous results. Finally, we reparameterize the problem with $S_1 \triangleq D(S_0 + \epsilon)$, $\gamma_1 S_1 \triangleq D(\gamma S_0 - \epsilon)$ with $\gamma_1 = \gamma - \frac{\epsilon}{S_0} < \gamma < 1$, $\widehat{p} = p(1 - \beta_0\delta)$ and $\widehat{p}' = p'(1 - \beta_0\delta) + \beta_0\delta$

For $p_\lambda(\omega) \propto |\operatorname{Re}(S_\lambda(\omega))| + |\operatorname{Im}(S_\lambda(\omega))|$, the above criteria can be achieved by taking

$$M > n \left( \frac{2L_{\cos}}{\frac{(1-\gamma)S_0}{2I_\lambda} + \frac{n\pi\beta_0\delta}{1-\beta_0\delta}} \right)^2 \tag{236}$$

for $\lambda > \frac{2L_{\cos}n\left(2L_s + s_{\max}\frac{e^{-1}}{2x_{\max} - \delta_{\max}}\right)}{(1-\gamma)S_0}$. Reparameterizing with $S_1, \gamma_1$, we obtain

$$M > n \left( \frac{2L_{\cos}}{\frac{(1-\gamma_1)S_1/D - 2\epsilon}{2I_\lambda} + \frac{n\pi\beta_0\delta}{1-\beta_0\delta}} \right)^2, \quad D > \frac{1}{\delta\epsilon^2}, \quad \lambda > \frac{2L_{\cos}n\left(2L_s + s_{\max}\frac{e^{-1}}{2x_{\max} - \delta_{\max}}\right)}{(1-\gamma_1)S_1/D - 2\epsilon} \tag{237}$$

As before, we pick $M_{\min}, \lambda_{\min}$ such that the above conditions are satisfied. We can also choose other bounds on $M$ and $\lambda$ such that the above conditions are satisfied, and the LSH is valid. ∎

Note that here we have considered the randomness of model initialization to be part of the randomness of the hashing routine.

### E.2.5 Auxiliary results used to prove results in this subsection E.2.4

**Lemma 19.** *Suppose* $\mathrm{sim}_d$ *is defined as Eq. (7) and* $\widehat{\mathrm{sim}}_d$ *is defined as Eq. (9). Then, we have the following concentration bound:*

$$\mathbb{E}\left|\widehat{\mathrm{sim}}_d(G_c, G_q) - \mathrm{sim}_d(G_c, G_q)\right| \leq \sqrt{\frac{n}{M}\mathbb{E}\left[\left(\boldsymbol{T}_{q,d}(\omega_u)^\top \boldsymbol{T}_{c,d}(\omega_u)\right)^2\right]} \tag{238}$$

**Proof** We observe that:

$$\mathbb{E}\left|\widehat{\mathrm{sim}}_d(G_c, G_q) - \mathrm{sim}_d(G_c, G_q)\right|$$

$$\leq \sqrt{\mathbb{E}\left|\widehat{\mathrm{sim}}_d(G_c, G_q) - \mathrm{sim}_d(G_c, G_q)\right|^2} = \sqrt{\mathrm{Var}\left(\widehat{\mathrm{sim}}_d(G_c, G_q)\right)} \tag{239}$$

$$= \sqrt{\mathrm{Var}\left(\frac{1}{M}\sum_{m\in[M]}\sum_{u\in[n]}\boldsymbol{T}_{q,d}(\omega_u)^\top \boldsymbol{T}_{c,d}(\omega_u)\right)} \tag{240}$$

$$= \sqrt{\frac{n}{M}\mathrm{Var}\left(\boldsymbol{T}_{q,d}(\omega_u)^\top \boldsymbol{T}_{c,d}(\omega_u)\right)} = \sqrt{\frac{n}{M}\mathbb{E}\left[\left(\boldsymbol{T}_{q,d}(\omega_u)^\top \boldsymbol{T}_{c,d}(\omega_u)\right)^2\right]} \tag{241}$$

Here, Eq. (241) follows from the i.i.d sampling of $\omega_u$.

**Lemma 20.** *Suppose* $\mathrm{sim}_d$ *is defined as Eq. (7) and* $\widehat{\mathrm{sim}}_d$ *is defined as Eq. (3). Then, we have the following concentration bound:*

$$-\left(\frac{nL_s}{\lambda} + \frac{ns_{\max}}{\lambda}\frac{e^{-1}}{x_{\max} - \delta_{\max}}\right) \leq \mathrm{sim}_d(G_c, G_q) - \mathrm{sim}_d(G_c, G_q) \leq \frac{nL_s}{\lambda} \tag{242}$$

*where* $L_s$ *is the Lipschitz constant for* $s$; $\delta_{\max} \triangleq \max_{c,q}||\mathrm{SORT}(\boldsymbol{x}^{(q)}) - \mathrm{SORT}(\boldsymbol{x}^{(c)})||_\infty$; *and* $\max\{||\boldsymbol{X}^{(q)}||_{\infty,\infty}, ||\boldsymbol{X}^{(c)}||_{\infty,\infty}\} < x_{\max}$

*Proof.* Let $s_\lambda$ denote the fourier inverse of $S_\lambda$.

$$\mathrm{sim}_d(G_c, G_q) = \sum_{u\in[n]}\int_{\mathbb{R}} S_\lambda(\iota\omega)e^{\iota\omega(\boldsymbol{x}^{(q)}(u)[d] - \boldsymbol{x}^{(c)}(u)[d])}d\omega \tag{243}$$

$$= \sum_{u\in[n]} s_\lambda(\boldsymbol{x}^{(q)}(u)[d] - \boldsymbol{x}^{(c)}(u)[d]) \tag{244}$$

We shall bound the deviation of the smoothed score function $s_\lambda$ from the original score function

$$s_\lambda(x) = \int_{\mathbb{R}} s(x - t)\,\mathcal{F}^{-1}[\mathrm{LPF}_\lambda](t)dt \quad \text{using } \mathcal{F}^{-1}[fg] = \mathcal{F}^{-1}[f] * \mathcal{F}^{-1}[g] \tag{245}$$

$$= \int_{\mathbb{R}} s(x - t)\lambda e^{\lambda t}H(-t)dt = \int_{-\infty}^{0} s(x - t)\lambda e^{\lambda t}dt \tag{246}$$

(where $H(\cdot)$ is the Heaviside step function)

$$= \int_0^\infty s(x + \frac{t}{\lambda})e^{-t}dt \quad \text{substitution with } t \mapsto -\lambda t \tag{247}$$

$$= \int_0^\infty s(x)e^{-t}dt + \int_0^\infty (s(x + \frac{t}{\lambda}) - s(x))e^{-t}dt \tag{248}$$

$$= s(x) + \underbrace{\int_0^\infty (s(x + \frac{t}{\lambda}) - s(x))e^{-t}dt}_{\mathcal{I}} \tag{249}$$

We shall use the fact that $s$ is clipped to 0 outside the domain $[-2x_{\max}, 2x_{\max}]$. We have the following possible cases:

Case 1 $x + \frac{t}{\lambda} > 2x_{\max} \implies t > \lambda(2x_{\max} - x)$
Case 2 $2x_{\max} \geq x + \frac{t}{\lambda} \geq -2x_{\max} \implies \lambda(2x_{\max} - x) \geq t > 0 > \lambda(-2x_{\max} - x)$

This lets us split the integral in $\mathcal{I}$ into two in order to bound the term.

$$
\int_0^\infty (s(x + \frac{t}{\lambda}) - s(x))e^{-t}dt = \int_0^{\lambda(2x_{\max}-x)} (s(x + \frac{t}{\lambda}) - s(x))e^{-t}dt
$$
$$
+ \int_{\lambda(2x_{\max}-x)}^\infty (0 - s(x))e^{-t}dt \tag{250}
$$
$$
= \underbrace{\left[ \int_0^{\lambda(2x_{\max}-x)} (s(x + \frac{t}{\lambda}) - s(x))e^{-t}dt \right] - s(x)e^{-\lambda(2x_{\max}-x)}}_{\mathcal{J}}
$$
$$
\tag{251}
$$

We now bound $|\mathcal{J}|$ as follows:

$$
|\mathcal{J}| \leq \int_0^{\lambda(2x_{\max}-x)} L_s \frac{t}{\lambda} e^{-t}dt \quad (s \text{ is Lipschitz with constant } L_s) \tag{252}
$$
$$
= \frac{L_s}{\lambda} \left[ -(t+1)e^{-t} \right]_{t=0}^{\lambda(2x_{\max}-x)} \leq \frac{L_s}{\lambda} \left[ -(t+1)e^{-t} \right]_{t=0}^{\lambda(2x_{\max}+\max\|x\|_\infty)} \tag{253}
$$
$$
= \frac{L_s}{\lambda} \left( 1 - e^{-\lambda(2x_{\max}+\delta_{\max})} - \lambda(2x_{\max}+\delta_{\max})e^{-\lambda(2x_{\max}+\delta_{\max})} \right) \tag{254}
$$
$$
\leq \frac{L_s}{\lambda} \left[ -(t+1)e^{-t} \right]_{t=0}^\infty = \frac{L_s}{\lambda} \cdot 1 \tag{255}
$$

The bound in (253) relies on integrating over a larger interval. This yields the bound Eq. (254). However, for purposes of this proof, we use the looser bound Eq. (255) by integrating over $(0, \infty)$.

Using the fact that $0 \leq s(\cdot) \leq s_{\max}$ in Eq (251)

$$
-|\mathcal{J}| - s_{\max}e^{-\lambda(2x_{\max}-x)} \leq \mathcal{I} \leq |\mathcal{J}| \tag{256}
$$
$$
-\frac{L_s}{\lambda} - s_{\max}e^{-\lambda(2x_{\max}-x)} \leq \mathcal{I} \leq \frac{L_s}{\lambda} \tag{257}
$$
$$
-\frac{L_s}{\lambda} - \frac{s_{\max}e^{-1}}{\lambda(2x_{\max}-\delta_{\max})} \leq \mathcal{I} \leq \frac{L_s}{\lambda} \tag{258}
$$

$\blacksquare$

Eq. (258) blows up near $(2x_{\max} - \delta_{\max}) \approx 0$ as it is a much looser bound than Eq. (257). However we keep it as it leads to a simpler closed form expression for $\lambda$.

# F LIST OF GNNs

We collect the following list from Pytorch Geometric.

1. **GNN**
   (1) Gated GNN (Li et al., 2016; Gilmer et al., 2017) (Already showed)
   (2) GCN (Kipf et al., 2017)
   (3) ChebConv (Defferrard et al., 2016)
   (4) SAGE (Hamilton et al., 2017)
   (5) ResGatedGraphConv (Bresson et al., 2017)
   (6) GAT (Veličković et al., 2018)
   (7) AGNNConv (Thekumparampil et al., 2018)
   (8) GIN (Xu et al., 2019)
   (9) SGConv (Wu et al., 2019)
   (10) TAGConv (Du et al., 2017)
   (11) APPNP (Gasteiger et al., 2018)
   (12) SSGConv (Zhu et al., 2021)
   (13) MFConv (Duvenaud et al., 2015)
2. **Graph Transformers**
   (1) Graph Transformer (GraphGPS-style) (Rampášek et al., 2022)
   (2) Graphormer (Ying et al., 2021)
   (3) Spectral Attention Network (SAN) (Kreuzer et al., 2021)
   (4) Exphormer (Shirzad et al., 2023)
   (5) NodeFormer (Wu et al., 2023)

Here, we will take node embeddings $x$ to be column vectors, but the graph embedding $X$ to have $x$ along rows. As such we will use $\Theta$ for the parameters right multiplied and $W$ for left multiplied. $D, A, L$ refer to the degree, adjacency and Laplacian matrices respectively. Similarly, $\hat{D}, \hat{A}, \hat{L}$ refer to the normalized degree, adjacency and Laplacian matrices respectively.

We demonstrate transformations for various graph layers that can be used to maintain/induce permutations in the output, which would be required for showing exchangeability at a certain layer. Where applicable, we may take arbitrary permutation $\pi_2$ on the input and a corresponding $\pi_1$ in the output. For some cases the permutations are more restrictive (such as $\pi_1 = \pi_2$).

These transformations can then be composed to generate the permutation inducing transformation for the entire network.

We have shown transformation for architectures such as the MLP (FF) and GRU (GRU). For a given permutation (where it is clear from context), we define the transformed versions as follows:

$$\mathrm{GRU}^*(X\pi, H\pi) = \mathrm{GRU}(X, H)\pi$$
$$\mathrm{FF}^*(X\pi) = \mathrm{FF}(X)\pi$$

or if the input and output permutations are different:

$$\mathrm{FF}^*(X\pi_2) = \mathrm{FF}(X)\pi_1$$

## F.1 GRAPH NEURAL NETWORK

Based on the original formulation, $x$ can be row or column vector and therefore $\pi$ is pre-multiplied or post-multiplied.

**(1) GCN** (Kipf et al., 2017):

$$X' = \hat{D}^{-1/2}\hat{A}\hat{D}^{-1/2}X\Theta \tag{259}$$

$$X'\pi = \hat{D}^{-1/2}\hat{A}\hat{D}^{-1/2}X(\Theta\pi) \tag{260}$$

$$X'\pi_1 = \hat{D}^{-1/2}\hat{A}\hat{D}^{-1/2}(X\pi_2)(\pi_2^\top\Theta\pi_1) \tag{261}$$

**(2) ChebConv** (Defferrard et al., 2016): It uses Chebyshev polynomial filters on the rescaled Laplacian. The Chebyshev polynomials are defined as $T_0(x) = 1$, $T_1(x) = x$ and $T_k(x) = 2xT_{k-1}(x) - T_{k-2}(x)$ for $k \geq 2$.

$$\boldsymbol{X}^{(k)} = \sum_{\ell=0}^{K} T_\ell(\tilde{L})\, \boldsymbol{X}^{(k-1)}\Theta_\ell \tag{262}$$

$$\boldsymbol{X}^{(k)}\boldsymbol{\pi}_1 = \sum_{\ell=0}^{K} T_\ell(\tilde{L})\, (\boldsymbol{X}^{(k-1)}\boldsymbol{\pi}_2)(\boldsymbol{\pi}_2^\top \Theta_\ell \boldsymbol{\pi}_1) \tag{263}$$

**(3) SAGEConv** (Hamilton et al., 2017): We take the aggregate function to be permutation equivariant (eg. mean/sum).

$$\boldsymbol{x}_i^{(k)} = \sigma\big(\boldsymbol{W}_1 \boldsymbol{x}_i^{(k-1)} + \boldsymbol{W}_2 \cdot \text{AGGREGATE}(\{\boldsymbol{x}_j^{(k-1)}\})\big) \tag{264}$$

$$\boldsymbol{\pi}\boldsymbol{x}_i^{(k)} = \sigma\big((\boldsymbol{\pi}\boldsymbol{W}_1\boldsymbol{\pi}^\top)\boldsymbol{\pi}\boldsymbol{x}_i^{(k-1)} + (\boldsymbol{\pi}\boldsymbol{W}_2\boldsymbol{\pi}^\top) \cdot \text{AGGREGATE}(\{\boldsymbol{\pi}\boldsymbol{x}_j^{(k-1)}\})\big) \tag{265}$$

or, there may be a layer before the aggregation (allowing for more flexibility in the transformation):

$$\boldsymbol{x}_i^{(k)} = \sigma\big(\boldsymbol{W}_1 \boldsymbol{x}_i^{(k-1)} + \boldsymbol{W}_2 \cdot \text{AGGREGATE}(\{\text{FF}(\boldsymbol{x}_j^{(k-1)})\})\big) \tag{266}$$

$$\boldsymbol{\pi}_1\boldsymbol{x}_i^{(k)} = \sigma\big((\boldsymbol{\pi}_1\boldsymbol{W}_1\boldsymbol{\pi}_2^\top)\boldsymbol{\pi}_2\boldsymbol{x}_i^{(k-1)} + (\boldsymbol{\pi}_1\boldsymbol{W}_2\boldsymbol{\pi}_2^\top) \cdot \text{AGGREGATE}(\{\text{FF}^*(\boldsymbol{\pi}_2\boldsymbol{x}_j^{(k-1)})\})\big) \tag{267}$$

**(4) ResGatedGraphConv** (Bresson et al., 2017): Adds a residual connection over a gated convolution mechanism.

$$\boldsymbol{x}_i^{(k)} = \boldsymbol{W}_1 \boldsymbol{x}_i^{(k-1)} + \sum_{j\in\mathcal{N}(i)} \boldsymbol{W}_2 \boldsymbol{x}_j^{(k-1)} \odot \sigma(\boldsymbol{W}_3 \boldsymbol{x}_i^{(k-1)} + \boldsymbol{W}_4 \boldsymbol{x}_j^{(k-1)}) \tag{268}$$

$$
\begin{aligned}
\boldsymbol{\pi}_1\boldsymbol{x}_i^{(k)} = {} & (\boldsymbol{\pi}_1\boldsymbol{W}_1\boldsymbol{\pi}_2^\top)(\boldsymbol{\pi}_2\boldsymbol{x}_i^{(k-1)}) \\
& + \sum_{j\in\mathcal{N}(i)} (\boldsymbol{\pi}_1\boldsymbol{W}_2\boldsymbol{\pi}_2^\top)(\boldsymbol{\pi}_2\boldsymbol{x}_j^{(k-1)}) \odot \sigma((\boldsymbol{\pi}_1\boldsymbol{W}_3\boldsymbol{\pi}_2^\top)(\boldsymbol{\pi}_2\boldsymbol{x}_i^{(k-1)}) \\
& + (\boldsymbol{\pi}_1\boldsymbol{W}_4\boldsymbol{\pi}_2^\top)(\boldsymbol{\pi}_2\boldsymbol{x}_j^{(k-1)}))
\end{aligned} \tag{269}
$$

**(5) GAT** (Veličković et al., 2018): The attention score $\alpha$ can be made invariant.

$$\boldsymbol{x}_i^{(k)} = \sum_{j\in\mathcal{N}(i)} \alpha_{ij}^{(h)} W^{(h)} \boldsymbol{x}_j^{(k-1)} \tag{270}$$

$$\boldsymbol{\pi}\boldsymbol{x}_i^{(k)} = \sum_{j\in\mathcal{N}(i)} \alpha_{ij}^{(h)} (\boldsymbol{\pi}_h W^{(h)} \boldsymbol{\pi}^\top)\boldsymbol{\pi}\boldsymbol{x}_j^{(k-1)} \tag{271}$$

$$\alpha_{ij} = \frac{\exp\big(\text{LeakyReLU}\big(\mathbf{a}^T[\boldsymbol{W}\boldsymbol{x}_i\|\boldsymbol{W}\boldsymbol{x}_j]\big)\big)}{\sum_{k\in\mathcal{N}(i)\cup\{i\}} \exp\big(\text{LeakyReLU}\big(\mathbf{a}^T[\boldsymbol{W}\boldsymbol{x}_i\|\boldsymbol{W}\boldsymbol{x}_k]\big)\big)} \tag{272}$$

$$\alpha_{ij} = \frac{\exp\big(\text{LeakyReLU}\big(\mathbf{a}^T[\boldsymbol{W}\boldsymbol{\pi}^\top\boldsymbol{\pi}\boldsymbol{x}_i\|\boldsymbol{W}\boldsymbol{\pi}^\top\boldsymbol{\pi}\boldsymbol{x}_j]\big)\big)}{\sum_{k\in\mathcal{N}(i)\cup\{i\}} \exp\big(\text{LeakyReLU}\big(\mathbf{a}^T[\boldsymbol{W}\boldsymbol{\pi}^\top\boldsymbol{\pi}\boldsymbol{x}_i\|\boldsymbol{W}\boldsymbol{\pi}^\top\boldsymbol{\pi}\boldsymbol{x}_k]\big)\big)} \tag{273}$$

If the aggregation is concatenation instead of sum, the output will not be exchangeable for all dimensions. rather, each block of dimensions corresponding to a head will be exchangeable.

**(6) AGNNConv** (Thekumparampil et al., 2018):

$$\boldsymbol{X}' = \boldsymbol{P}\boldsymbol{X} \tag{274}$$

Where,

$$\boldsymbol{P}_{i,j} = \frac{\exp(\beta \cdot \cos(\boldsymbol{x}_i, \boldsymbol{x}_j))}{\sum_{k\in\mathcal{N}(i)\cup\{i\}} \exp(\beta \cdot \cos(\boldsymbol{x}_i, \boldsymbol{x}_k))} = \frac{\exp\left(\beta \cdot \frac{(\boldsymbol{\pi}\boldsymbol{x}_i)^\top \boldsymbol{\pi}\boldsymbol{x}_j}{\|\boldsymbol{\pi}\boldsymbol{x}_i\|\|\boldsymbol{\pi}\boldsymbol{x}_j\|}\right)}{\sum_{k\in\mathcal{N}(i)\cup\{i\}} \exp\left(\beta \cdot \frac{(\boldsymbol{\pi}\boldsymbol{x}_i)^\top \boldsymbol{\pi}\boldsymbol{x}_k}{\|\boldsymbol{\pi}\boldsymbol{x}_i\|\|\boldsymbol{\pi}\boldsymbol{x}_k\|}\right)} \tag{275}$$

So this layer is equivarient to any permutation $\boldsymbol{\pi}$.

**(7) GIN** (Xu et al., 2019):

$$\boldsymbol{X}' = \text{FF}\left((1+\epsilon)\cdot\boldsymbol{X} + \mathbf{A}\boldsymbol{X}\right) \tag{276}$$

$$\boldsymbol{X}'\boldsymbol{\pi}_1 = \text{FF}^*\left((1+\epsilon)\cdot(\boldsymbol{X}\boldsymbol{\pi}_2) + \mathbf{A}(\boldsymbol{X}\boldsymbol{\pi}_2)\right) \tag{277}$$

A powerful injective update via MLP which combines self-feature (with learnable epsilon) plus neighbor sum.

(8) **SGConv** (Wu et al., 2019): A K-step precomputed propagation that simplifies convolution.

$$\boldsymbol{X}' = \left(\mathbf{D}^{-1/2}\,\hat{\mathbf{A}}\,\mathbf{D}^{-1/2}\right)^K \boldsymbol{X}\boldsymbol{\Theta}, \quad \hat{\mathbf{A}} = \mathbf{A} + \mathbf{I} \tag{278}$$

$$\boldsymbol{X}'\boldsymbol{\pi}_1 = \left(\mathbf{D}^{-1/2}\,\hat{\mathbf{A}}\,\mathbf{D}^{-1/2}\right)^K (\boldsymbol{X}\boldsymbol{\pi}_2)(\boldsymbol{\pi}_2^\top\boldsymbol{\Theta}\boldsymbol{\pi}_1) \tag{279}$$

$$\tag{280}$$

(9) **TAGConv** (Du et al., 2017):

$$\boldsymbol{X}' = \sum_{k=0}^{K}\left(\mathbf{D}^{-1/2}\,\mathbf{A}\,\mathbf{D}^{-1/2}\right)^k \boldsymbol{X}\boldsymbol{\Theta}_k \tag{281}$$

$$\boldsymbol{X}'\boldsymbol{\pi}_1 = \sum_{k=0}^{K}\left(\mathbf{D}^{-1/2}\,\mathbf{A}\,\mathbf{D}^{-1/2}\right)^k (\boldsymbol{X}\boldsymbol{\pi}_2)(\boldsymbol{\pi}_2^\top\boldsymbol{\Theta}_k\boldsymbol{\pi}_1) \tag{282}$$

(10) **APPNP** (Gasteiger et al., 2018):

$$\boldsymbol{X}^{(0)} = \boldsymbol{X} \tag{283}$$

$$\boldsymbol{X}^{(k)} = (1-\alpha)\hat{D}^{-1/2}\hat{A}\hat{D}^{-1/2}\boldsymbol{X}^{(k-1)} + \alpha\boldsymbol{X}^{(0)} \tag{284}$$

$$\boldsymbol{X}' = \boldsymbol{X}^{(K)} \tag{285}$$

This layer is equivariant to any permutation $\boldsymbol{\pi}$.

$$\boldsymbol{X}^{(0)}\boldsymbol{\pi} = \boldsymbol{X}\boldsymbol{\pi} \tag{286}$$

$$\boldsymbol{X}^{(k)}\boldsymbol{\pi} = (1-\alpha)\hat{D}^{-1/2}\hat{A}\hat{D}^{-1/2}\boldsymbol{X}^{(k-1)}\boldsymbol{\pi} + \alpha\boldsymbol{X}^{(0)}\boldsymbol{\pi} \tag{287}$$

$$\boldsymbol{X}'\boldsymbol{\pi} = \boldsymbol{X}^{(K)}\boldsymbol{\pi} \tag{288}$$

(11) **SSGConv** (Zhu et al., 2021):

$$\boldsymbol{X}' = (1-\alpha)\left(\mathbf{D}^{-1/2}\,\hat{\mathbf{A}}\,\mathbf{D}^{-1/2}\right)^K \boldsymbol{X}\,\boldsymbol{\Theta}_1 + \alpha\,\boldsymbol{X}\,\boldsymbol{\Theta}_2 \tag{289}$$

$$\boldsymbol{X}'\boldsymbol{\pi}_1 = (1-\alpha)\left(\mathbf{D}^{-1/2}\,\hat{\mathbf{A}}\,\mathbf{D}^{-1/2}\right)^K \boldsymbol{X}\boldsymbol{\pi}_2\,\boldsymbol{\pi}_2^\top\boldsymbol{\Theta}_1\boldsymbol{\pi}_1 + \alpha\,\boldsymbol{X}\boldsymbol{\pi}_2\,\boldsymbol{\pi}_2^\top\boldsymbol{\Theta}_2\boldsymbol{\pi}_1 \tag{290}$$

Skip-connection version of SGConv with initial-feature mixing via $\alpha$.

(12) **MFConv** (Duvenaud et al., 2015): This has a distinct weight matrix for nodes of each degree.

$$\boldsymbol{x}'_i = \boldsymbol{W}_{\deg(i)}\,\boldsymbol{x}_i + \sum_{j\in\mathcal{N}(i)} \hat{\boldsymbol{W}}_{\deg(i)}\,\boldsymbol{x}_j \tag{291}$$

$$\boldsymbol{\pi}_1\boldsymbol{x}'_i = (\boldsymbol{\pi}_1\boldsymbol{W}_{\deg(i)}\boldsymbol{\pi}_2^\top)(\boldsymbol{\pi}_2\boldsymbol{x}_1) + \sum_{j\in\mathcal{N}(1)} (\boldsymbol{\pi}_1\hat{\boldsymbol{W}}_{\deg(i)}\boldsymbol{\pi}_2^\top)(\boldsymbol{\pi}_2\boldsymbol{x}_j) \tag{292}$$

## F.2 GRAPH TRANSFORMERS

**Multi-Head Attention (MHA)** Before examining specific Graph Transformer architectures, we first establish the standard Multi-Head Attention (MHA) mechanism that forms the foundation of most transformer-based models. The MHA operation transforms input representations $\boldsymbol{H}^{(\ell)} \in \mathbb{R}^{n\times d}$ through learned query ($\boldsymbol{Q}$), key ($\boldsymbol{K}$), and value ($\boldsymbol{V}$) projections:

$$\boldsymbol{Q}^{(h)} = \boldsymbol{H}^{(\ell)}\boldsymbol{W}_Q^{(h)}, \quad \boldsymbol{K}^{(h)} = \boldsymbol{H}^{(\ell)}\boldsymbol{W}_K^{(h)}, \quad \boldsymbol{V}^{(h)} = \boldsymbol{H}^{(\ell)}\boldsymbol{W}_V^{(h)} \tag{293}$$

$$\alpha_{ij}^{(h)} = \text{softmax}_j\left(\frac{\boldsymbol{Q}_i^{(h)}(\boldsymbol{K}_j^{(h)})^\top}{\sqrt{d_k}} + B_{ij}\right) \tag{294}$$

$$\boldsymbol{Z}^{(h)} = \alpha^{(h)}\boldsymbol{V}^{(h)} \tag{295}$$

$$\text{MHA}_B(\boldsymbol{H}^{(\ell)}) = \text{Concat}(\boldsymbol{Z}^{(1)}, \ldots, \boldsymbol{Z}^{(\ell)})\boldsymbol{W}_O \tag{296}$$

where each attention head $h \in \{1, \ldots, \ell\}$ computes scaled dot-product attention independently, and $\boldsymbol{W}_O$ projects the concatenated multi-head output. Given the input $\boldsymbol{H} \mapsto \boldsymbol{H}\boldsymbol{\pi}_2$, we can transform $\boldsymbol{W}_Q^{(h)}, \boldsymbol{W}_K^{(h)}$, and $\boldsymbol{W}_V^{(h)}$ as $\boldsymbol{W}^{(h)} \mapsto \boldsymbol{\pi}_2^\top\boldsymbol{W}^{(h)}$. And the output of MHA can be transformed by $\boldsymbol{\pi}_1$ by $\boldsymbol{W}_O \mapsto \boldsymbol{W}_O\boldsymbol{\pi}_1$.

Using the above, we define $\text{MHA}_B^*$ such that $\text{MHA}_B^*(\boldsymbol{X}\boldsymbol{\pi}) = \text{MHA}_B(\boldsymbol{X})\boldsymbol{\pi}$.

Note that in general, different attention mechanisms are dealt with similarly - the attention parameters can be used to undo the effect of a preceding permutation, hence the attention score computation remains unchanged.

Transformer layers also typically include Layer Normalization, that we will largely omit here, as it is straightforward to see that it is permutation equivariant.

**(1) Graph Transformer** (Rampášek et al., 2022):

$$\boldsymbol{Q}^{(h)} = \boldsymbol{H}^{(\ell)}\boldsymbol{W}_Q^{(h)}, \quad \boldsymbol{K}^{(h)} = \boldsymbol{H}^{(\ell)}\boldsymbol{W}_K^{(h)}, \quad \boldsymbol{V}^{(h)} = \boldsymbol{H}^{(\ell)}\boldsymbol{W}_V^{(h)} \tag{297}$$

$$\alpha_{ij}^{(h)} = \mathrm{softmax}_j\Big(\frac{\boldsymbol{Q}_i^{(h)}(\boldsymbol{K}_j^{(h)})^\top}{\sqrt{d_k}} + B_{ij}\Big) \tag{298}$$

$$\boldsymbol{Z}^{(h)} = \alpha^{(h)}\boldsymbol{V}^{(h)} \tag{299}$$

$$\tilde{\boldsymbol{H}}^{(\ell+1)} = \boldsymbol{H}^{(\ell)} + \mathrm{MHA}_B(\boldsymbol{H}^{(\ell)}) \tag{300}$$

$$\boldsymbol{H}^{(\ell+1)} = \tilde{\boldsymbol{H}}^{(\ell+1)} + \mathrm{FF}(\tilde{\boldsymbol{H}}^{(\ell+1)}) \tag{301}$$

We observe that the following transformations are sufficient,

$$\tilde{\boldsymbol{H}}^{(\ell+1)}\boldsymbol{\pi} = \boldsymbol{H}^{(\ell)}\boldsymbol{\pi} + \mathrm{MHA}_B^*(\boldsymbol{H}^{(\ell)}\boldsymbol{\pi}) \tag{302}$$

$$\boldsymbol{H}^{(\ell+1)}\boldsymbol{\pi} = \tilde{\boldsymbol{H}}^{(\ell+1)}\boldsymbol{\pi} + \mathrm{FF}^*(\tilde{\boldsymbol{H}}^{(\ell+1)}\boldsymbol{\pi}) \tag{303}$$

**(2) Graphormer** (Ying et al., 2021): Firstly, the graphormer adds centrality encodings to the node embedding $\boldsymbol{x}^{(0)}$. Hence these encoding require the same permutation as that of the input node features. The graphormer adds spatial and edge encodings as attention biases $B_{ij}$. As our transformation does not affect the Q-K dot product, it does not affect the attention scores.

$$\boldsymbol{Q}^{(h)} = \boldsymbol{H}^{(\ell)}\boldsymbol{W}_Q^{(h)}, \quad \boldsymbol{K}^{(h)} = \boldsymbol{H}^{(\ell)}\boldsymbol{W}_K^{(h)}, \quad \boldsymbol{V}^{(h)} = \boldsymbol{H}^{(\ell)}\boldsymbol{W}_V^{(h)} \tag{304}$$

$$\alpha_{ij}^{(h)} = \mathrm{softmax}_j\Big(\frac{\boldsymbol{Q}_i^{(h)}(\boldsymbol{K}_j^{(h)})^\top}{\sqrt{d_k}} + b_{\mathrm{enc}}^{\mathrm{SPD}}(\mathrm{SPD}(i,j)) + b_{\mathrm{enc}}^{\mathrm{edge}}(\text{edge-path}(i,j))\Big) \tag{305}$$

$$\boldsymbol{Z}^{(h)} = \alpha^{(h)}\boldsymbol{V}^{(h)} \tag{306}$$

$$\boldsymbol{H}^{(\ell+1)} = \mathrm{FF}\big(\boldsymbol{H}^{(\ell)} + \mathrm{MHA}(\boldsymbol{H}^{(\ell)})\big) \tag{307}$$

Hence, the same transformations as the graph transformer follow, as $\alpha_{i,j}^{(h)}$ remains unchanged.

**(3) Spectral Attention Network (SAN)** (Kreuzer et al., 2021):

$$\tilde{\boldsymbol{H}}^{(\ell)} = \boldsymbol{H}^{(\ell)} + \boldsymbol{S} \tag{308}$$

$$\boldsymbol{Q}^{(h)} = \tilde{\boldsymbol{H}}^{(\ell)}\boldsymbol{W}_Q^{(h)}, \quad \boldsymbol{K}^{(h)} = \tilde{\boldsymbol{H}}^{(\ell)}\boldsymbol{W}_K^{(h)}, \quad \boldsymbol{V}^{(h)} = \tilde{\boldsymbol{H}}^{(\ell)}\boldsymbol{W}_V^{(h)} \tag{309}$$

$$\alpha_{ij}^{(h)} = \mathrm{softmax}_j\Big(\frac{\boldsymbol{Q}_i^{(h)}(\boldsymbol{K}_j^{(h)})^\top}{\sqrt{d_k}}\Big) \tag{310}$$

$$\boldsymbol{Z}^{(h)} = \alpha^{(h)}\boldsymbol{V}^{(h)} \tag{311}$$

$$\boldsymbol{H}^{(\ell+1)} = \mathrm{FF}\big(\boldsymbol{H}^{(\ell)} + \mathrm{MHA}(\tilde{\boldsymbol{H}}^{(\ell)})\big) \tag{312}$$

Given the Laplacian eigendecomposition $\boldsymbol{L} = \boldsymbol{U}\boldsymbol{\Lambda}\boldsymbol{U}^\top$, the LPE Transformer processes concatenated eigenvalue-eigenvector pairs $[\lambda_i; \boldsymbol{u}_i]$ to produce learned positional encodings $\boldsymbol{S} \in \mathbb{R}^{n \times d}$. If $\boldsymbol{H}$ is permuted, we require the encodings of the LPE Transformer to also be transformed to permute $\boldsymbol{S}$. The remaining steps in the transformation can proceed as in the previous cases.

$$\tilde{\boldsymbol{H}}^{(\ell)}\boldsymbol{\pi} = \boldsymbol{H}^{(\ell)}\boldsymbol{\pi} + \boldsymbol{S}\boldsymbol{\pi} \tag{313}$$

**(4) Exphormer** (Shirzad et al., 2023): The changes here pertain to the expander graph and the global virtual nodes. As these can be regarded as structural changes to the graph before applying the graph transformer, we can take the same transformations as the graph transformer.

**(5) NodeFormer** (Wu et al., 2023): Notably, the modification over the base graph transformer is related to the computation of the attention, which uses kernelized attention to speed up the otherwise quadratic self-attention. The above outlined transformation are again sufficient as the kernelized operations preserve the transformation to $\boldsymbol{W}_Q, \boldsymbol{W}_K$ so that the attention is still invariant.

**(6) Gophormer** (Zhao et al., 2021): The proximity score term in the attention can be seen as a structural bias that is not affected by the permutations along the embedding dimension. Once

again, by transforming the $\boldsymbol{W}_Q, \boldsymbol{W}_K, \boldsymbol{W}_V$ matrices accordingly, we ensure that the same transformations as the graph transformer follow.

$$\boldsymbol{Q}^{(h)} = \boldsymbol{H}^{(\ell)} \boldsymbol{W}_Q^{(h)}, \quad \boldsymbol{K}^{(h)} = \boldsymbol{H}^{(\ell)} \boldsymbol{W}_K^{(h)}, \quad \boldsymbol{V}^{(h)} = \boldsymbol{H}^{(\ell)} \boldsymbol{W}_V^{(h)} \tag{314}$$

$$\alpha_{uv}^{(h)} = \mathrm{softmax}_{v \in \mathcal{S}_i} \left( \frac{\boldsymbol{Q}_u^{(h)} (\boldsymbol{K}_v^{(h)})^\top}{\sqrt{d_k}} + b^{\mathrm{prox}}(u, v) \right) \tag{315}$$

$$Z_u^{(h)} = \sum_{v \in \mathcal{S}_i} \alpha_{uv}^{(h)} V_v^{(h)} \tag{316}$$

$$\boldsymbol{H}_{\mathcal{S}_i}^{(\ell+1)} = \mathrm{FF}\big(\boldsymbol{H}_{\mathcal{S}_i}^{(\ell)} + \mathrm{MHA}(\boldsymbol{H}_{\mathcal{S}_i}^{(\ell)})\big) \tag{317}$$

**(7) SpecFormer** (Bo et al., 2023): SpecFormer computes a spectral filter $g$ from eigenvalues via a Transformer. Since the eigenvalues $\lambda_i$ are independent of the feature dimension, the filter computation is unaffected by feature permutations $\boldsymbol{\pi}$. The spectral convolution $\boldsymbol{X}' = \boldsymbol{U}\mathrm{Diag}(g)\boldsymbol{U}^\top \boldsymbol{X}$ is naturally equivariant: input $\boldsymbol{X}\boldsymbol{\pi}$ yields output $\boldsymbol{X}'\boldsymbol{\pi}$. For different input/output permutations, we can transform any subsequent linear layer $\boldsymbol{W} \mapsto \boldsymbol{\pi}_2^\top \boldsymbol{W} \boldsymbol{\pi}_1$.

# G   ADDITIONAL DETAILS ABOUT EXPERIMENTS

## G.1   DATASETS

We build retrieval datasets from four benchmarks in the TU Graph Dataset collection (Morris et al., 2020): ptc-fr, ptc-fm, cox2, and ptc-mr. Each dataset contains 500 queries and a corpus of 100,000 graphs, following the setup in (Roy et al., 2022; Lou et al., 2020). To sample graphs, we adopt the BFS-based extraction strategy introduced in (Lou et al., 2020): starting from a randomly chosen node, a BFS traversal is performed until the induced subgraph spans between 5 and 25 nodes. This method is applied independently to construct both query and corpus graphs.

For **subgraph matching (SM)**, binary relevance labels are generated using the VF2 subgraph isomorphism algorithm (Hagberg et al., 2020). A corpus graph $G_c$ is marked relevant to a query $G_q$ if $G_q$ is a subgraph of $G_c$, i.e., $\mathrm{rel}(G_c, G_q) = [\![G_q \subset G_c]\!]$, where $[\![\cdot]\!]$ denotes the indicator function.

For **graph edit distance (GED)**, we use the GEDLIB solver (Blumenthal et al., 2019), setting insertion cost $\mathrm{e}_\oplus = 1$ and deletion cost $\mathrm{e}_\ominus = 2$. Relevance is determined by thresholding the computed GED: $\mathrm{rel}(G_c, G_q) = [\![\mathrm{GED}(G_c, G_q) \leq \mathrm{Thrs}]\!]$, for a fixed threshold Thrs. Results under a symmetric cost setting (Eq. cost GED) with $\mathrm{e}_\oplus = \mathrm{e}_\ominus = 1$ are also reported in Appendix.

For all datasets, we partition the 500 queries into 60% train, 20% validation, and 20% test splits. Dataset statistics for the subgraph matching and GED tasks are summarized in Table 6 and Table 7, respectively.

Table 6: Graph statistics for each dataset generated for Subgraph Matching (SM).

| Dataset | Query Graphs | | Corpus Graphs | | $\mathbb{E}\left[\frac{|y=1|}{|y=0|}\right]$ |
|---|---|---|---|---|---|
| | **Nodes** | **Edges** | **Nodes** | **Edges** | Label |
| | (min / max / avg) | (min / max / avg) | (min / max / avg) | (min / max / avg) | Ratio |
| **PTC-FR** | (6 / 15 / 12.65) | (6 / 15 / 12.41) | (16 / 25 / 18.68) | (15 / 28 / 20.17) | 0.13 |
| **PTC-FM** | (7 / 15 / 12.58) | (7 / 15 / 12.35) | (16 / 25 / 18.70) | (15 / 28 / 20.14) | 0.12 |
| **COX2** | (6 / 15 / 13.21) | (6 / 16 / 12.82) | (16 / 25 / 19.65) | (15 / 26 / 20.24) | 0.12 |
| **PTC-MR** | (6 / 15 / 12.66) | (7 / 15 / 12.41) | (16 / 25 / 18.72) | (15 / 28 / 20.18) | 0.12 |

Table 7: Graph statistics for each dataset generated for GED.

| Dataset | Query Graphs | | Corpus Graphs | | $\mathbb{E}\left[\frac{|y=1|}{|y=0|}\right]$ |
|---|---|---|---|---|---|
| | **Nodes** | **Edges** | **Nodes** | **Edges** | Label |
| | (min / max / avg) | (min / max / avg) | (min / max / avg) | (min / max / avg) | Ratio |
| **PTC-FR** | (9 / 14 / 11.14) | (8 / 16 / 12.25) | (6 / 20 / 14.66) | (5 / 24 / 15.77) | 0.07 |
| **PTC-FM** | (9 / 14 / 11.09) | (8 / 15 / 12.08) | (6 / 20 / 14.64) | (5 / 24 / 15.73) | 0.07 |
| **COX2** | (9 / 15 / 11.61) | (8 / 17 / 12.90) | (7 / 20 / 15.48) | (6 / 20 / 15.79) | 0.04 |
| **PTC-MR** | (9 / 14 / 10.90) | (8 / 15 / 11.71) | (6 / 20 / 14.67) | (5 / 24 / 15.80) | 0.08 |

## G.2   EMBEDDING MODEL ARCHITECTURE

To supervise retrieval with transport-based distances, we train a neural scoring model composed of a GNN encoder and a Gumbel-Sinkhorn aligner, optimized using pairwise ranking loss (Roy et al., 2022; Jain et al., 2024). Here, $\mathrm{init}_\theta$ is an LRL implemented as a single-layer MLP that maps node features to a 10-dimensional embedding space. $\mathrm{msg}_\theta$ is a message passing block consisting of two linear message functions (forward and reverse), each mapping concatenated node-edge features to a 20-dimensional hidden state, followed by a GRU with hidden size 10 to aggregate incoming messages. $\mathrm{upd}_\theta$ is a two-layer aggregation MLP: the first layer expands the node embedding to 20 dimensions, and the second reduces it back to 10 dimensions to produce the final node representation. To compute the permutation matrix $\boldsymbol{P}$, we solve a linear assignment problem via 10 Sinkhorn iterations at a temperature of 0.1.

Separate models are trained for each supervision type—Subgraph Matching (SM) and Graph Edit Distance (GED)—based on their respective distance formulations using Eq. (1). The model is trained

to assign lower distance scores to relevant corpus graphs compared to irrelevant ones, using the following hinge-based loss:

$$\sum_{q} \sum_{\substack{c:\text{rel}(G_c, G_q)=1 \\ c':\text{rel}(G_{c'}, G_q)=0}} [\Delta(G_c, G_q) - \Delta(G_{c'}, G_q) + \gamma]_+,$$

where $\gamma \in \{0.1, 0.5\}$ is a fixed margin, and $\Delta(\cdot, \cdot)$ is the transport-based distance (Eq. (1)). We set the node embedding dimensionality to $D = 10$ in all experiments.

### G.3 FOURIER-MAP AND HASHCODE TRAINING

We adopt the training framework proposed by Roy et al. (2023) to improve the quality of Fourier-based representations and optimize the hashcodes derived from them. Specifically, we apply two neural networks $\Psi_q$ and $\Psi_c$ that take as input the Fourier representations $\widehat{\boldsymbol{T}}_{q,d}$ and $\widehat{\boldsymbol{T}}_{c,d}$ of query and corpus graphs respectively, and output transformed feature vectors:

$$\boldsymbol{z}_q = \Psi_q(\widehat{\boldsymbol{T}}_{q,d}), \qquad \boldsymbol{z}_c = \Psi_c(\widehat{\boldsymbol{T}}_{c,d}). \tag{318}$$

These transformed vectors are trained using a binary cross-entropy loss that promotes high cosine similarity between relevant query-corpus pairs:

$$\min_{\phi_q, \phi_c} \sum_{(G_q, G_c)} -\text{rel}(G_c, G_q) \log(1 + \cos(\boldsymbol{z}_q, \boldsymbol{z}_c)) - (1 - \text{rel}(G_c, G_q)) \log(1 - \cos(\boldsymbol{z}_q, \boldsymbol{z}_c)). \tag{319}$$

To generate binary hashcodes from the transformed fourier feature vectors, we use a learned projection matrix $\boldsymbol{W} \in \mathbb{R}^{\dim_h \times \dim_T}$ and apply the random hyperplane method:

$$f^{(d)}(G_q) = \text{sign}(\boldsymbol{W}\boldsymbol{z}_q), \qquad h^{(d)}(G_c) = \text{sign}(\boldsymbol{W}\boldsymbol{z}_c). \tag{320}$$

for each $d \in [D] = [10]$. In practice $\dim_T = 10, \dim_h = 64$. We set the number of $\omega$ samples $M = 10$. We use the frequency cutoff $\lambda$ in the low pass filter as 100. During training, we use $\tanh(\boldsymbol{W}\boldsymbol{z})$ as a differentiable approximation to $\text{sign}(\boldsymbol{W}\boldsymbol{z})$, and optimize $\boldsymbol{W}$ using the following composite loss:

$$\mathcal{L}_{\text{hash}} = \lambda_1 \Delta_1 + \lambda_2 \mu_2 + \lambda_3 \mu_3, \tag{321}$$

where:

- $\Delta_1$: **Collision Minimizer** — Encourages higher hashcode overlap between $G_q$ and its most relevant corpus graphs compared to irrelevant ones.
- $\Delta_2$: **Fence-Sitting Penalty** — Penalizes intermediate values of $\tanh(\boldsymbol{W}\boldsymbol{z})$ to enforce hash bits near $\pm 1$.
- $\Delta_3$: **Bit Balance** — Promotes equal usage of $+1$ and $-1$ bits across all corpus hashcodes.

We use the default hyperparameters and network configurations proposed in FourierHashNet (Roy et al., 2023) for $\Psi_q, \Psi_c$, and the loss weights $\mu_i$.

This training process improves both retrieval relevance and the discriminability of learned hashcodes. Algorithm 1 and 2 summarize the index construction and query retrieval procedures based on these learned hashcodes.

### G.4 BASELINES

We compare GRAPHHASH against a range of methods that fall into three broad categories: LSH-based methods operating on single-vector graph embeddings, inverted index-based multi-vector retrieval using FAISS, and graph-based ANN using DiskANN. We also include a naive random sampling baseline for reference.

**Hyperplane based hashing** These methods rely on locality-sensitive hashing (LSH) applied to a single-vector embedding for each graph, typically obtained via mean pooling over node representations.

- **FourierHashNet** (Roy et al., 2023): A learned LSH scheme that approximates hinge-based dominance distances through Fourier transformation. It encodes asymmetric containment-style similarities in a form suitable for efficient hash-based retrieval using random hyperplanes in the frequency domain. We use the default hyperparameters and network configurations proposed in FourierHashNet (Roy et al., 2023). Specifically, we use $\omega = 10$ samples for the Fourier features, a trainable Fourier map optimized using the BCE loss with embedding dimension 10, and

hashcodes of length $64$. We train using the loss function defined in Eq. (321), sweeping across all combinations of $\lambda$ and other hyperparameters as described in their original paper. To evaluate efficiency–effectiveness tradeoffs, we vary the number of hash table buckets from $2^1$ to $2^{60}$ during retrieval.

- **Random Hyperplane (RH) Hashing**: A classical LSH method that applies cosine similarity hashing to mean-pooled graph vectors. Since it uses symmetric cosine distance, it does not capture subgraph asymmetry or node-level structure. We train the baseline using the same loss function as in FourierHashNet (Eq. (321)), sweeping over all hyperparameter combinations reported in their work. The hashcode dimension is set to $64$, and we vary the number of selected hyperplanes (i.e., the subset size) from $2^1$ to $2^{60}$ to generate the tradeoff curves.

**Inverted Index (IVF)**   We implement the inverted file index from FAISS (Douze et al., 2024) in a multi-vector setup, where each corpus graph is decomposed into its node embeddings. These are indexed independently, and during retrieval, each query node probes the index. Retrieved nodes are then aggregated by graph ID to form the candidate set. This simulates node-level matching using learned dense vectors.

For the FAISS baseline, we use the IVF-Flat indexing scheme with `nlist` = 128 clusters. The index is built over node-level embeddings extracted from the corpus graphs. Depending on the specified distance metric (`cosine` or `l2`), we use either inner product similarity or Euclidean distance. For cosine similarity, all corpus embeddings are L2-normalized prior to indexing.

**Graph-Based ANN (DiskANN)**   DiskANN (Simhadri et al., 2023) builds compact HNSW-style proximity graphs for approximate nearest neighbor retrieval at scale. In our setting, each node embedding from the corpus is indexed independently, and the query node embeddings probe this graph. Retrieved node hits are aggregated to rank corpus graphs. DiskANN offers scalability and fast retrieval, but operates with symmetric distances (e.g., $L_2$, cosine) which may not align well with asymmetric retrieval objectives.

We employ the StaticMemoryIndex implementation with cosine or Euclidean distance as the retrieval metric. The memory-based index is built using a graph degree of 16, build-time complexity of 32, and a search-time initial complexity of $2^{21}$. We disable product quantization (PQ) and OPQ refinements by setting use_pq_build=False and use_opq=False, respectively, opting for full-precision vectors. During index construction, we set alpha=1.2 and filter_complexity=32, with multi-threading enabled using 16 threads. We vary the top-$K$ parameter during querying to generate the efficiency–accuracy tradeoff plots.

**Random Sampling**   This baseline selects a fixed number of graphs uniformly at random from the corpus, without using any learned embeddings or indexing structure. It serves as a lower-bound reference to contextualize retrieval performance. Here, we simulate retrieval by uniformly sampling a fixed number of corpus items for each query. We sweep over the number of retrieved items using the set: $\{10, 100, 1000, 2000\} \cup \{5000, 10000, \ldots, 95000\}$, to generate efficiency-accuracy tradeoff curves.

### G.5    EVALUATION METRICS

**MAP**   To assess the trade-off between retrieval accuracy and candidate set size, we compute the Mean Average Precision (MAP). For a query graph $G_q \in \mathcal{Q}$, let $\mathcal{C}_{q\oplus} \subseteq \mathcal{C}$ denote the set of relevant corpus graphs. Given a retrieved ranking $\Pi_q$ over retrieved candidate set $\mathcal{R}_q$, the average precision (AP) is computed as:

$$\text{AP}(G_q) = \frac{1}{|\mathcal{C}_{q\oplus}|} \sum_{r=1}^{|\pi_q|} \text{Prec@}r \cdot \mathbb{I}[\Pi_q(r) \in \mathcal{C}_{q\oplus}],$$

where $\text{Prec@}r$ is the precision at rank $r$, and $\mathbb{I}[\cdot]$ is the indicator function. We compute MAP by averaging AP across all test queries in $\mathcal{Q}_{\text{test}}$:

$$\text{MAP} = \frac{1}{|\mathcal{Q}_{\text{test}}|} \sum_{G_q \in \mathcal{Q}_{\text{test}}} \text{AP}(G_q).$$

This formulation penalizes high precision with low recall, ensuring models are rewarded only when most number of relevant items are retrieved with high retrieval accuracy.

**AUC**   To summarize the trade-off between accuracy and candidate set size, we convert the MAP vs. candidate set size curve into a single scalar metric by computing the area under the trade-curve.

We normalize the candidate set size by the total corpus size $|\mathcal{C}|$, and numerically integrate the MAP values over the normalized x-axis.

**Normalized Discounted Cumulative Gain (NDCG)**     We also report NDCG to evaluate the quality of ranked lists. For each query $G_q$, let $\text{rel}_q(r) \in \{0, 1\}$ denote the relevance label of the item ranked at position $r$ in $\Pi_q$. The DCG at rank $k$ is given by:

$$\text{DCG@}k = \sum_{r=1}^{k} \frac{2^{\text{rel}_q(r)} - 1}{\log_2(r+1)},$$

and the corresponding ideal DCG (IDCG) is computed from a perfect ranking. The NDCG is then:

$$\text{NDCG@}k = \frac{\text{DCG@}k}{\text{IDCG@}k}.$$

We average NDCG over all test queries to obtain a corpus-level evaluation. This metric does not penalize high precision with low recall. We set $k = 1000$.

### G.6    HARDWARE AND LICENSES

All experiments were run on a local NAS server configured with seven NVIDIA RTX A6000 GPUs (48GB each), a 96-core processor, and 20TB of storage, operating under Debian 6.1. All model components, including GNN encoders and hash function training, were executed on GPU memory without resource bottlenecks.

Regarding licensing, GMN (Li et al., 2019) is distributed under the MIT license. The implementations of Isonet (Roy et al., 2022) and FourierHashNet (Roy et al., 2023) are open source and have been cited appropriately in our work. Our full codebase and datasets will be released for public use upon publication.

# H ADDITIONAL EXPERIMENTS

We present supplementary experimental results to support the findings in the main paper. These include validations of embedding exchangeability on additional datasets and evaluation of retrieval performance under alternate metrics and supervision settings. Our goal is to assess whether the trends observed in the main experiments persist across diverse configurations.

## H.1 ADDITIONAL EXCHANGEABILITY RESULTS

The following experiments reuse the same setup as before: 5,000 GNNs are trained independently on a subset of 1,024 query-corpus graph pairs, each with $D = 10$ embedding dimensions, and trained for 20 epochs using a pairwise ranking loss. For a fixed node in one corpus graph, we collect the scalar embedding values across dimensions $d \in [D]$ from all models.

**Covariance of Node embeddings**    Another consequence of exchangeability is the symmetry of higher order moments of the embedding. Specifically, we expect the covariance between two dimensions to remain constant across all pairs of dimensions, which is a stronger demonstration of symmetry in the joint distribution.

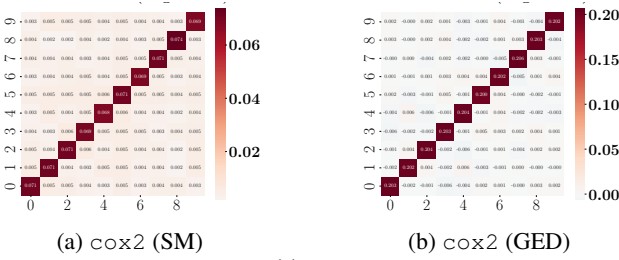

(a) cox2 (SM)        (b) cox2 (GED)

Figure 8: Sample covariance matrix for the $\boldsymbol{X}^{(c)}[v, d]$ for the highlighted nodes in Figures 1,9. The figure shows that the off-diagonal covariances are roughly, which strongly indicates that the coupling between dimensions is symmetric.

Figure 8 shows the covariance matrices for two nodes from different graphs. The $[i, j]^t h$ entry of each matrix matrix represents the estimate for $\mathrm{Cov}(\boldsymbol{X}^{(c)}[v, i], \boldsymbol{X}^{(c)}[v, j])$. We observe that all the off diagonal elements are close to one another, and similarly, all diagonal elements too are close to one another, which indicates that there is symmetry in the coupling between dimensions.

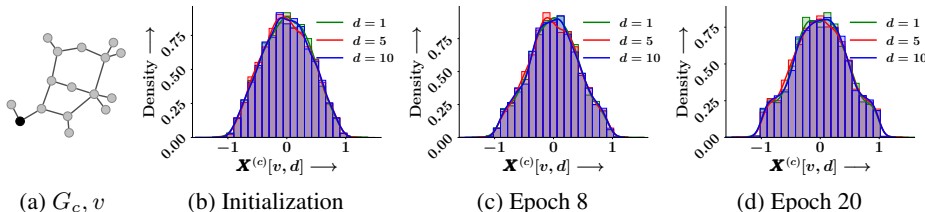

(a) $G_c, v$      (b) Initialization      (c) Epoch 8      (d) Epoch 20

Figure 9: Empirical probability density of $\boldsymbol{X}^{(c)}[v, d]$ for the highlighted node $v$ in the example corpus graph $G_c$ in ptc-fr, obtained using 5,000 independently trained instances of the GNN model under GED-based supervision. Panels (b)–(d) show the density of $\boldsymbol{X}^{(c)}[v, d]$ at initialization and at intermediate stages of training. The observed similarity of distributions across embedding dimensions reaffirms the exchangeability result (Theorem 5) in a different dataset and task setting.

**Marginal distributions on a different dataset**    In Section 5.1, we validated the exchangeability of embedding dimensions by examining the marginal distributions of node embeddings across dimensions, under repeated training runs. Here, we present an additional experiment on a different dataset (PTC-FR) and a different supervision signal (GED with asymmetric costs), to confirm the generality of our claims. Figure 9 shows the distribution of $\boldsymbol{X}^{(c)}[v, d]$ for three representative dimensions ($d = 1, 5, 10$) at three points during training. Similar to the findings on cox2(main paper), the distributions remain near-identical across dimensions and throughout training. This supports the robustness of Theorem 5, even under varied datasets and training objectives.

**Remark.**    For the distribution plots of node embeddings (Figure 1 and Figure 9), we use histograms with 25 bins and apply kernel density estimation for smoothing. These visualizations are generated using the built-in functionality of the seaborn library.

## H.2 Further Evaluation of GraphHash's Retrieval Performance

In the main paper (Section 5.2), we evaluated GRAPHHASH under two supervision signals—Subgraph Matching (SM) and asymmetric GED—using conservative MAP as the primary evaluation metric. Here, we extend that analysis along two axes.

First, we report additional results on a more commonly used GED variant, where both insertion and deletion costs are set to $e_\oplus = e_\ominus = 1$. This equal-cost GED setting alters the notion of relevance and allows us to assess the generality of our approach under a different supervision signal.

Second, we evaluate retrieval performance using NDCG, a position-sensitive ranking metric that complements MAP. These additional results evaluate whether the trends observed in the main paper persist under both metric and supervision signal variations.

### H.2.1 MAP on Equal-Cost GED

In the main paper, we evaluated retrieval performance under asymmetric GED costs ($e_\oplus = 1, e_\ominus = 2$). Here, we assess whether the key trends persist under the equal-cost variant where $e_\oplus = e_\ominus = 1$, a widely used formulation in the literature.

Figure 10 shows the MAP vs. retrieved graphs trade-off curves for all baselines under equal-cost GED supervision. We summarize our observations below:

1. **GRAPHHASH and FourierHashNet remain the strongest performers across all datasets.** Even under equal-cost supervision, both methods consistently outperform other baselines in MAP across retrieval budgets.
2. **FourierHashNet shows marginal improvement in this regime**, particularly on `ptc-fr`, where it slightly surpasses GRAPHHASH, and on `cox2` and `ptc-mr`, where its MAP approaches that of GRAPHHASH at lower candidate counts. However, FourierHashNet often fails to span the full selectivity spectrum, unlike GRAPHHASH, which yields a smoother and more complete accuracy-efficiency trade-off.
3. **RH Hashing remains unstable.** While it occasionally matches GRAPHHASH on `cox2` and `ptc-mr`, its high variance limits its practical utility.
4. **DiskANN, IVF, and Random sampling continue to underperform.** As in the asymmetric setting, these methods yield substantially lower MAP, highlighting the advantage of trainable indexing strategies like GRAPHHASH and FourierHashNet.

These trends are consistent with our findings from the main paper and further validate the generality of GRAPHHASH across different supervision regimes.

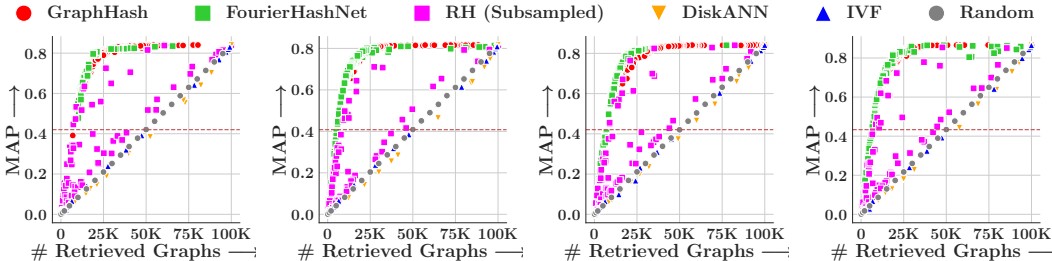

(a) `ptc-fm` (Eq. cost GED) (b) `cox2` (Eq. cost GED) (c) `ptc-fr` (Eq. cost GED)(d) `ptc-mr` (Eq. cost GED)

Figure 10: Trade-off between mean average precision (MAP) and number of retrieved graphs, for all the methods, *viz.*, GRAPHHASH, FourierHashNet (Roy et al., 2023), Random Hyperplane (RH) (Charikar, 2002; Indyk et al., 1997), IVF (Douze et al., 2024),DiskANN (Simhadri et al., 2023) and Random, across all datasets. Retrieval based on Equal cost GED ($e_\bullet = 1$). Horizontal red line denotes 50% of exhaustive MAP. Our method shows a better trade-off than others in majority of the cases.

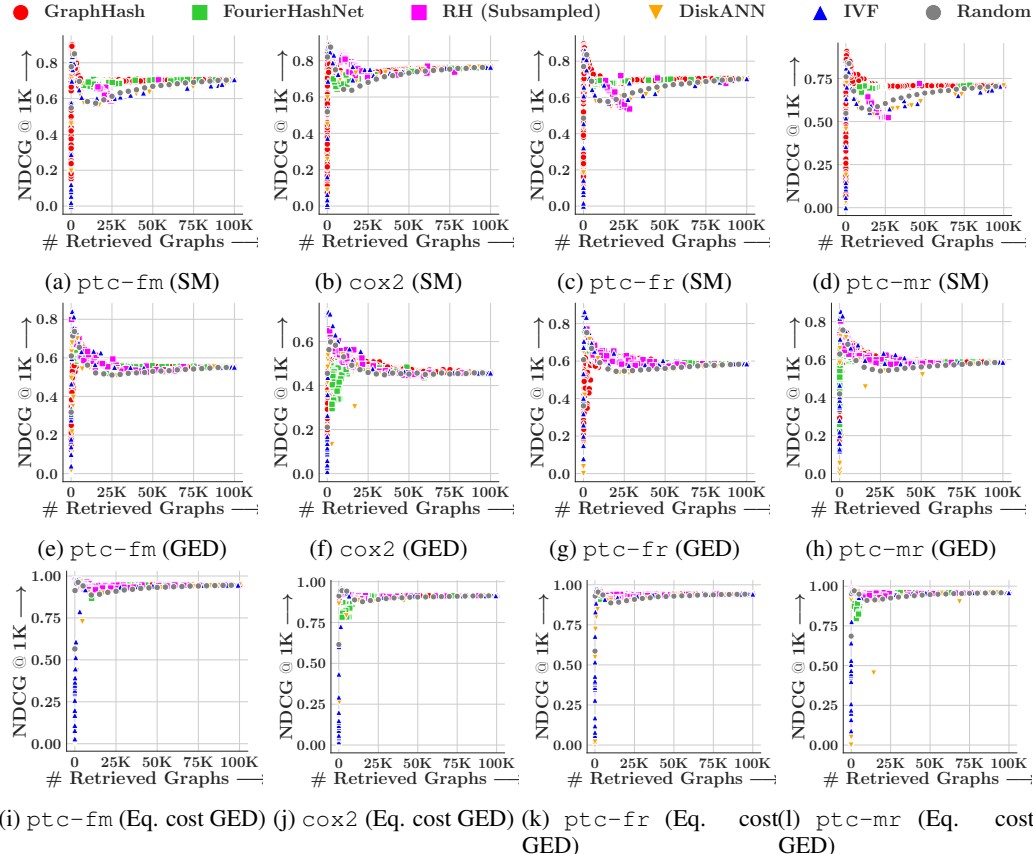

Figure 11: Trade-off between NDCG at top 1000 and number of retrieved graphs, for all the methods, *viz.*, GRAPHHASH, FourierHashNet (Roy et al., 2023), Random Hyperplane (RH) (Charikar, 2002; Indyk et al., 1997), IVF (Douze et al., 2024),DiskANN (Simhadri et al., 2023) and Random, across all datasets. Top row: Retrieval based on Subgraph Matching (SM); Middle row: Retrieval based on GED; Bottom row: Retrieval based on Equal cost GED ($e_\bullet = 1$). Our method shows a better trade-off than others in majority of the cases.

### H.2.2 EVALUATION USING NDCG

To complement our MAP-based evaluation, we assess ranking quality using NDCG across all datasets and relevance definitions. Figure 11 reports results for Subgraph Matching, unequal-cost GED, and equal-cost GED.

1. **GRAPHHASH consistently achieves the highest or near-highest NDCG across all datasets and relevance settings.** This confirms that GRAPHHASH not only retrieves more relevant graphs overall, but also ranks them effectively near the top of the candidate list.
2. **Relative gains over baselines are smaller compared to MAP.** While GRAPHHASH leads in most cases, RH hashing performs competitively under unequal-cost GED, and nearly all baselines exhibit similar performance under equal-cost GED. This suggests that some methods manage to prioritize a few relevant graphs early, even if overall recall is limited.
3. **DiskANN and IVF show competitive NDCG despite low MAP.** These methods often retrieve a handful of highly relevant graphs early in the ranking, which boosts NDCG but fails to capture the full relevant set.
4. **Random sampling yields flat and significantly lower NDCG.** This reinforces the importance of structured indexing and learning-based methods for meaningful ranked retrieval.

Overall, NDCG results validate our MAP findings and demonstrate that GRAPHHASH excels at not just retrieving relevant graphs but also ranking them effectively within large candidate pools.

### H.2.3 CLARIFICATION ON RH (SUBSAMPLED)

In Figure 4 of the main paper and Figures 10 and 11 in the appendix, we display retrieval performance as scatter plots, as described in Section 5.2. The label "RH (Subsampled)" in these figures refers to a subsampling of the full set of trade-off points obtained for the Random Hyperplane (RH) method. This subsampling was performed solely to prevent visual clutter and improve readability of the main figures.

To ensure full transparency, Figures 12 and 13 present the complete set of RH performance points generated via a comprehensive hyperparameter sweep. Specifically, we vary the hash table size and the loss weights in Eq. (321), following the experimental protocol recommended in the FourierHash-Net (Roy et al., 2023). These figures show retrieval performance for all datasets across all three supervision signals (Subgraph Matching, GED, and Equal-cost GED), evaluated using both MAP and NDCG at top 1000.

We make the following observations:

1. **Consistency with main trends:** Even with the full set of hyperparameter configurations, the qualitative findings from the earlier results remain consistent—GRAPHHASH outperforms RH on both MAP and NDCG for Subgraph Matching (SM), and also on MAP for GED. RH achieves comparable performance only on NDCG for GED, but remains less reliable overall.
2. **Pronounced variability:** With more points shown, the performance of RH appears highly scattered, especially at fixed retrieval sizes. This reinforces its sensitivity to hyperparameter selection.
3. **Practical tuning challenge:** The high variance observed for RH across sweeps suggests that achieving consistently strong performance would require extensive tuning, which may not be practical in real-world deployments.

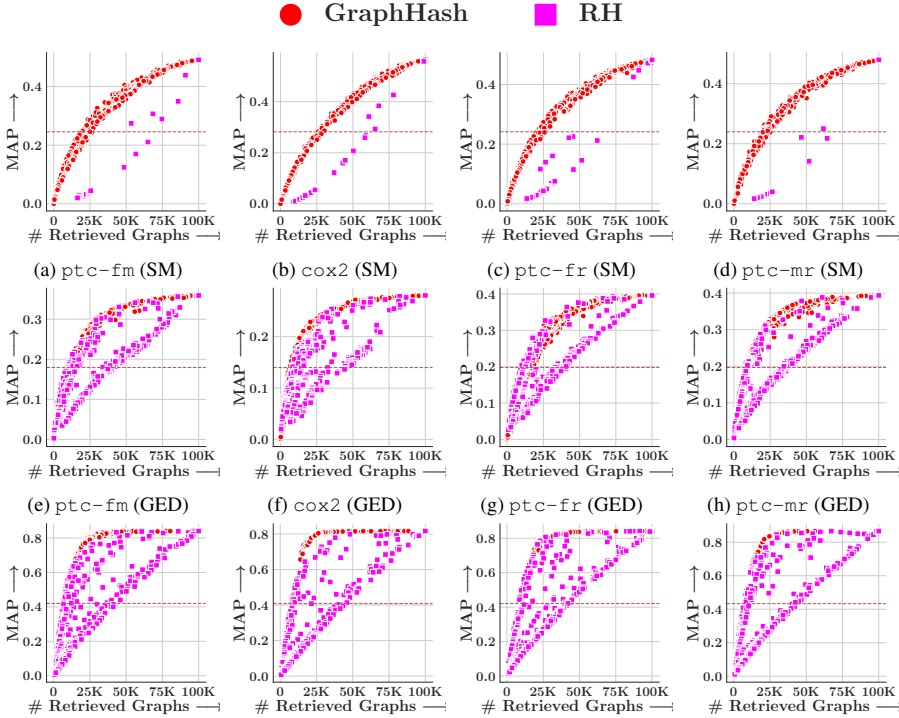

(i) `ptc-fm` (Eq. cost GED) (j) `cox2` (Eq. cost GED) (k) `ptc-fr` (Eq. cost GED)(l) `ptc-mr` (Eq. cost GED)

Figure 12: Trade-off between MAP and number of retrieved graphs taking all points. Top row: Subgraph Matching (SM); Middle row: GED; Bottom row: Equal cost GED ($e_\bullet = 1$). Horizontal red line denotes 50% of exhaustive MAP.

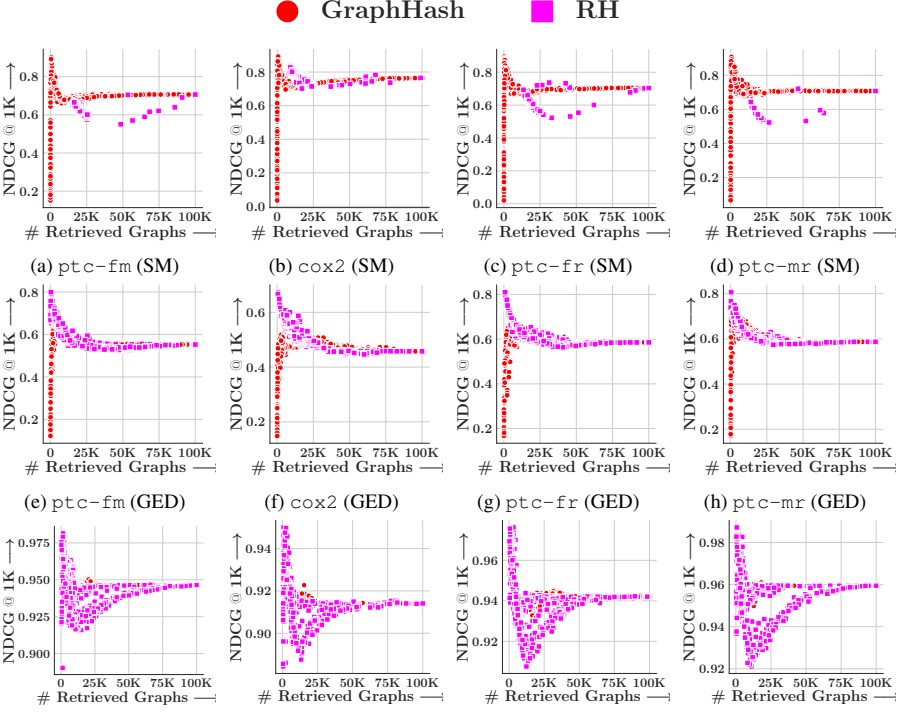

(i) `ptc-fm` (Eq. cost GED) (j) `cox2` (Eq. cost GED) (k) `ptc-fr` (Eq. cost GED)(l) `ptc-mr` (Eq. cost GED)

Figure 13: Trade-off between NDCG at 1000 and number of retrieved graphs taking all points. Top row: Subgraph Matching (SM); Middle row: GED; Bottom row: Equal cost GED ($e_\bullet = 1$).

### H.2.4 EVALUATION ON LARGER GRAPHS

We synthetically generate larger versions of `cox2` and `ptc-fr` by combining graphs in the original datasets for the Subgraph Matching task. The gold relevance labels are approximated as the set of graphs made up of relevant items of the original data. We generate $10^4$ corpus items for either dataset, and plot the tradeoff curves as in Figure 4. We observe that GRAPHHASH performs better than the baselines in high accuracy regime

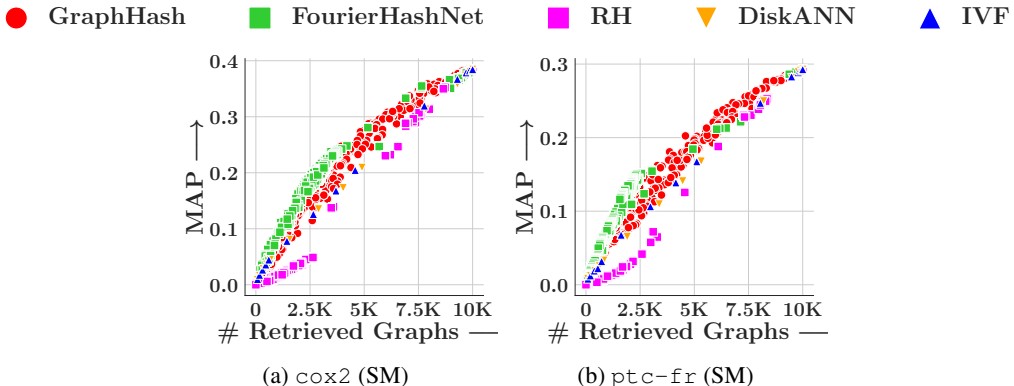

(a) cox2 (SM)          (b) ptc-fr (SM)

Figure 14: Trade-off between mean average precision (MAP) and number of retrieved graphs, for GRAPHHASH, FourierHashNet (Roy et al., 2023), Random Hyperplane (RH) (Charikar, 2002; Indyk et al., 1997), IVF (Douze et al., 2024) and DiskANN (Simhadri et al., 2023), across two datasets with synthetically generated large graphs under Subgraph Matching supervision.

### H.2.5 EVALUATION ON LARGER CORPUS

In this set of experiments, we evaluate GRAPHHASH on a larger corpus of $1M$ items.

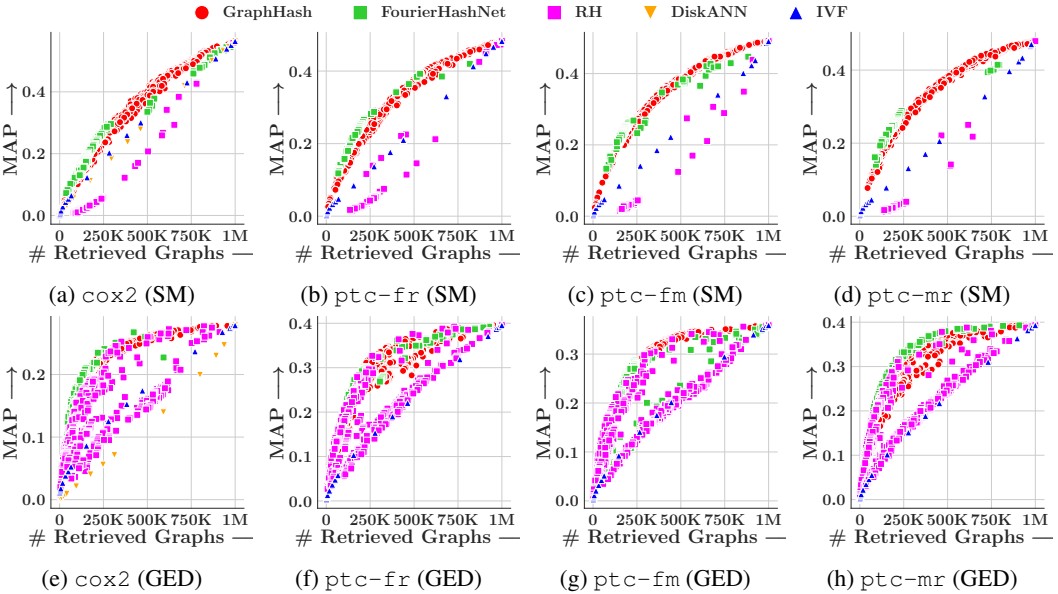

(a) cox2 (SM)          (b) ptc-fr (SM)          (c) ptc-fm (SM)          (d) ptc-mr (SM)

(e) cox2 (GED)          (f) ptc-fr (GED)          (g) ptc-fm (GED)          (h) ptc-mr (GED)

Figure 15: Trade-off between mean average precision (MAP) and number of retrieved graphs, for GRAPHHASH, FourierHashNet (Roy et al., 2023), Random Hyperplane (RH) (Charikar, 2002; Indyk et al., 1997), IVF (Douze et al., 2024), and DiskANN (Simhadri et al., 2023) across all datasets for a million sized corpus. Top row: Retrieval based on Subgraph Matching (SM); Bottom row: Retrieval based on GED

### H.2.6 ABLATION STUDIES

**Ablation on** $\dim_h$   Here, we present the trade-off curves for MAP versus number of retrieved graphs for each choice of $\dim_h$, the size of the hashcode. The below tradeoff has been summarised to Figure 5 in the main paper. Owing to the larger number of values of $\dim_h$, we use a colorscale for the scatterplot.

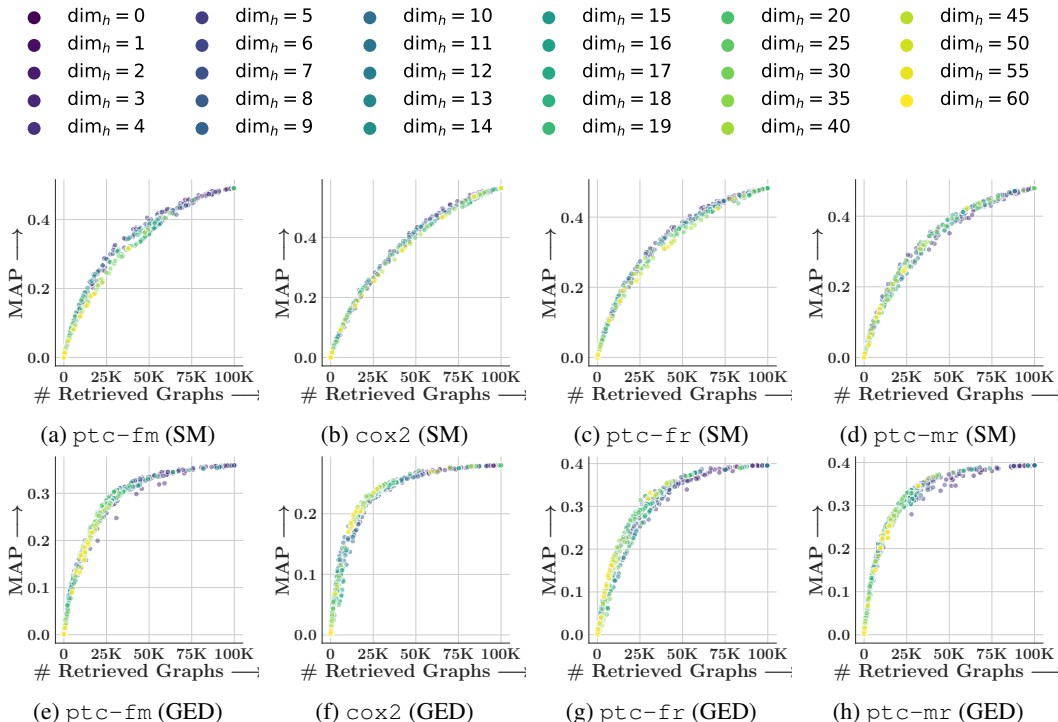

Figure 16: Trade-off between mean average precision (MAP) and number of retrieved graphs, for GRAPHHASH for different values of the hashcode size $\dim_h$

**Ablation with** $D$   Here, we perform experiments ablating the embedding dimension of the netowrk, and the number of hash tables used.

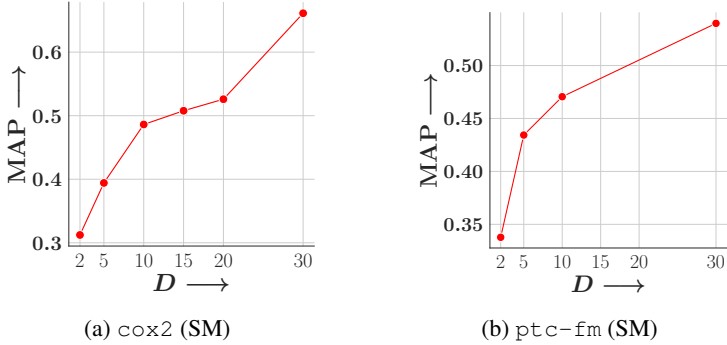

Figure 17: The exhaustive MAP achieved by an embedding model trained on the node aligned loss with respect to the embedding dimension of the model.

We see that MAP increases monotonically with $D$, as is expected as the higher dimension allows for richer feature representation without hitting the bottleneck in training requirements.

**Ablation with number of hash tables**   We also perform ablation over the number of hash tables. Note that for GRAPHHASH the number of hash tables corresponds to the number of dimensions of

the embedding utilised, which implies a monotone behavior in the performance. We seek to find if the accuracy losses are comparatively low, which could help cut time and memory.

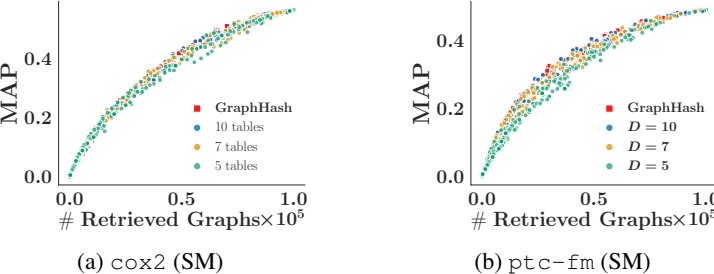

(a) cox2 (SM)  (b) ptc-fm (SM)

Figure 18: Trade of plot showing MAP vs the number of retrieved corpus items for different variants of GRAPHHASH that uses a different number of hash tables for retrieving results.

We observe that the drop in performance is not too significant from 10 to 7, although it is noticeable for 5. Ultimately, this vindicates our decision to use all 10 hash tables

**Stability of random hyperplane seeding**  Next, we evaluate the stability of the random hyperplane hashing scheme over multiple random seeds. In this setting, we set 10 different random seeds for the hyperplanes, keeping the embeddings and fourier maps fixed. We then evaluate the retrieval performance on the best hyperparameters found from GRAPHHASH.

We report the mean and standard devation in AUC over these 10 runs.

| Dataset (Task) | Mean AUC | Std |
|---|---|---|
| ptc-fm (SM) | 0.342685 | 0.006966 |
| cox2 (SM) | 0.369972 | 0.009179 |
| ptc-fm (GED) | 0.289546 | 0.007598 |
| cox2 (GED) | 0.238293 | 0.005878 |

Table 19: Mean and standard deviation of AUC over 10 different random seeds for RH seeding.

We also plot the tradeoff curves for the different random seeds, contrasting their performance with the final version of GRAPHHASH. Each color denotes a different seed.

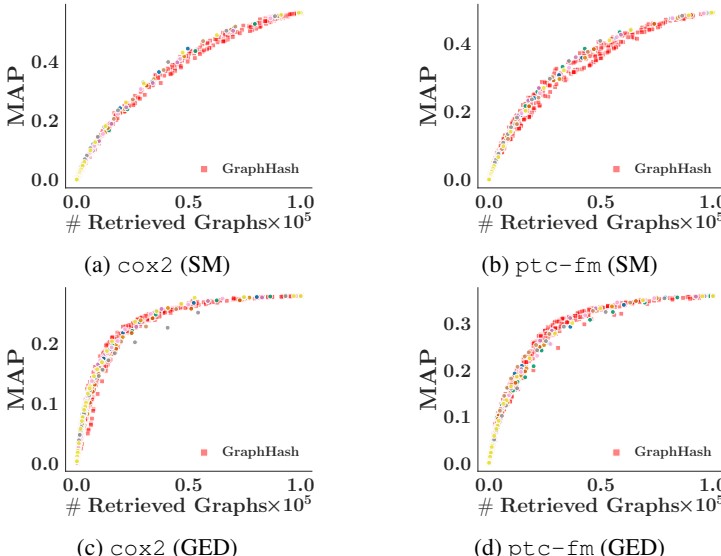

(a) cox2 (SM)  (b) ptc-fm (SM)

(c) cox2 (GED)  (d) ptc-fm (GED)

Figure 20: Tradeoff curves comparing GRAPHHASH (red) with different random seeds for Random Hyperplane hashing across both tasks on cox2 and ptc-fm. Each color denotes a different seed.

We observe that the variation in performance between different seeds is very minimal, as the different values coincide with the tradeoff trajectory of the best performing hyperparameters of GRAPHHASH.

**Stability of fourier map dimension** $\dim_T$    We also ablate over the size of the fourier representation $\dim_T$. In our formulation, we have reparameterized $\dim_T = 4nM$, where $n$ is the size of the graphs. In our experiment we ablate over $M$.

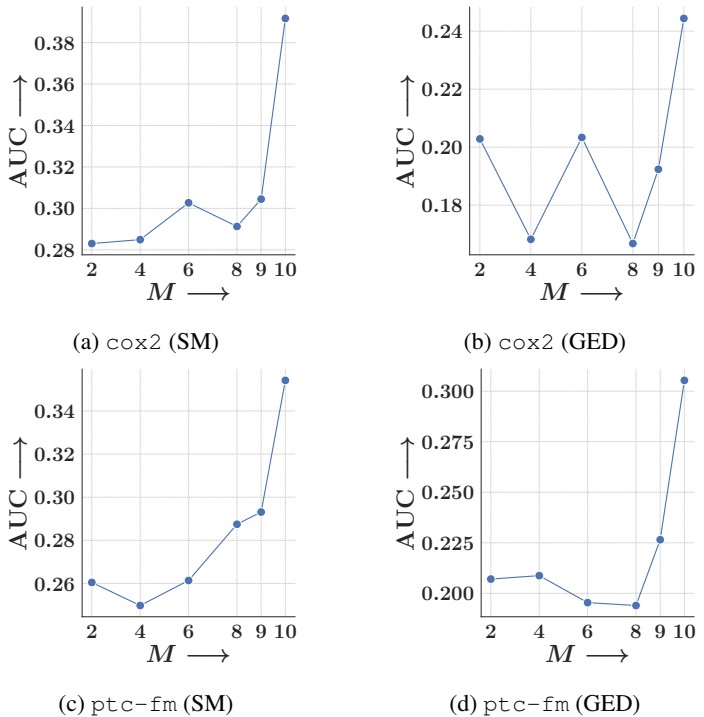

(a) cox2 (SM)    (b) cox2 (GED)

(c) ptc-fm (SM)    (d) ptc-fm (GED)

Figure 21: Comparison of AUC of the MAP vs retrieval ratio curve for different values of the per-dimension-fourier frequencies $M$, across two datasets on both tasks.

We compare the AUC generated by the tradeoff curve generated for each value of $M$. We observe a sharp decline in the performance when going down from $10$ fourier frequencies per dimension.

### H.2.7 COMPARISON OF sim AND $\text{sim}_d$

**Direct comparison of** sim **vs.** $\text{sim}_d$   We compare the quality of the approximation by plotting the scatter plots of the scores obtained by sim and $\text{sim}_d$ for all the datasets and tasks. Specifically, we compare the mean 1D score, *i.e.* $\frac{1}{D}\sum_{i=1}^{D}\text{sim}_d^{(i)}$ against the true score sim scaled by $\frac{1}{D}$. For each $G_c, G_q$ pair in the test set, we compute these two values and plot them.

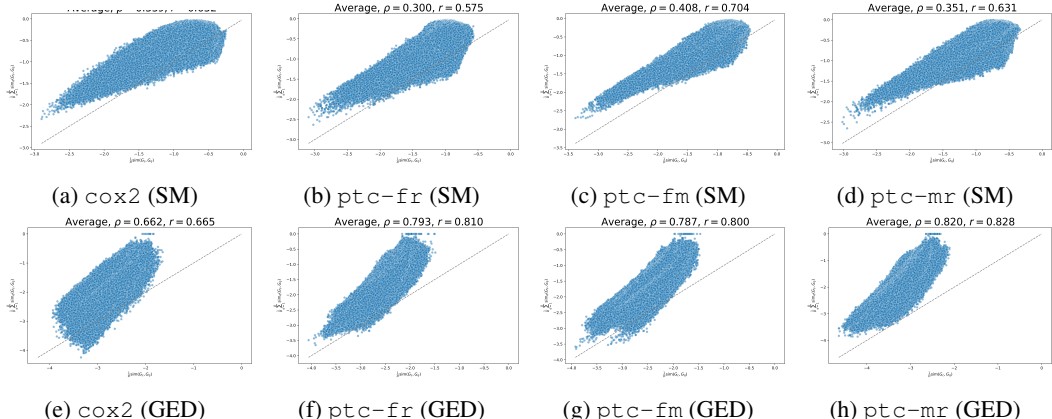

| (a) cox2 (SM) | (b) ptc-fr (SM) | (c) ptc-fm (SM) | (d) ptc-mr (SM) |
|---|---|---|---|

| (e) cox2 (GED) | (f) ptc-fr (GED) | (g) ptc-fm (GED) | (h) ptc-mr (GED) |
|---|---|---|---|

Figure 22: Scatter plots comparing the mean 1D similarity scores (y-axis) with the true similarity scores (x-axis) computed with sinkhorn iterations, for the (top) Subgraph Matching and (bottom) Graph Edit Distance task across different datasets.

**Decay of** $|\text{sim}_d(G_c, G_q) - \text{sim}(G_c, G_q)|$ **with increasing** $D$   Next, we empirically validate the concentration result from Proposition 7 by plotting the average absolute error $|\text{sim}_d(G_c, G_q) - \text{sim}(G_c, G_q)|$ over all pairs $(G_c, G_q)$ in the test set as a function of $D$. We note that the deviation decreases with increasing $D$, confirming the result.

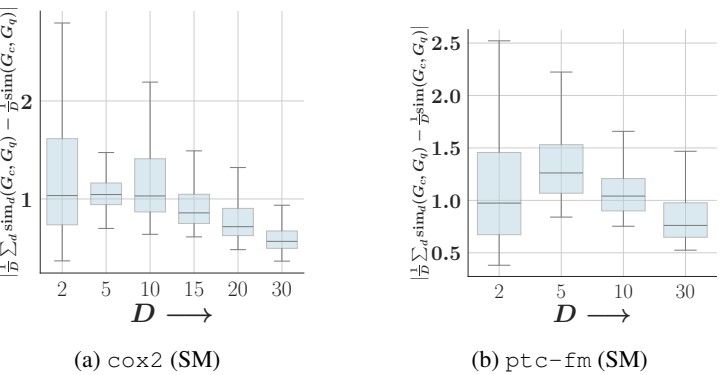

| (a) cox2 (SM) | (b) ptc-fm (SM) |
|---|---|

Figure 23: Boxplot of average absolute error $|\frac{1}{D}\sum_d \text{sim}_d(G_c, G_q) - \text{sim}(G_c, G_q)|$ as a function of $D$ for the Subgraph Matching task on different datasets.

### H.2.8 EVALUATION OF LSH METHODS UNDER ALIGNED SCORING FUNCTIONS

To ensure a fair comparison across LSH-based retrieval strategies, we evaluate each method using graph embeddings specifically trained to align with its intended scoring function. That is, while GRAPHHASH is evaluated under transport-based supervision, FourierHashNet and Random Hyperplane (RH) methods are applied on embeddings trained for hinge and cosine-based scoring, respectively.

**GRAPHHASH: Transport-Based Scoring with GNN Embeddings.** For GRAPHHASH, we use node-level embeddings produced by a GNN encoder, trained using a pairwise ranking loss (Eq. (G.2)) based on the transport distance $\Delta(G_c, G_q)$ (Eq. (1)).

For the baselines that require a single-vector representation of graphs, we adopt the GEN architecture from (Li et al., 2019), which aggregates node embeddings into a global graph-level vector via mean pooling.

**FourierHashNet: Hinge Distance over Aggregated Graph Embeddings (GEN + FourierHashNet).** FourierHashNet is designed for asymmetric hinge-based distances over global graph embeddings. We apply it on GEN representations trained using the ranking loss in Eq. (G.2), where $\mathrm{rel}(G_c, G_q) = \|a_q - a_c\|_+$, and $a_q, a_c$ denote the pooled graph embeddings. Here, $[\cdot]_+$ is the ReLU function.

**RH: Cosine Similarity-Based Hashing (GEN + RH).** To align with RH's reliance on cosine similarity, we again use GEN-pooled embeddings and train them with the ranking loss in Eq. (G.2), setting $\mathrm{rel}(G_c, G_q) = -\cos(a_q, a_c)$. This setup ensures that the learned representations are optimized for RH's angle-based locality-sensitive hashing.

**Summary.** Each method is thus benchmarked under conditions it was designed for: transport distance with GRAPHHASH, hinge distance with FourierHashNet, and cosine similarity with RH. This isolates the performance of the retrieval mechanism from mismatches in training objectives or input embeddings.

**Observations.** Figures 24 and 25 present retrieval performance across all datasets and supervision types. Figure 24 reports MAP trade-offs, while Figure 25 reports NDCG. We observe that:

1. **Exhaustive scores reveal superiority of transport-based supervision.** Across all datasets and similarity signals, GRAPHHASH consistently achieves higher exhaustive MAP and NDCG compared to both GEN + FourierHashNet and GEN + RH. This confirms that transport-based supervision captures a more powerful and fine-grained notion of graph relevance.
2. **RH shows significantly reduced variance when used with compatible supervision.** Unlike earlier results where RH was applied to transport-trained embeddings and exhibited high variability (Figure 4), the GEN + RH setup shows much smoother and more stable trade-offs. This emphasizes the importance of matching the embedding training signal to the retrieval method.
3. **FourierHashNet benefits from hinge-compatible embeddings.** When used with GEN-trained embeddings under hinge distance supervision, FourierHashNet exhibits broader coverage of the selectivity spectrum, yielding smoother MAP and NDCG trade-off curves. This again reinforces the value of scoring-function alignment between embedding training and LSH mechanism.
4. **Despite improvements, GRAPHHASH retains overall dominance.** Even though GEN-based variants show improved performance over their misaligned counterparts, they still fall short of GRAPHHASH in nearly all retrieval settings. This underscores the strength of the transport scoring model in both relevance estimation and downstream index quality.

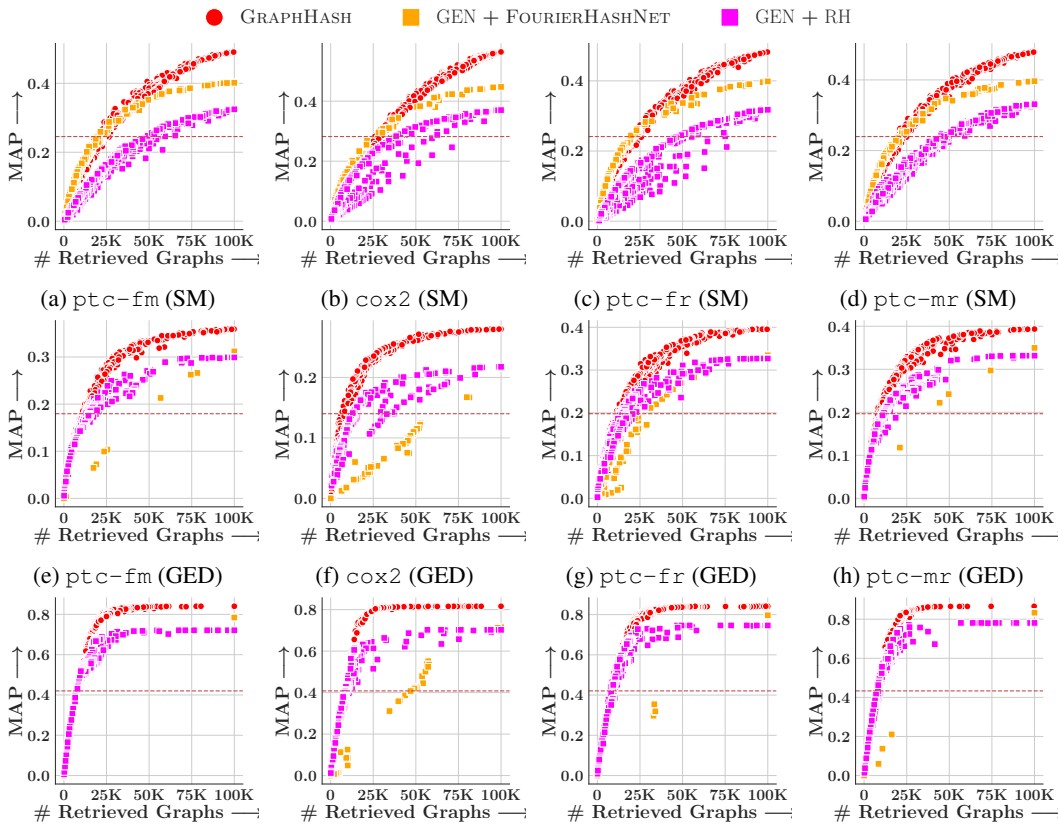

(a) `ptc-fm` (SM)  (b) `cox2` (SM)  (c) `ptc-fr` (SM)  (d) `ptc-mr` (SM)

(e) `ptc-fm` (GED)  (f) `cox2` (GED)  (g) `ptc-fr` (GED)  (h) `ptc-mr` (GED)

(i) `ptc-fm` (Eq. cost GED) (j) `cox2` (Eq. cost GED) (k) `ptc-fr` (Eq. cost GED)(l) `ptc-mr` (Eq. cost GED)

Figure 24: Trade-off between mean average precision (MAP) and number of retrieved graphs, for all the methods, *viz.*, GRAPHHASH, FourierHashNet (Roy et al., 2023) using GEN embeddings, Random Hyperplane (RH) (Charikar, 2002; Indyk et al., 1997) using GEN embeddings, across all datasets. Top row: Retrieval based on Subgraph Matching (SM); Middle row: Retrieval based on GED; Bottom row: Retrieval based on Equal cost GED ($e_\bullet = 1$). Horizontal red line denotes 50% of exhaustive MAP. Our method shows a better trade-off than others in majority of the cases.

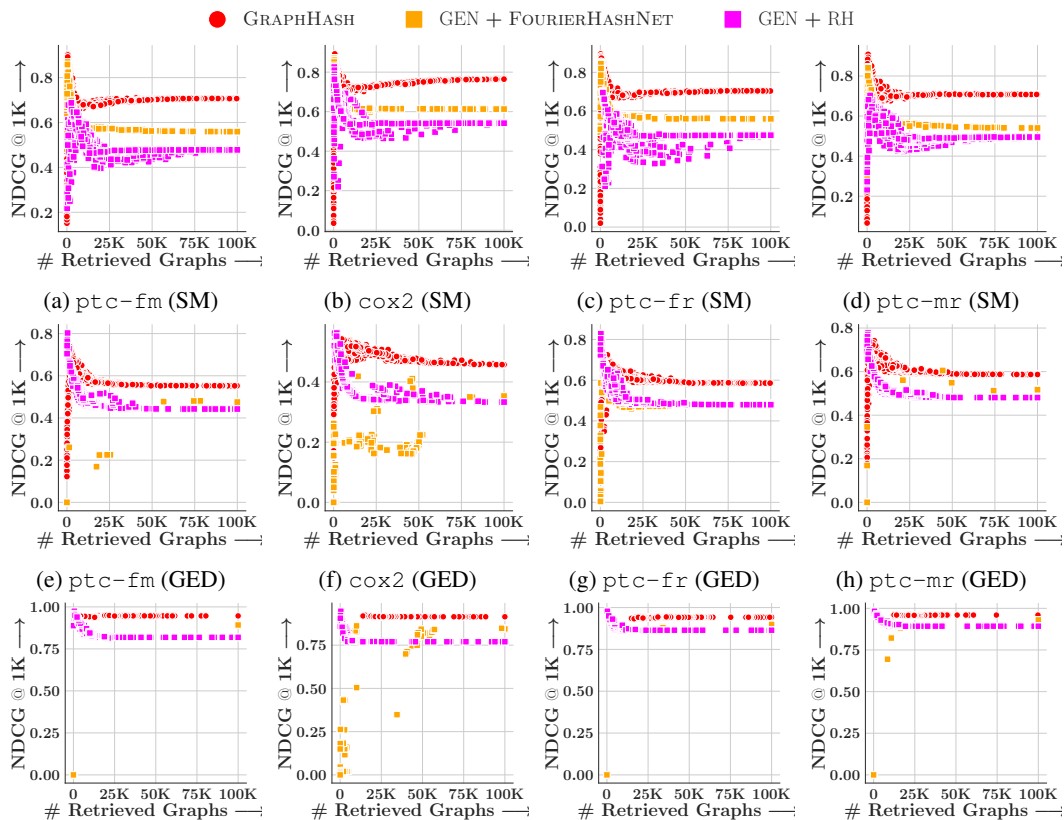

(a) ptc-fm (SM)    (b) cox2 (SM)    (c) ptc-fr (SM)    (d) ptc-mr (SM)

(e) ptc-fm (GED)    (f) cox2 (GED)    (g) ptc-fr (GED)    (h) ptc-mr (GED)

(i) ptc-fm (Eq. cost GED) (j) cox2 (Eq. cost GED) (k) ptc-fr (Eq. cost GED)(l) ptc-mr (Eq. cost GED)

Figure 25: Trade-off between NDCG at top 10000 and number of retrieved graphs, for all the methods, *viz.*, GRAPHHASH, FourierHashNet (Roy et al., 2023) using GEN embeddings, Random Hyperplane (RH) (Charikar, 2002; Indyk et al., 1997) using GEN embeddings, across all datasets. Top row: Retrieval based on Subgraph Matching (SM); Middle row: Retrieval based on GED; Bottom row: Retrieval based on Equal cost GED ($e_\bullet = 1$). Our method shows a better trade-off than others in majority of the cases.

