# OpenReview forum: "Exchangeability of GNN Representations  with Applications to Graph Retrieval"
_ICLR.cc/2026/Conference — ICLR 2026 Oral_

### Official Review · Reviewer_UMLi · 2025-10-24

**Soundness:** 2
**Presentation:** 3
**Contribution:** 3
**Rating:** 6
**Confidence:** 2

**Summary:**

This paper claims to uncover an exchangeability property in GNN embeddings, suggesting that node embedding dimensions are probabilistically symmetric and invariant under permutation. Based on this assumption, the authors approximate transportation-based graph similarities using Euclidean distances between order statistics and build a unified LSH framework supporting multiple graph similarity measures. While the idea is conceptually interesting, the theoretical justification and empirical validation of the claimed exchangeability remain questionable and insufficiently supported.

**Strengths:**

- The writing is good.
- The quite novel angle is graph matching area.
- The performance is superior.

**Weaknesses:**

- The assumptions required to achieve exchangeability are overly restrictive and may not hold in realistic training scenarios.
- The experimental validation of exchangeability is weak.
- The choice of baselines is limited.

**Questions:**

- The introduction states that “the expected embedding matrix $\mathbb{E}[[x(u)]_{u\in V}]$ collapses to a rank-one matrix.” However, no empirical evidence (e.g., covariance analysis or SVD spectrum visualization) is provided to substantiate this claim.
- Real-world graphs typically vary in size. How does the proposed GraphHash framework handle graphs with different numbers of nodes, especially when a query graph contains more nodes than any graph seen during training?
- Do the padding nodes used to equalize graph sizes introduce additional noise or bias that may negatively affect retrieval performance?
- The conditions described in Sec. 3.1 for achieving exchangeability seem too strong. Many standard graph learning losses are not permutation invariant, and the optimizer’s equivariance depends critically on this property of the loss function.
- More SOTA works regarding graph match should be included, such as GNN-PE[1], IsoNet++[2]
- No hyperparameter sensitivity analysis.


- The proposed experiment for validating the exchangeability of embedding dimensions is conceptually flawed and cannot support the stated claim. The authors attempt to verify exchangeability by checking whether the marginal distributions of embedding elements are identical. However, this approach only measures the stability or randomness of parameter initialization and model convergence, not the permutation invariance or exchangeability of embedding dimensions within a single trained model.
In particular:
	- Exchangeability concerns the joint distribution of embedding dimensions within a model, not the marginal consistency of embeddings across independently trained models.
	- The observed similarity (or dissimilarity) of marginal distributions across models provides no evidence for or against permutation invariance in the learned representation space.
	- The experiment conflates stochastic training variation with structural symmetry in the embedding space.
	- No statistical test is performed on the joint dependency structure between embedding dimensions, which is essential for verifying exchangeability.


[1] Ye Y, Lian X, Chen M. Efficient exact subgraph matching via gnn-based path dominance embedding[J]. Proceedings of the VLDB Endowment, 2024, 17(7): 1628-1641.

[2] Ramachandran A, Raj V, Roy I, et al. Iteratively refined early interaction alignment for subgraph matching based graph retrieval[J]. Advances in Neural Information Processing Systems, 2024, 37: 77593-77629.

---

> ### Author Response · Authors · 2025-11-21
> **Response to Reviewer UMLi (1/4)**
>
> We thank the reviewer for their suggestions, which will help improve our paper. We have incorporated these discussions in the paper. Sec 5.1 and Appendix H.1 contain the experiments on exchangeability. Sec 5.2 and Appendix H.2.4 onwards contain the experiments on graph retrieval. Section E.1.6 contains additional discussions about theoretical setup.
>
>
> > The assumptions required to achieve exchangeability are overly restrictive. Many standard graph learning losses are not permutation invariant,
>
> We imposed a few simplifying assumptions only for brevity. In fact, our exchangeability results continue to hold even when these conditions are not explicitly met, including architectures that incorporate more complex operations such as normalization layers.
>
>
> ### Permutation invariance
>
> Even when we consider settings where the loss is not permutation invariant  (for example, a node classification task), exchangeable 'representations' exist within the middle of the network rather than at the end.
>
> We may partition such network into two parts ― the embedding network, and the classifier/prediction head. We may write $\mathbf{X} = \mathrm{NN}(G)$ where we refer to $\mathbf{X}$ as the embeddings and  $\hat{\mathbf{y}} = \mathrm{Clf}(\mathbf{X})$ where $\hat{\mathbf{y}}$ is the prediction label vector across nodes.  We can characterize and prove the exchangeability of $\mathbf{X}$ for this setting.
>
> Let the parameters of the entire network at $t$ timesteps be represented by $\theta   = (\theta  _{\mathrm{NN} },\theta  _{\mathrm{Clf} })$, coresponding to the parameters of either network. Let us also define the permutation inducing transformation as $\Gamma  _{\pi} = \Gamma  _{\mathrm{NN},\pi} \otimes \Gamma  _{\mathrm{Clf}, \pi}$, i.e. $\Gamma  _{\pi} (\theta) = (\Gamma  _{\mathrm{NN}, \pi}(\theta  _{\mathrm{NN}}),\Gamma  _{\mathrm{Clf}, \pi}(\theta  _{\mathrm{Clf}}))$. Given the dataset, we may reparameterise the loss function as $\mathcal{L}(\mathbf{X},\mathrm{Clf})$, or equivalently, $\mathcal{L}(\mathbf{X},\theta  _{\mathrm{Clf}})$.
>
> Given any permutation $\pi$, if there exists $\Gamma  _{\mathrm{NN},\pi},\Gamma  _{\mathrm{Clf},\pi}$ such that
> - $\mathbf{X}\mapsto\mathbf{X}\mathbf{\pi}$ under $\Gamma  _{\mathrm{NN}, \pi}$, and
> - the loss is invariant under $(\pi,\Gamma  _{\mathrm{Clf},\pi})$, i.e. $\mathcal{L}(\mathbf{X},\theta  _\mathrm{Clf}) = \mathcal{L}(\mathbf{X}\mathbf{\pi},\Gamma  _{\mathrm{Clf},\pi}(\theta  _\mathrm{Clf}))$
>
> Under these conditions, exchangeability follows with the same steps - exchangeability at initialisation, equivariance of gradient, equivariance of update step.
>
> To illustrate this, consider a three class classification task with a single layer for both $\mathrm{NN}$ and $\mathrm{Clf}$. Let the input feature be $\mathbf{feat}$. Let us focus on one channel/node of $\mathbf{X}$ denoted as $\mathbf{x} = \mathbf{X}[:,\bullet]$ and $\hat{\mathbf{y}}[\bullet] = y$. We have: $\mathbf{x} = \mathrm{NN}(\mathbf{feat}) = \sigma(\mathbf{feat}\mathbf{\Theta}  _{\mathrm{NN}})$. Hence, we will have:
> $\hat{y} = \mathrm{Softmax}\left([(\mathbf{x} \cdot\mathbf{w}  _1)\, \,(\mathbf{x}\cdot\mathbf{w}  _2)\,(\mathbf{x}\cdot\mathbf{w}  _3)]\right)$.
>
> The transformation $\Gamma  _{\text{NN},\pi}$ can then represented as, $\Theta  _{\mathrm{NN}} \mapsto \Theta  _{\mathrm{NN}}\mathbf{\pi}$ and $[\mathbf{w}  _1, \mathbf{w}  _2, \mathbf{w}  _3] \mapsto [\mathbf{\pi}^\top\mathbf{w}  _1, \mathbf{\pi}^\top\mathbf{w}  _2, \mathbf{\pi}^\top\mathbf{w}  _3]$. Under this transformation $\mathbf{x}\mapsto\mathbf{x}\mathbf{\pi}$ but $\hat{y}$ remains invariant---therefore, the loss is invariant.
>
> Hence, our results remain valid even when the loss itself is not permutation-invariant. This is because such losses may still exhibit invariance under a joint transformation consisting of a permutation of intermediate representations together with a corresponding permutation-induced transformation of the parameters.
>
>
> ### Normalizations
>
> Batch norm, layer norm, etc.  do not break exchanegability condition. If the network without the norm layers can be shown to give exchangeable embeddings, the same will hold for the embeddings for the network with batch norm, layer norm, adaptive optimizers, dropout etc.
>
> We denote a normalization layer as $NL  _{\gamma,\beta}$, where $\gamma$ and $\beta$ are parameters. Such layers allow us to extend permutation inducing transformation $\Gamma  _{\pi}$ to $\Gamma'  _{\pi}$. For simplicity, assume that the normalization layer $NL  _{\gamma,\beta}$ is applied on one layer $\ell$. Suppose, $\theta\to\Gamma  _{\pi} (\theta)$ gives $\mathbf{Z}\to \mathbf{Z} \pi$ in that $\ell$ layer (where $\mathbf{Z}\in \mathbb{R}^{n\times dim  _z}$). Then we can obtain a transformation $\Gamma'  _{\pi}$ such that  $\theta\cup\{\gamma,\beta\}\to\Gamma'  _{\pi} (\theta\cup\{\gamma,\beta\})$ will also give  $\mathbf{Z}\to \mathbf{Z} \pi$.

---

> > ### Author Response · Authors · 2025-11-21
> > **Response to Reviewer UMLi (2/4)**
> >
> > (Contd. from above: The assumptions required to achieve exchangeability are overly restrictive.)
> >
> >
> >
> > Consider batch norm for example. Let the batch of inputs be $G  _1,G  _2,\cdots,G  _B$ and a single batch norm layer, with the corresponding inputs $\mathbf{Y}  _1,\mathbf{Y}  _2,\cdots,\mathbf{Y}  _B$ to the layer. Then, we have: $\mathbf{Z}  _1,\mathbf{Z}  _2,\cdots, \mathbf{Z}  _B = \mathrm{BatchNorm}(\mathbf{Y}  _1,\mathbf{Y}  _2,\cdots,\mathbf{Y}  _B; \gamma,\beta)$.
> > Suppose: $\widehat{\mathbf{Y}} = \frac{[\mathbf{Y}  _1,\mathbf{Y}  _2,\cdots,\mathbf{Y}  _B] - \overline{\mathbf{Y}}}{\sqrt{\mathrm{Var}(\mathbf{Y}  _1,\mathbf{Y}  _2,\cdots,\mathbf{Y}  _B)  + \epsilon} }$  where $\overline{\mathbf{Y}}$ is the batch mean. Then, we have:  $\mathbf{Z}  _1,\mathbf{Z}  _2,\cdots, \mathbf{Z}  _B = \widehat{\mathbf{Y}} \odot\gamma+\beta$. Now, suppose $\theta\to\Gamma  _{\pi} (\theta)$ gives $\mathbf{Y} \to \mathbf{Y}\pi$. This would give $\widehat{\mathbf{Y}} \to \widehat{\mathbf{Y}} \pi$. Suppose, we now transform $\gamma \to \gamma \pi$ and $\beta\to \beta\pi$. Then, $\mathbf{Z}  _1,\mathbf{Z}  _2,\cdots, \mathbf{Z}  _B \to \mathbf{\widehat{Y}} \pi \odot (\gamma \pi) + \beta\pi = (\widehat{\mathbf{Y}}\odot \gamma+\beta) \pi = \mathbf{Z}  _1\pi,\mathbf{Z}  _2\pi,\cdots, \mathbf{Z}  _B\pi$.
> >
> > The same holds for layer or feature norms too.
> >
> > Hence, $\Gamma'  _{\pi} (\theta\cup\{\gamma,\beta\}) = (\Gamma  _{\pi}(\theta), \gamma \pi, \beta\pi)$. Therefore, Lemma 2 holds true even when we apply Batch norm or Layer norm on each layer/feature. Since Lemma 2 is used to prove Lemma 3, 4 and these lemmas are used to prove the final result in Theorem 5, our results of exchangeability remain the same, regardless of normalization layer.
> >
> > ### Optimizers
> >
> > Adaptive optimizers do not violate exchangeability either. In appendix E.1.5, we have provided several optimizers, including even Adagrad and RMSprop, which shows that our results also hold good for a wide variety of optimizers. Lemma 11 captures a large class of optimizers which capture Adam, SGD, Adagrad and RMSprop.
> >
> > > Do the padding nodes used to equalize graph sizes introduce additional noise or bias that may negatively affect retrieval performance?
> >
> > Padding nodes do **not** introduce heuristic noise or bias. Instead, they allow us to express node insertions and deletions in a fully permutation-equivariant manner.
> > As shown in paper[X], suppose $p$ is the permutation map corresponding to $P$. Suppose $\nu _q[u] = 1$ if  $u$ is a valid node in $G _q$ (i.e., not padded). Then $\mathrm{ReLU}(\nu _q[u] - \nu  _c[p(u)]) > 0$ if $u$ is valid node in  $G _q$ and $p(u)$ is not valid node  in $G _c$. This implies that node $u$ is deleted from query to corpus graph. If n _+ is node addition and n _- is node deletion cost; e _+ edge addition and e _+ is edge deletion cost, unequal cost GED
> >
> > $\min _{P }
> >     \frac{e _-}{2}\,
> >     \left\|\mathrm{ReLU} \left(A  _q - PA  _cP^\top\right)\right\| _{1,1}+
> >     \frac{e _+}{2}\,
> >     \left\|\mathrm{ReLU} \left(PA  _c P^\top - A  _q\right)\right\| _{1,1}
> >     +  n _-\,\left\|\mathrm{ReLU} \left(\nu _q - P\nu _c\right)\right\|
> >     +\  n _+\left\|\mathrm{ReLU} \left(P\nu _q - \nu _c\right)\right\|.$
> >
> > Note that indicator of padded nodes $(1-\nu)$ is used to express the deletion or addition of nodes, which are crucial for computing the relevance distance. Hence, zero padding  does *not* add noise. It encodes legitimate edit operations and allows us to faithfully model GED and retrieval relevance even when graphs differ in size.
> >
> >  [X] Graph Edit Distance with General Costs Using Neural Set Divergence, NeurIPS 2024.
> >
> > > How does the proposed GraphHash framework handle graphs with different numbers of nodes, especially when a query graph contains more nodes than any graph seen during training?
> >
> > GraphHash can naturally support query graphs that are larger than any graph seen during training. Given a query graph with $n$ nodes (possibly much larger than in training) and node-embedding dimension $D$, the GNN encoder first produces node embeddings $X_q \in \mathbb{R}^{n \times D}$.  Since GNN parameters are shared across nodes, this step is size-agnostic and applies to arbitrarily large graphs.  Also note that the node embedding dimension $D$ remains unchanged for larger graphs. Subsequently, it computes a Fourier map t_d.
> >
> > The final hash is obtained by projecting each $t_d$ using a random hyperplane $W$ with iid entries. If we encounter query graph with more nodes, we can extend it to any dimensionality simply by drawing more iid samples. Thus, all components of GraphHash (GNN $\rightarrow$ per-dimension processing $\rightarrow$ pointwise Fourier map $\rightarrow$ random hyperplanes) can operate on variable-length node sets, allowing seamless handling of query graphs larger than those seen during training.

---

> ### Author Response · Authors · 2025-11-21
> **Response to Reviewer UMLi (3/4)**
>
> > The choice of baselines is limited
>
> There are two challenges in any graph search or any information retrieval system:
>
> (1) The number of graphs in the corpus is typically very large, making exhaustive scoring at query time infeasible. This necessitates an efficient indexing mechanism that can quickly filter out the vast majority of irrelevant graphs.
>
> (2) The system must approximate the true similarity between a query graph and candidate corpus graphs. Methods such as Isonet, Isonet++ fall into this category: they provide refined similarity scores once a manageable set of candidates is obtained.
>
> Modern search pipelines address these challenges through a retriever–reranker architecture: (1) The retriever performs fast quick filtering. (2) The reranker evaluates a small candidate set using a more precise (and more expensive) scoring function.
>
> Our contribution lies in the first stage. GraphHash aims to efficiently prune the corpus and enable scalable search. In contrast, Isonet++ is a scoring function and operate independently of the retriever. They are orthogonal and can be plugged on top of any retrieval system, including ours. Moreover, Isonet++ is an early interaction model, where we cannot even compute the query embeddings independently of the paired graph corpus graph and therefore, it does not at all support indexing.
> Nevertheless, we experimented with Isonet++ and  observed that inference time of Isonet++ per query is 32.52s, whereas our method takes 3.54s.
>
> We found that CorGII [B] (suggested by Rev 3FEM) acts as a retriever like us. Hence, we compare our method against it. We measure the area under the trade off curve between MAP and efficiency. Following numbers show that we perform better.
>
> ||GraphHash|CorGII|
> |-|-|-|
> |cox2|0.392|0.302|
> |ptc-fm|0.354|0.314|
> |ptc-fr|0.350|0.306|
> |ptc-mr|0.352|0.231|
>
> We also worked with SWWL[C] (suggested by reviweer 3FEM). From SWWL, we extact the SW encoding of the graph embedding which are used to compute the score. We exhaustively score each pair, then pick the top-k. Thus this is equivalent to an idealised setting with a perfect retrieval mechanism; any further indexing in a practical setting will further degrade performance.
>
> We present the area under the trade off curve between MAP and efficiency (AUC) for subgraph matching (SM) and GED. Following numbers show that we perform better.
>
> ||GraphHash|SWWL|
> |-|-|-|
> |cox2 (SM)|0.392|0.027|
> |ptc-fm (SM)|0.354|0.023|
> |ptc-fr (SM)|0.350|0.024|
> |ptc-mr (SM)|0.352|0.023|
> |cox2 (GED)|0.392|0.218|
> |ptc-fm (GED)|0.354|0.273|
> |ptc-fr (GED)|0.350|0.302|
> |ptc-mr (GED)|0.352|0.316|
>
> As the SW similarity is symmetric, it does not capture our assymmetric score function, which results in in poor perforance especially for subgraph matching.
>
> [B] Contextual Tokenization for Graph Inverted Indices, Chakraborty et al. 2025.
> [C] Gaussian process regression with Sliced Wasserstein Weisfeiler-Lehman graph kernels, Perez et al. 2024.
> > .. “the expected embedding matrix collapses to a rank-one matrix.” ..  empirical evidence ..  to substantiate this claim.
>
> In the following experiment we compute the fraction of the top most singular value $\frac{\sigma _1 ^2}{\sum _i{\sigma _i ^2}}$ of the mean graph embedding matrix over multiple training runs. We vary the number of samples computed for the sample mean as (n) to observe the trend. We observe that this fraction converges to 1 with large number samples, which indicate that the rank of the embedding matrix is 1.
>
> |number of samples|cox2 (SM)|cox2 (GED)|
> |-|-|-|
> |1|0.871|0.897|
> |4|0.937|0.903|
> |16|0.980|0.897|
> |32|0.989|0.909|
> |64|0.996|0.909|
> |128|0.997|0.909|
> |256|0.9988|0.894|
> |512|0.99947|0.920|
> |1024|0.99988|0.941|
> |2048|0.99987|0.930|
> |4096|0.99994|0.935|
> > hyperparameter sensitivity analysis.
>
> We analyze sensitivity with three hyperparameters: Embedding dimension D,
> hashcode dimension dim_h and Fourier feature dimension dim_T.
>
> **Variation with D:**
> In the following, we vary $D$ and present results for
>
> (1) MeanErr=$\frac{1}{D}|\sum _d\mathrm{sim} _d(G _c,G _q) -  \mathrm{sim}(G _c,G _q)|$.
>
> (2) Err(d)=$|\mathrm{sim} _d(G _c,G _q) - \frac{1}{D}\mathrm{sim}(G _c,G _q)|$ for an arbitrary $d$, we set $d=1$.
>
> Cox2 (Subgraph Matching)
> |D|MeanErr|Err(d)|
> |-|-|-|
> |2|1.19|1.16|
> |5|1.07|1.07|
> |10|1.17|1.18|
> |30|0.69|0.67|
>
> PTC-FM  (Subgraph Matching)
> |D|MeanErr|Err(d)|
> |-|-|-|
> |2|1.08|1.05|
> |5|1.35|1.38|
> |10|1.21|1.22|
> |30|0.88|0.87|
>
> As $D$ increases,  the difference between the Euclidean similarity and transportation based similarity decreases, thereby improving the quality of approximation.
> We further measure the quality of Eucledian similarity for different $D$ in terms of the trade off between MAP and efficiency  for subgraph isomorphism task. We present the area under the curve (AUC) as follows.
>
> |Dataset|D=5 |D=6|D=7|D=8|D=9|D=10|
> |-|-|-|-|-|-|-|
> |PTC-FM|0.331|0.337|0.346|0.347|0.355|0.354|
> |COX2|0.374|0.377|0.384|0.382|0.385|0.392|
>
> We note that as $D$ increases, AUC increases.

---

> ### Author Response · Authors · 2025-11-21
> **Response to Reviewer UMLi (4/4)**
>
> > [Contd] hyperparameter sensitivity analysis.
>
> **Variation with dim_h**
> We vary dim_h (number of hash bits) and obtain different the trade off curve between MAP and #no of retrieved graphs. We present the area under the curve (AUC)  for different values of dim_h.
>
>
> | dim of h | cox2 (SM) | cox2 (Uneq cost GED) | ptc-fm (SM) | ptc-fm (Uneq cost GED) |
> |-|-|-|-|-|
> | 1| 0.028| 0.028| 0.025| 0.071|
> | 5| 0.348| 0.209| 0.322| 0.275|
> | 10 | 0.366| 0.230| 0.331| 0.290|
> | 20 | 0.348| 0.231| 0.300| 0.287|
> | 30 | 0.342| 0.231| 0.303| 0.259|
> | 40 | 0.343| 0.231| 0.281| 0.243|
> | 50 | 0.342| 0.231| 0.259| 0.208|
> | 60 | 0.337| 0.229| 0.258| 0.205|
>
> We observe that dim_h =10 is a sweet spot, where we obtain the best trade off.
>
>
>
> **Variation with dim_T**
> Next we obtain AUC of the trade off curve against different values of dim${} _T$ (Fourier feature dimension) as follows. We parameterize dim${} _T$ as $=4nM$, where $n$ is the graph size.
>
> |M| cox2 (SM) | cox2 (Uneq cost GED) | ptc-fm (SM) | ptc-fm (Uneq cost GED) |
> |-|-|-|-|-|
> | 2|0.283|0.202|0.260|0.207|
> | 4|0.285|0.168|0.250|0.209|
> | 6|0.303|0.203|0.261|0.195|
> | 8|0.291|0.167|0.287|0.194|
> | 9|0.304|0.192|0.293|0.227|
> |10|0.392|0.244|0.354|0.305|
>
>
> We observe that there is a sharp decrease in performance as we reduce $M$ from $10$.
>
>
> > The experimental validation of exchangeability is weak.
>
> Yes, exchangeability concerns the joint distribution of the embedding dimensions. However, the source of randomness that induces this distribution is the random initialization of model parameters. A standard consequence of exchangeability of an embedding vector $X$ is that all coordinates $X[:,1], X[:,2], \ldots, X[:, D]$  must share the same marginal distribution.
>
> To probe this necessary condition, we train the model under multiple random initializations, extract the embeddings, and compare the empirical marginal distributions (via histograms) of $X[:,1], X[:,2], \ldots$. Because exchangeability requires identical marginals, the fact that the empirical distributions of $X[:,1]$ and $X[:,2]$ (and others) are nearly indistinguishable provides evidence consistent with exchangeability.
>
> We emphasize that this is not intended as a direct test of full exchangeability. Direct test of full exchangeability requires computation of differences between $p _X$ (joint distribution of $X[1],...,X[D]$) and $p _{X\pi}$ for all possible permutations $\pi$, which is computationally intensive. Therefore, we resorted to this necessary condition check.
>
> * if the marginals had differed noticeably across dimensions, that would have immediately ruled out exchangeability;
> * the observed similarity supports--- but does not conclusively prove--- the presence of exchangeability.
>
> This experiment therefore serves as a sanity check that the embedding dimensions do not exhibit asymmetries inconsistent with exchangeability.
>
> To perform a direct test, we compute the Maximum Mean Discrepancy statistics between $p_X$ and $p_{X\pi}$ ($p_{X}$ is the joint distribution of $X$). We draw samples from $p_X$ and $p_{X\pi}$ and estimate MMD^2 between these distributions as follows:
> $\widehat{MMD^2}(S,S')= T1+T2-2T3$
> where: $T1=[\sum _{x,y\in S}K(x,y)-\sum _{x\in S} K(x,x)]/(|S|(|S|-1))$,
> $T2=[\sum _{x',y'\in S'} K(x',y')-\sum _{x'} K(x',x')]/(|S'|(|S'|-1))$,
> $T3= \sum _{x\in S, x' \in S'} K(x,x')/(|S||S'|)$ and $S \sim p_X$ and $S' \sim p _{X \pi}$.
>
> For the kernel we use the standard setting of an RBF kernel with scale set to the median dispersion of the data.
>
>
> We sample 100 different permutations and compute the estimator of MMD^2 for each, and report the mean and standard deviation across the 100 observations. **Note that sample estimator of MMD^2 can be negative**. Following results show that the MMD values extremely small and therefore  $p_X$ and $p_{X\pi}$ are close.
>
>
> || $\widehat{MMD^2}$| StdDev |
> |-|-|-|
> |Cox2 + GED| -3.89e-5|   2.69e-05  |
> |Cox2 + SM|-1.18e-06|   3.28e-05 |
>
>
> Randomness from other sources e.g., droput or SGD does not conflate with exchangeability, since they are indepdenent and the key assumptions of exchangeability remained the same.

---

> > ### Comment · Reviewer_UMLi · 2025-11-26
> > **Thanks for your rebuttal**
> >
> > Most of my concerns have been addressed.
> > I have two minor questions:
> > 1. In the hyperparameter analysis, why does the performance under different values of $dim_h$ never reach the best performance achieved under varying $M$?
> > 2. Since the analysis of $M$ only includes values less than 10, how can the conclusion “a sharp decrease in performance as we reduce M from 10” be justified?

---

> ### Author Response · Authors · 2025-11-26
> **Reply to reviewer UMLi**
>
> Many thanks for your reply.
>
>
> > In the hyperparameter analysis, why does the performance under different values of $dim_h$
>  never reach the best performance achieved under varying $M$?
>
>
>
> The parameter $dim _h$ controls the number of hash bits. Together with the number of trials, it acts as a knob that allows one to move along the operating trajectory of the MAP-efficiency trade-off curve. For a desired operating point, a user can choose an appropriate value of $dim _h$, thereby achieving the required MAP at a chosen efficiency level. Hence, for any upper bound $H$, one should ideally vary $dim _h$ over the full range $[1, H]$ (along with the number of trials) to obtain the complete trade-off curve. This is what we have done for all methods in our main comparison in our paper.
>
> However, in the earlier experiment analyzing the effect of $dim _h$, we fixed $dim _h$ to a single value and varied only the number of trials. This produced only a partial set of trade-off points and therefore underestimated the overall AUC. We acknowledge that this presentation could be confusing and thank the reviewer for raising the question.
>
> To address this, we now report the AUC obtained by varying $dim  _h$ across the full range $[1, H]$. As $H$ increases, the resulting AUC values converge to those reported earlier for experiments varying $M$. Because the scatter plots become increasingly dense for large $H$, sometimes minor decreases in AUC can occur due to cluttered or overlapping operating points.
>
>
>  | $H$  $(dim _h \le H)$   |   cox2 (SM) |   cox2 (Uneq. cost GED) |   ptc-fm (SM) |   ptc-fm (Uneq. cost GED) |
> |--------------:|----------------:|-----------------:|------------------:|-------------------:|
> |             1 |          0.028 |           0.028  |            0.025 |             0.071 |
> |             5 |          0.355 |           0.213 |            0.329  |             0.277 |
> |            10 |          0.390 |           0.239 |            0.362  |             0.298  |
> |            20 |          0.391 |           0.242  |            0.360 |             0.306 |
> |            30 |          0.391 |           0.244 |            0.356 |             0.306 |
> |            40 |          0.392 |           0.244  |            0.354 |             0.306 |
> |            50 |          0.392 |           0.244  |            0.354 |             0.306 |
> |            60 |          0.392 |           0.244 |            0.354 |             0.305 |
>
>
>
> > Since the analysis of $M$ only includes values less than 10, how can the conclusion “a sharp decrease in performance as we reduce M from 10” be justified?
>
> We understand your concern. We should have written that:
>
> We experimented with $M=1,2,...,10$ and at among these values, at $M=10$, the AUC is the highest. Increase in $M$ may increase performance, but index size will also increase and therefore, we did not explore $M>10$.
>
>
> Please let us know if you still have any further concerns.

---

### Official Review · Reviewer_knCv · 2025-10-26

**Soundness:** 4
**Presentation:** 4
**Contribution:** 3
**Rating:** 8
**Confidence:** 2

**Summary:**

The paper introduces and proves a probabilistic symmetry in trained GNN node embeddings: under standard i.i.d. layer-wise initializations and across a broad class of architectures, the embedding dimensions behave as exchangeable random variables. Experiments validate this claim by showing near-identical per-dimension marginals across massive independently trained models. Leveraging this insight, the authors propose new graph retrieval methods based on order-statistic features, achieving competitive performance compared to prior work.

**Strengths:**

- The paper introduces an interesting perspective for understanding GNN representations and proposes a series of sensible applications that leverage this new finding.
- The reduction from permutation-equivariant training dynamics to exchangeability of embeddings is clear and broadly applicable under the paper’s assumptions.
-  The transition from transport similarity to Fourier features and ultimately to random-hyperplane LSH is elegant and well-justified, making the paper logically coherent throughout.
- Training with many random seeds and demonstrating that per-dimension marginals remain stable provides direct and compelling empirical support for the theoretical claim.

**Weaknesses:**

- The guarantees rely on permutation-invariant losses. Common heads (e.g., linear classifiers, feature-wise norms) can break this unless parameters are aligned under permutations; the practical scope should be specified more sharply.

- While the embedding dimensions are exchangeable, this does not imply independence—identical marginals do not preclude cross-coordinate dependencies. The proposed 1-D order-statistic surrogate could ignore the joint structure of the embeddings. Without quantifying inter-dimensional dependence (e.g., via covariance or effective rank) or demonstrating robustness to such dependencies, it remains unclear when this 1-D compression faithfully preserves information critical for retrieval.

**Questions:**

How do retrieval quality and the Proposition-7 approximation error scale with $D$? Can you report MAP vs $D$ and discuss the smallest $D$ where the 1-D surrogate is reliable?

---

> ### Author Response · Authors · 2025-11-21
> **Response to Reviewer knCv (1/3)**
>
> We thank the reviewer for their suggestions, which will help improve our paper. We have incorporated these discussions in the paper. Sec 5.1 and Appendix H.1 contain the experiments on exchangeability. Sec 5.2 and Appendix H.2.4 onwards contain the experiments on graph retrieval. Section E.1.6 contains additional discussions about theoretical setup.
>
>
> > The guarantees rely on permutation-invariant losses. the practical scope should be specified more sharply.
>
> We imposed a few simplifying assumptions only for brevity. In fact, our exchangeability results continue to hold even when these conditions are not explicitly met, including architectures that incorporate more complex operations such as normalization layers.
>
>
> ### Permutation invariance
>
> Even when we consider settings where the loss is not permutation invariant  (for example, a node classification task), exchangeable 'representations' exist within the middle of the network rather than at the end.
>
> We may partition such network into two parts--- the embedding network, and the classifier/prediction head. We may write $\mathbf{X} = \mathrm{NN}(G)$ where we refer to $\mathbf{X}$ as the embeddings and  $\hat{\mathbf{y}} = \mathrm{Clf}(\mathbf{X})$ where $\hat{\mathbf{y}}$ is the prediction label vector across nodes.  We can characterize and prove the exchangeability of $\mathbf{X}$ for this setting.
>
> Let the parameters of the entire network at $t$ timesteps be represented by $\theta   = (\theta  _{\mathrm{NN} },\theta  _{\mathrm{Clf} })$, coresponding to the parameters of either network. Let us also define the permutation inducing transformation as $\Gamma  _{\pi} = \Gamma  _{\mathrm{NN},\pi} \otimes \Gamma  _{\mathrm{Clf}, \pi}$, i.e. $\Gamma  _{\pi} (\theta) = (\Gamma  _{\mathrm{NN}, \pi}(\theta  _{\mathrm{NN}}),\Gamma  _{\mathrm{Clf}, \pi}(\theta  _{\mathrm{Clf}}))$. Given the dataset, we may reparameterise the loss function as $\mathcal{L}(\mathbf{X},\mathrm{Clf})$, or equivalently, $\mathcal{L}(\mathbf{X},\theta  _{\mathrm{Clf}})$.
>
> Given any permutation $\pi$, suppose there exists $\Gamma  _{\mathrm{NN},\pi},\Gamma  _{\mathrm{Clf},\pi}$ such that
> - $\mathbf{X}\mapsto\mathbf{X}\mathbf{\pi}$ under $\Gamma  _{\mathrm{NN}, \pi}$, and
> - the loss is invariant under $(\pi,\Gamma  _{\mathrm{Clf},\pi})$, i.e. $\mathcal{L}(\mathbf{X},\theta  _\mathrm{Clf}) = \mathcal{L}(\mathbf{X}\mathbf{\pi},\Gamma  _{\mathrm{Clf},\pi}(\theta  _\mathrm{Clf}))$
>
> Under these conditions, exchangeability follows with the same steps - exchangeability at initialisation, equivariance of gradient, equivariance of update step.
>
> To illustrate this, consider a three class classification task with a single layer for both $\mathrm{NN}$ and $\mathrm{Clf}$. Let the input feature be $\mathbf{feat}$. Let us focus on one channel/node of $\mathbf{X}$ denoted as $\mathbf{x} = \mathbf{X}[:,\bullet]$ and $\hat{\mathbf{y}}[\bullet] = y$. We have: $\mathbf{x} = \mathrm{NN}(\mathbf{feat}) = \sigma(\mathbf{feat}\mathbf{\Theta}  _{\mathrm{NN}})$. Hence, we will have:
> $\hat{y} = \mathrm{Softmax}\left([(\mathbf{x} \cdot\mathbf{w}  _1),(\mathbf{x}\cdot\mathbf{w}  _2),(\mathbf{x}\cdot\mathbf{w}  _3)]\right)$.
>
> The transformation $\Gamma  _{\text{NN},\pi}$ can then represented as, $\Theta  _{\mathrm{NN}} \mapsto \Theta  _{\mathrm{NN}}\mathbf{\pi}$ and $[\mathbf{w}  _1, \mathbf{w}  _2, \mathbf{w}  _3] \mapsto [\mathbf{\pi}^\top\mathbf{w}  _1, \mathbf{\pi}^\top\mathbf{w}  _2, \mathbf{\pi}^\top\mathbf{w}  _3]$. Under this transformation $\mathbf{x}\mapsto\mathbf{x}\mathbf{\pi}$ but $\hat{y}$ remains invariant---therefore, the loss is invariant.
>
> Hence, our results remain valid even when the loss itself is not permutation-invariant. This is because such losses may still exhibit invariance under a joint transformation consisting of (i) a permutation of intermediate representations; and, (ii) a corresponding permutation-induced transformation of the parameters in the subsequent layer.
>
>
> ### Layer normalizations:
>
> Batch norm, layer norm, etc.  do not break exchanegability condition. If the network without the norm layers can be shown to give exchangeable embeddings, the same will hold for the embeddings for the network with batch norm or layer norm.
>
> We denote a normalization layer as $NL  _{\gamma,\beta}$, where $\gamma$ and $\beta$ are parameters. Such layers allow us to extend permutation inducing transformation $\Gamma  _{\pi}$ to $\Gamma'  _{\pi}$. For simplicity, assume that the normalization layer $NL  _{\gamma,\beta}$ is applied on one layer $\ell$. Suppose, $\theta\to\Gamma  _{\pi} (\theta)$ gives $\mathbf{Z}\to \mathbf{Z} \pi$ in that $\ell$ layer (where $\mathbf{Z}\in \mathbb{R}^{n\times dim  _z}$).
> Then we can obtain a transformation $\Gamma'  _{\pi}$ such that  $\theta\cup\{\gamma,\beta\}\to\Gamma'  _{\pi} (\theta\cup\{\gamma,\beta\})$ will also give  $\mathbf{Z}\to \mathbf{Z} \pi$.

---

> ### Author Response · Authors · 2025-11-21
> **Response to Reviewer knCv (2/3)**
>
> ### Layer normalizations [Contd.]
>
> **Consider batch norm first**. Let the batch of inputs be $G  _1,G  _2,\cdots,G  _B$ and a single batch norm layer, with the corresponding inputs $\mathbf{Y}  _1,\mathbf{Y}  _2,\cdots,\mathbf{Y}  _B$ to the layer. Then, we have: $\mathbf{Z}  _1,\mathbf{Z}  _2,\cdots, \mathbf{Z}  _B = \mathrm{BatchNorm}(\mathbf{Y}  _1,\mathbf{Y}  _2,\cdots,\mathbf{Y}  _B; \gamma,\beta)$.
> Suppose: $\widehat{\mathbf{Y}} = \frac{[\mathbf{Y}  _1,\mathbf{Y}  _2,\cdots,\mathbf{Y}  _B] - \overline{\mathbf{Y}}}{\sqrt{\mathrm{Var}(\mathbf{Y}  _1,\mathbf{Y}  _2,\cdots,\mathbf{Y}  _B)  + \epsilon} }$  where $\overline{\mathbf{Y}}$ is the batch mean. Then, we have:  $\mathbf{Z}  _1,\mathbf{Z}  _2,\cdots, \mathbf{Z}  _B = \widehat{\mathbf{Y}} \odot\gamma+\beta$. Now, suppose $\theta\to\Gamma  _{\pi} (\theta)$ gives $\mathbf{Y} \to \mathbf{Y}\pi$. This would give $\widehat{\mathbf{Y}} \to \widehat{\mathbf{Y}} \pi$. Suppose, we now transform $\gamma \to \gamma \pi$ and $\beta\to \beta\pi$. Then, $\mathbf{Z}  _1,\mathbf{Z}  _2,\cdots, \mathbf{Z}  _B \to \mathbf{\widehat{Y}} \pi \odot (\gamma \pi) + \beta\pi = (\widehat{\mathbf{Y}}\odot \gamma+\beta) \pi = \mathbf{Z}  _1\pi,\mathbf{Z}  _2\pi,\cdots, \mathbf{Z}  _B\pi$.
>
> Consider layer norm. Assume  the corresponding input is $\mathbf{y}$ and output in one channel is $\mathbf{z} = \mathrm{LayerNorm}(\mathbf{y};  \gamma , \beta )$.  Suppose: $\widehat{\mathbf{y}} = \frac{\mathbf{y}- y \mathbf{1}}{\sqrt{\mathrm{Var}(\mathbf{y})+ \epsilon} }$  where $y$ is the feature mean. Then, we have:  $\mathbf{z}= \widehat{\mathbf{y}}\odot \gamma+\beta$. Now, suppose $\theta\to\Gamma  _{\pi} (\theta)$ gives $\mathbf{y} \to \mathbf{y}\pi.$  This would give $\widehat{\mathbf{y}} \to \widehat{\mathbf{y}} \pi$. Suppose, we now transform $\gamma \to \gamma \pi$ and $\beta\to \beta\pi$. Then, $\mathbf{z} \to \mathbf{\widehat{y}}\odot(\gamma \pi) + \beta\pi = \mathbf{z}\pi$.
>
> Hence, $\Gamma'  _{\pi} (\theta\cup\{\gamma,\beta\}) = (\Gamma  _{\pi}(\theta), \gamma \pi, \beta\pi)$. Therefore, Lemma 2 holds true even when we apply Batch norm or Layer norm on each layer/feature. Since Lemma 2 is used to prove Lemma 3, 4 and these lemmas are used to prove the final result in Theorem 5, our results of exchangeability remain the same, regardless of normalization layer.
>
> ### Optimizers
>
>
>
> Adaptive optimizers too do not violate exchangeability. In our appendix E.1.5, we have provided several optimizers including even Adagrad and RMSprop, which shows that our results also hold good for a wide variety of optimizers. Lemma 11 captures a large class of optimizers which capture Adam, SGD, Adagrad and RMSprop.
>
>
>
> > it remains unclear when this 1-D compression faithfully preserves information critical for retrieval.
>
> Indeed, our embeddings do exhibit inter-dimensional dependence. Nevertheless, the proposed surrogate for similarity remains capable of accurately approximating the transportation-based similarity, given the form of their expressions.
>
> We note that the embeddings remain uniformly bounded even as $D$ increases, because each coordinate encodes information from a fixed $K$-hop neighbourhood. Hence, increasing $D$ refines the representation but does not increase its magnitude.
>
> Let $Z  _d[u,u'] = s\big(x  _q(u) - x  _c(u')\big)[d]$.
> While bounding
> $Pr \left(\left|\frac{\mathrm{sim}(G  _c,G  _q)}{D} - \mathrm{sim}  _d(G  _c,G  _q)\right| \ge \epsilon \right)$, we had to bound
> $\sum  _{u,u'} \frac{\mathrm{Var}\left(\sum  _d Z  _d[u,u']\right)}{D^2 \epsilon^2}$ (Appendix E.2.1).
> If the dimensions were independent, we would have $\mathrm{Var}(\sum  _d Z  _d) = O(D)$, making $\mathrm{Var}(\sum  _d Z  _d)/D^2 = O(1/D)$, so concentration improves with larger $D$ for iid samples.
>
> Indeed in our setting, the dimensions are dependent, yet due to the $L  _2$-boundedness of the embeddings, we still obtain the same $O(1/D)$ behaviour. Define $\bar Z  _d = Z  _d - \mathbb{E}(Z  _d)$. Then
>
>
> \begin{align}
> \frac{\mathrm{Var}\left(\sum  _d Z  _d[u,u']\right)}{D^2}
> &= \frac{\sum  _{d,d'} \mathbb{E}\big(\bar Z  _d[u,u'] \, \bar Z  _{d'}[u,u']\big)}{D^2} \\\\
> &\le \frac{\left(\sum  _d |\bar Z  _d[u,u']|\right)\left(\sum  _{d'} |\bar Z  _{d'}[u,u']|\right)}{D^2} \\\\
> &= \frac{\|\bar{\mathbf{Z}}[u,u']\|  _1^2}{D^2} \\\\
> &\le \frac{D \|\bar{\mathbf{Z}}[u,u']\|  _2^2}{D^2} \\\\
> & = \frac{\|\bar{\mathbf{Z}}[u,u']\|  _2^2}{D}. \tag{A}
> \end{align}
>
>
> Since the embeddings are $L  _2$-bounded, $\|\bar{\mathbf{Z}}[u,u']\|  _2^2 = O(1)$, and therefore $\mathrm{Var}\left(\sum  _d Z  _d[u,u']\right)/D^2 = O(1/D)$. Thus, even with interdependent dimensions, the concentration inequality remains valid, and the surrogate similarity continues to converge to the transportation-based similarity as $D$ increases.

---

> > ### Author Response · Authors · 2025-11-21
> > **Response to Reviewer knCv (3/3)**
> >
> > > How do retrieval quality and the Proposition-7 approximation error scale with D? Can you report MAP vs  D?
> >
> > When $D=1$, the one-dimensional surrogate exactly aligns with the transportation-based similarity. However, as $D$ increases, the variance across dimensions and the emerging cross-coordinate correlations begin to influence the surrogate, causing deviations from the ideal transportation similarity. But as $D$ gradually increases, $||\bar{\pmb{Z}} [u,u']||  _2 ^2/D$ in Eq. (A) (response to previous concern) decreases and makes the concentration tigher.
> >
> >
> >  In the following, we vary $D$ and present results for
> >
> >
> > (1) MeanErr=$\frac{1}{D}|\sum  _d\mathrm{sim}  _d(G  _c,G  _q) -  \mathrm{sim}(G  _c,G  _q)|$.
> >
> >
> > (2) Err(d)=$|\mathrm{sim}  _d(G  _c,G  _q) - \frac{1}{D}\mathrm{sim}(G  _c,G  _q)|$ for an arbitrary $d$, we set $d=1$.
> >
> >
> >
> > Cox2 (Subgraph Matching)
> > |   D | MeanErr|Err(d)|
> > |-|-|-|
> > |2|1.19|1.16|
> > |5|1.07|1.07|
> > |10|1.17|1.18|
> > |15|1.01|0.99|
> > |20|0.85|0.85|
> > |30|0.69|0.67|
> >
> >
> > As $D$ increases, we observe that the difference between the Euclidean similarity and transportation based similarity decreases, thereby improving the quality of approximation.
> >
> >
> >
> >
> > We further measure the quality of Eucledian similarity for different $D$ in terms of the trade off between MAP and efficiency  for subgraph isomorphism task. We present the area under the curve (AUC) as follows.
> >
> >
> >
> > |Dataset|D=5 |D=6|D=7 |D=8|D=9|D=10 |
> >  |-|-|-|-|-|-|-|
> > |PTC-FM|0.331|0.337|0.346|0.347|0.355|0.354|
> > |COX2|0.374|0.377|0.384|0.382|0.385|0.392|
> >
> > We observe that as $D$ increases, AUC increases.

---

> > > ### Comment · Reviewer_knCv · 2025-11-24
> > >
> > > Thanks for the clarification and additional result. I believe this is an exciting paper that meets the standards of ICLR, and I will maintain my positive score.

---

> > > > ### Author Response · Authors · 2025-11-28
> > > >
> > > > Thank you for encouraging comments . Please note that we already included all the results in the paper too.

---

### Official Review · Reviewer_3FEM · 2025-10-31

**Soundness:** 2
**Presentation:** 3
**Contribution:** 2
**Rating:** 4
**Confidence:** 3

**Summary:**

The paper formalizes *embedding-dimension exchangeability* for GNNs under specific conditions (i.i.d. initialization, permutation-invariant loss, and equivariant optimization) and leverages this property to approximate high-dimensional transport-based graph similarity via per-dimension sorted one-dimensional matching.

This leads to the **GraphHash** framework, which applies Fourier feature mapping and LSH for efficient graph retrieval.

Experiments on small TU-style molecular datasets (PTC-FM/FR/MR, COX2) show favorable accuracy–efficiency trade-offs compared with several baselines.

**Strengths:**

1. **Novel theoretical formulation — Exchangeability of GNN embeddings.**
   The paper introduces the concept of *embedding-dimension exchangeability* in GNNs, showing that under mild assumptions, different embedding dimensions can be treated as exchangeable random variables.
   This extends classical symmetry analyses (e.g., permutation invariance) into the probabilistic domain and provides a new theoretical lens for understanding GNN representations.

2. **Principled link between theory and computational efficiency.**
   Using the exchangeability property, the authors reduce the high-dimensional optimal transport (OT) distance to a sum of one-dimensional sorted distances.
   This theoretically justified simplification avoids solving full OT, achieving a significant computational reduction.

3. **Unified and efficient retrieval framework.**
   The proposed **GraphHash** system combines this OT-based distance with locality-sensitive hashing (LSH), providing a unified, efficient, and theoretically grounded approach to graph retrieval.

4. **Strong empirical performance.**
   On four TU benchmark datasets (PTC-FR/FM/MR, COX2), GraphHash consistently outperforms several strong baselines (FourierHashNet, DiskANN, IVF) in both *Subgraph Matching (SM)* and *Graph Edit Distance (GED)* tasks, achieving higher MAP and NDCG scores.

5. **Good scalability and trade-off behavior.**
   The method achieves a favorable accuracy–efficiency balance, maintaining more than 50% of exhaustive MAP performance with much lower retrieval cost.

6. **Reproducibility and clarity.**
   The theoretical derivations are detailed, and the authors provide code, data splits, and reproducibility documentation.

**Weaknesses:**

## **Weaknesses and Limitations**

1. **Idealized theoretical assumptions.**
   The exchangeability theorem relies on assumptions rarely satisfied in modern GNNs. Components such as normalization layers (BatchNorm, LayerNorm), multi-head attention, temperature scaling, dropout, and adaptive optimizers can break permutation equivariance. The paper does not analyze which architectural choices preserve or violate exchangeability, limiting the generality of the theorem.

2. **Zero-padding heuristic lacks theoretical justification.**
   When graphs have different node counts, the smaller graph is zero-padded before per-dimension sorting and matching. This can distort the node-value distribution and create spurious matches, especially for graphs with large node-count disparities. No theoretical bound or comparison with **Unbalanced/Partial OT** is provided, leaving the robustness of this approximation unclear.

3. **Weak statistical validation of exchangeability.**
   The empirical evidence mainly consists of qualitative plots. There are no formal statistical tests (e.g., KS, MMD, or energy distance) or analyses of inter-dimensional dependence (correlation, mutual information). It remains unclear whether dimensions are merely identically distributed or also weakly independent—an important distinction for the proposed matching approach.

4. **Limited task scope despite a broadly applicable distance.**
   While related works also focus on retrieval, the proposed transport-based distance could naturally extend to other tasks (e.g., k-NN classification, clustering, or graph alignment). Evaluating at least one additional downstream task would strengthen claims of broader applicability.

5. **Missing comparisons with the strongest recent methods.**
   The experiments lack comparisons with several contemporary state-of-the-art systems, such as **IsoNet++**, **CORGII**, and recent **sliced-Wasserstein/set-hashing** approaches like **SLoSH** and **SWWL**. Without these baselines, it is difficult to assess how GraphHash performs relative to the current best.

6. **Small embedding dimensionality in the chosen datasets.**
   All experiments use low-dimensional (≈10D) node embeddings on small molecular graphs. The exchangeability assumption and 1D matching approximation may not hold as well in higher-dimensional spaces where inter-dimensional dependencies are richer. A sensitivity analysis with respect to embedding dimensionality would help assess generality.

**Questions:**

1. **Zero-padding and simplification of the matching process**
   The current approach simplifies node alignment by zero-padding smaller graphs before per-dimension sorting and matching. Could the authors provide more discussion or ablation on this choice? It seems that this step simplifies the transport computation considerably, but it is unclear how much accuracy or theoretical rigor is sacrificed as a result. A deeper justification or empirical analysis would make the approach more convincing.

2. **Extension to other downstream tasks**
   While the paper focuses on graph retrieval, the proposed transport-based similarity measure appears more general. Would it be possible to show at least one additional downstream task (e.g., k-NN classification, clustering, or graph alignment) to demonstrate broader applicability and potential beyond retrieval?

3. **Dataset scale and comparison completeness**
   The current datasets (PTC-FR/FM/MR, COX2) are relatively small in both node count and embedding dimensionality. Could the authors evaluate the method on larger or more complex datasets to test its scalability? In addition, many strong contemporary baselines (e.g., IsoNet++, CORGII, SLoSH, SWWL) are missing. Including such comparisons would help non-specialist reviewers better assess the actual competitiveness and validity of the proposed approach.

4. **Optional clarifications related to weaknesses**
   Some of the weaknesses mentioned (e.g., idealized assumptions, limited statistical validation of exchangeability) might be addressed by additional experiments or discussion. The authors may selectively respond to these points if relevant.

---

**General comment:**
If the authors can convincingly address these concerns—especially providing more insight into the zero-padding mechanism, exploring at least one additional task, and extending experiments to larger or more challenging datasets—I would be happy to raise my score. A simple but effective idea, if rigorously justified, can indeed make for a strong and impactful contribution.

---

> ### Author Response · Authors · 2025-11-21
> **Response to Reviewer 3FEM (1/4)**
>
> We thank the reviewer for their suggestions, which will help improve our paper. We have incorporated these discussions in the paper. Sec 5.1 and Appendix H.1 contain the experiments on exchangeability. Sec 5.2 and Appendix H.2.4 onwards contain the experiments on graph retrieval. Section E.1.6 contains additional discussions about theoretical setup.
>
>
>
> > Idealized theoretical assumptions.
>  Components such as normalization layers (BatchNorm, LayerNorm), ... dropout, and adaptive optimizers can break permutation equivariance.
>
> Batch norm, layer Norm, adaptive optimizers, dropout do not break exchanegability condition. If the network without the norm layers can be shown to give exchangeable embeddings, the same will hold for the embeddings for the network with batch norm, layer norm, adaptive optimizers, dropout etc.
>
> We denote a normalization layer as $NL _{\gamma,\beta}$, where $\gamma$ and $\beta$ are parameters. Such layers allow us to extend permutation inducing transformation $\Gamma _{\pi}$ to $\Gamma' _{\pi}$. For simplicity, assume that the normalization layer $NL _{\gamma,\beta}$ is applied on one layer $\ell$. Suppose, $\theta\to\Gamma _{\pi} (\theta)$ gives $\mathbf{Z}\to \mathbf{Z} \pi$ in that $\ell$ layer (where $\mathbf{Z}\in \mathbb{R}^{n\times dim _z}$).
> Then we can obtain a transformation $\Gamma' _{\pi}$ such that  $\theta\cup\{\gamma,\beta\}\to\Gamma' _{\pi} (\theta\cup\{\gamma,\beta\})$ will also give  $\mathbf{Z}\to \mathbf{Z} \pi$.
>
>
> Consider batch norm first. Let the batch of inputs be $G _1,G _2,\cdots,G _B$ and a single batch norm layer, with the corresponding inputs $\mathbf{Y} _1,\mathbf{Y} _2,\cdots,\mathbf{Y} _B$ to the layer. Then, we have: $\mathbf{Z} _1,\mathbf{Z} _2,\cdots, \mathbf{Z} _B = \mathrm{BatchNorm}(\mathbf{Y} _1,\mathbf{Y} _2,\cdots,\mathbf{Y} _B; \mathbf{\gamma},\mathbf{\beta})$.
> Suppose: $\widehat{\mathbf{Y}} = \frac{[\mathbf{Y} _1,\mathbf{Y} _2,\cdots,\mathbf{Y} _B] - \overline{\mathbf{Y}}}{\sqrt{\mathrm{Var}(\mathbf{Y} _1,\mathbf{Y} _2,\cdots,\mathbf{Y} _B)  + \epsilon} }$  where $\overline{\mathbf{Y}}$ is the batch mean. Then, we have:  $\mathbf{Z} _1,\mathbf{Z} _2,\cdots, \mathbf{Z} _B = \widehat{\mathbf{Y}} \odot\mathbf{\gamma}+\mathbf{\beta}$. Now, suppose $\theta\to\Gamma _{\pi} (\theta)$ gives $\mathbf{Y} \to \mathbf{Y}\pi$. This would give $\widehat{\mathbf{Y}} \to \widehat{\mathbf{Y}} \pi$. Suppose, we now transform $\gamma \to \gamma \pi$ and $\beta\to \beta\pi$. Then, $\mathbf{Z} _1,\mathbf{Z} _2,\cdots, \mathbf{Z} _B \to \mathbf{\widehat{Y}} \pi \odot (\gamma \pi) + \beta\pi = (\widehat{\mathbf{Y}}\odot \mathbf{\gamma}+\mathbf{\beta}) \pi = \mathbf{Z} _1\pi,\mathbf{Z} _2\pi,\cdots, \mathbf{Z} _B\pi$.
>
> Consider layer norm. Assume  the corresponding input is $\mathbf{y}$ and output in one channel is $\mathbf{z} = \mathrm{LayerNorm}(\mathbf{y};  \gamma , \beta )$.  Suppose: $\widehat{\mathbf{y}} = \frac{\mathbf{y}- y \mathbf{1}}{\sqrt{\mathrm{Var}(\mathbf{y})+ \epsilon} }$  where $y$ is the feature mean. Then, we have:  $\mathbf{z}= \widehat{\mathbf{y}}\odot \mathbf{\gamma}+\mathbf{\beta}$. Now, suppose $\theta\to\Gamma _{\pi} (\theta)$ gives $\mathbf{y} \to \mathbf{y}\pi.$  This would give $\widehat{\mathbf{y}} \to \widehat{\mathbf{y}} \pi$. Suppose, we now transform $\gamma \to \gamma \pi$ and $\beta\to \beta\pi$. Then, $\mathbf{z} \to \mathbf{\widehat{y}}\odot(\gamma \pi) + \beta\pi = \mathbf{z}\pi$.
>
> Hence, $\Gamma' _{\pi} (\theta\cup\{\gamma,\beta\}) = (\Gamma _{\pi}(\theta), \gamma \pi, \beta\pi)$. Therefore, Lemma 2 holds true even when we apply Batch norm or Layer norm on each layer/feature. Since Lemma 2 is used to prove Lemma 3, 4 and these lemmas are used to prove the final result in Theorem 5, our results of exchangeability remain the same, regardles of normalization layer.
>
> Note that, dropout induces randomization which is independent from the randomization during initialization. Since dropout process sparsifies connection in an iid manner, it does not affect the exchangeability.
>
> Adaptive optimizers too do not violate exchangeability. In our appendix E.1.5, we have provided several optimizers including even Adagrad and RMSprop, which shows that our results also hold good for a wide variety of optimizers. Lemma 11 captures a large class of optimizers which capture Adam, SGD, Adagrad and RMSprop.

---

> > ### Author Response · Authors · 2025-11-21
> > **Response to Reviewer 3FEM (2/4)**
> >
> > ### Scope of our work beyond permutation invariance:
> >
> > When we consider settings where the loss is not permutation invariant, for example a node classification task, the 'representations' exist within the middle of the network rather than at the end and such representations are exchangeable.
> >
> > We may partition such network into two parts--- the embedding network, and the classifier/prediction head. We may write $\mathbf{X} = \mathrm{NN}(G)$ where we refer to $\mathbf{X}$ as the embeddings and  $\hat{\mathbf{y}} = \mathrm{Clf}(\mathbf{X})$ where $\hat{\mathbf{y}}$ is the prediction label vector across nodes.  We can characterize and prove the exchangeability of $\mathbf{X}$ for this setting.
> >
> > Let the parameters of the entire network at $t$ timesteps be represented by $\theta   = (\theta  _{\mathrm{NN} },\theta  _{\mathrm{Clf} })$, coresponding to the parameters of either network. Let us also define the permutation inducing transformation as $\Gamma  _{\pi} = \Gamma  _{\mathrm{NN},\pi} \otimes \Gamma  _{\mathrm{Clf}, \pi}$, i.e. $\Gamma  _{\pi} (\theta) = (\Gamma  _{\mathrm{NN}, \pi}(\theta  _{\mathrm{NN}}),\Gamma  _{\mathrm{Clf}, \pi}(\theta  _{\mathrm{Clf}}))$. Given the dataset, we may reparameterise the loss function as $\mathcal{L}(\mathbf{X},\mathrm{Clf})$, or equivalently, $\mathcal{L}(\mathbf{X},\theta  _{\mathrm{Clf}})$.
> >
> > Given any permutation $\pi$, if there exists $\Gamma  _{\mathrm{NN},\pi},\Gamma  _{\mathrm{Clf},\pi}$ such that
> > - $\mathbf{X}\mapsto\mathbf{X}\mathbf{\pi}$ under $\Gamma  _{\mathrm{NN}, \pi}$, and
> > - the loss is invariant under $(\pi,\Gamma  _{\mathrm{Clf},\pi})$, i.e. $\mathcal{L}(\mathbf{X},\theta  _\mathrm{Clf}) = \mathcal{L}(\mathbf{X}\mathbf{\pi},\Gamma  _{\mathrm{Clf},\pi}(\theta  _\mathrm{Clf}))$
> >
> > Under these conditions, exchangeability follows with the same steps - exchangeability at initialisation, equivariance of gradient, equivariance of update step.
> >
> > To illustrate this, consider a three class classification task with a single layer for both $\mathrm{NN}$ and $\mathrm{Clf}$. Let the input feature be $\mathbf{feat}$. Let us focus on one channel/node of $\mathbf{X}$ denoted as $\mathbf{x} = \mathbf{X}[:,\bullet]$ and $\hat{\mathbf{y}}[\bullet] = y$. We have: $\mathbf{x} = \mathrm{NN}(\mathbf{feat}) = \sigma(\mathbf{feat}\mathbf{\Theta}  _{\mathrm{NN}})$. Hence, we will have:
> > $\hat{y} = \mathrm{Softmax}\left([(\mathbf{x} \cdot\mathbf{w}  _1)\, \,(\mathbf{x}\cdot\mathbf{w}  _2)\,(\mathbf{x}\cdot\mathbf{w}  _3)]\right)$.
> >
> > The transformation $\Gamma  _{\text{NN},\pi}$ can then represented as, $\Theta  _{\mathrm{NN}} \mapsto \Theta  _{\mathrm{NN}}\mathbf{\pi}$ and $[\mathbf{w}  _1, \mathbf{w}  _2, \mathbf{w}  _3] \mapsto [\mathbf{\pi}^\top\mathbf{w}  _1, \mathbf{\pi}^\top\mathbf{w}  _2, \mathbf{\pi}^\top\mathbf{w}  _3]$. Under this transformation $\mathbf{x}\mapsto\mathbf{x}\mathbf{\pi}$ but $\hat{y}$ remains invariant---therefore, the loss is invariant.
> >
> > Hence, our results remain valid even when the loss itself is not permutation-invariant. This is because such losses may still exhibit invariance under a joint transformation consisting of a permutation of intermediate representations together with a corresponding permutation-induced transformation of the parameters.

---

> ### Author Response · Authors · 2025-11-21
> **Response to Reviewer 3FEM (3/4)**
>
> > Zero-padding heuristic lacks theoretical justification.
>
> Padding nodes do **not** introduce heuristic noise or bias. Instead, they allow us to express node insertions and deletions in a fully permutation-equivariant manner.
> As shown in paper[X], suppose $p$ is the permutation map corresponding to $P$. Suppose $\nu _q[u] = 1$ if  $u$ is a valid node in $G _q$ (i.e., not padded). Then $\mathrm{ReLU}(\nu _q[u] - \nu  _c[p(u)]) > 0$ if $u$ is valid node in  $G _q$ and $p(u)$ is not valid node  in $G _c$. This implies that node $u$ is deleted from query to corpus graph. If n _+ is node addition and n _- is node deletion cost; e _+ edge addition and e _+ is edge deletion cost, unequal cost GED
>
> $\min _{P }
>     \frac{e _-}{2}\,
>     \left\|\mathrm{ReLU} \left(A  _q - PA  _cP^\top\right)\right\| _{1,1}+
>     \frac{e _+}{2}\,
>     \left\|\mathrm{ReLU} \left(PA  _c P^\top - A  _q\right)\right\| _{1,1}
>     +  n _-\,\left\|\mathrm{ReLU} \left(\nu _q - P\nu _c\right)\right\|
>     +\  n _+\left\|\mathrm{ReLU} \left(P\nu _q - \nu _c\right)\right\|.$
>
> Note that indicator of padded nodes $(1-\nu)$ is used to express the deletion or addition of nodes, which are crucial for computing the relevance distance. Hence, zero padding  does *not* add noise. It encodes legitimate edit operations and allows us to faithfully model GED and retrieval relevance even when graphs differ in size.
>
>  [X] Graph Edit Distance with General Costs Using Neural Set Divergence, NeurIPS 2024.
>
>
> > Weak statistical validation of exchangeability.
>
>
>
>
>
>
> We compute the Maximum Mean Discrepancy statistics between $p _X$ and $p _{X\pi}$ ($p _{X}$ is the joint distribution of $X$). We draw samples from $p _X$ and $p _{X\pi}$ and estimate MMD^2 between these distributions as follows:
> $\widehat{MMD^2}(S,S')= T1+T2-2T3$
> where: $T1=[\sum _{x,y\in S}K(x,y)-\sum _{x\in S} K(x,x)]/(|S|(|S|-1))$,
> $T2=[\sum _{x',y'\in S'} K(x',y')-\sum _{x'} K(x',x')]/(|S'|(|S'|-1))$,
> $T3= \sum _{x\in S, x' \in S'} K(x,x')/(|S||S'|)$ and $S \sim p _X$ and $S' \sim p _{X \pi}$.
>
> For the kernel we use the standard setting of an RBF kernel with scale set to the median dispersion of the data.
>
>
> We sample 100 different permutations and compute the estimator of MMD^2 for each permutation, and report the mean and standard deviation across the 100 observations. Note that sample estimator of MMD^2 can be negative. Following results show that the MMD values extremely small and therefore  $p _X$ and $p _{X\pi}$ are close.
>
>
> || $\widehat{MMD^2}$| StdDev |
> |-|-|-|
> |Cox2 + GED| -3.89e-5|   2.69e-05  |
> |Cox2 + SM|-1.18e-06|   3.28e-05 |
>
>
> > Limited task scope despite a broadly applicable distance... Evaluating at least one additional downstream task would strengthen claims of broader applicability.
>
> We apply our method for graph alignment.
> We compute MAP* for transportation-based similarity and MAP for approximate similarity and compute the % accuracy loss in terms of (MAP*-MAP)/MAP*.
>
> |Dataset| (MAP*-MAP)/MAP* (in percentage)|
> |-|-|
> |ptc-fr|8.44\%|
> |ptc-fm|8.18\%|
> |ptc-mr|6.73|
> |cox2|11.89\%|
>
> Furthermore, we show that the true alignment distance our approximate distanace are highly correlated.
>
> |Dataset|CorrCoeff|
> |-|-|
> |cox2(SM)|0.632|
> |cox2(GED)|0.665|
> |ptc-fr(SM)|0.575|
> |ptc-fr(GED)|0.810|
> |ptc-mr(SM)|0.631|
> |ptc-mr(GED)|0.828|
> |ptc-fm(SM)|0.704|
> |ptc-fm(GED)|0.800|
>
>
> > Dataset scale: extending experiments to larger or more challenging datasets
>
>
> We increase the number of nodes to 2x.  Following tables shows the efficiency in terms of  fraction of graphs retrieved to achieve atleast a certain MAP=$m^*$ expressed as a percentage of the exhaustive MAP for PTC-FR (first table) and Cox2 (second table) datasets for subgraph matching task. Lower k/C is better and indicates higher efficiency.
>
> Efficiency for PTC FR  (SM)
> |MAP$^*$|Graphhash|Fhash|RH|DiskANN|IVF|
> |-|-|-|-|-|-|
> |60\%|**0.39**|0.49|0.61|0.82|0.80|
> |70\%|**0.55**|0.60|0.73|0.82|0.80|
> |80\%|**0.63**|0.94|0.80|0.82|0.80|
>
> Efficiency for Cox2 (SM)
> |MAP$^*$|Graphhash|Fhash|RH|DiskANN|IVF|
> |-|-|-|-|-|-
> |60%|0.38|**0.35**|0.62|0.78|0.78|
> |70%|**0.48**|0.51|0.69|0.78|0.78|
> |80%|**0.60**|0.69|0.80|0.78|0.78|
>
> We observe that our method achieves the same MAP with greater efficiency in majority of the cases.
>
> Next, we evaluate our method on a large *number* of corpus graphs (|C|=1M).  We report AUC of trade of curve between MAP and efficiency as follows.  It shows that our method is better in most cases.
>
>
> |Dataset|Graphhash|Fhash|RH|IVF|
> |-|-|-|-|-|
> | cox2 (SM) |  **0.361** |  0.332 |   0.213 | 0.274 |
> | cox2 (GED) |      **0.230** | 0.222 |   0.190 |  0.154 |
> | ptc-fm (SM)  |      **0.347** |0.322 | 0.161 |   0.216 |
> | ptc-fm (GED) | **0.284** |0.270 |   0.231 |    0.186 |
> | ptc-fr (SM)  | **0.333** |0.317 |   0.157 |   0.217 |
> | ptc-fr (GED) |  **0.284** |0.270 |   0.231 | 0.186 |
> | ptc-mr (SM)  |  **0.337** |0.288 |   0.122 |      0.205 |
> | ptc-mr (GED) |0.320 |**0.339** |   0.256 |      0.200 |

---

> ### Author Response · Authors · 2025-11-21
> **Response to Reviewer 3FEM (4/4)**
>
> > Missing comparisons with the strongest recent methods:
>
>
>
> There are two challenges in any graph search or any information retrieval system:
>
> (1) The number of graphs in the corpus is typically very large, making exhaustive scoring at query time infeasible. This necessitates an efficient indexing mechanism that can quickly filter out the vast majority of irrelevant graphs.
>
> (2)  The system must approximate the true similarity between a query graph and candidate corpus graphs. Methods such as Isonet, Isonet++, and SWWL fall into this category: they provide refined similarity scores once a manageable set of candidates is obtained.
>
> Modern search pipelines address these challenges through a retriever–reranker architecture: (1) The retriever performs fast quick filtering. (2) The reranker evaluates a small candidate set using a more precise (and more expensive) scoring function.
>
> Our contribution lies in the first stage. Corgii too works in the first stage. Both Corgii and Graphhash aim to efficiently prune the corpus and enable scalable search.
> In contrast, Isonet++ and SWWL are scoring functions and operate independently of the retriever. They are orthogonal and can be plugged on top of any retrieval system, including ours.
>
>
> Therefore, we first compare our method against CORGII in terms of area under the trade off curve between MAP and efficiency. Following numbers show that we perform better.
>
> |Dataset|GraphHash|CorGII|
> |-|-|-|
> |cox2|0.392|0.302|
> |ptc-fm|0.354|0.314|
> |ptc-fr|0.350|0.306|
> |ptc-mr|0.352|0.231|
>
>
> From SWWL, we extact the SW encoding of the graph embedding which are used to compute the score. We exhaustively score each pair, then pick the top-k. Thus this is equivalent to an idealised setting with a perfect retrieval mechanism; any further indexing in a practical setting will further degrade performance.
>
> We present the area under the trade off curve between MAP and efficiency (AUC). Following numbers show that we perform better.
>
> |Dataset|GraphHash|SWWL|
> |-|-|-|
> |cox2 (SM)|0.392|0.027|
> |ptc-fm (SM)|0.354|0.023|
> |ptc-fr (SM)|0.350|0.024    |
> |ptc-mr (SM)|0.352|0.023|
> |cox2 (GED)|0.392|0.218|
> |ptc-fm (GED)|0.354|0.273|
> |ptc-fr (GED)|0.350|0.302|
> |ptc-mr (GED)|0.352|0.316|
>
> As the SW similarity is symmetric, it does not capture our assymmetric score function, which results in in poor perforance especially for subgraph matching.
>
> Isonet++ is an early interaction model, where we cannot even compute the query embeddings indepdently of the paired graph corpus graph and therefore, it does not support indexing. For each computation of query corpus pair it takes $32.52s$, whereas our method takes $3.54s$.
>
>
>
> >  A sensitivity analysis with respect to embedding dimensionality would help assess generality.
>
>  In the following, we vary $D$ and present results for
>
>
> (1) MeanErr=$\frac{1}{D}|\sum _d\mathrm{sim} _d(G _c,G _q) -  \mathrm{sim}(G _c,G _q)|$.
>
>
> (2) Err(d)=$|\mathrm{sim} _d(G _c,G _q) - \frac{1}{D}\mathrm{sim}(G _c,G _q)|$ for an arbitrary $d$, we set $d=1$.
>
>
>
> Cox2 (Subgraph Matching)
> |   D | MeanErr|Err(d)|
> |-|-|-|
> |2|1.19|1.16|
> |5|1.07|1.07|
> |10|1.17|1.18|
> |30|0.69|0.67|
>
>
> PTC _FM  (Subgraph Matching)
> |D|MeanErr|Err(d)|
> |-|-|-|
> | 2  | 1.08 | 1.05 |
> | 5  | 1.35 | 1.38 |
> | 10 | 1.21 | 1.22 |
> | 30 | 0.88 | 0.87 |
>
> As $D$ increases, we observe that the difference between
> the Euclidean similarity and transportation based similarity decreases, thereby improving the quality of approximation.
>
>
>
>
> We further measure the quality of Eucledian similarity for different $D$ in terms of the trade off between MAP and efficiency  for subgraph isomorphism task. We present the area under the curve (AUC) as follows.
>
>
>
> |Dataset|GraphHash, D=5 |GraphHash, D=6|GraphHash, D=7 |GraphHash, D=8|GraphHash, D=9| GraphHash D=10 |
>  |-|-|-|-|-|-|-|
> |PTC-FM|0.331|0.337|0.346|0.347|0.355|0.354|
> |COX2|0.374|0.377|0.384|0.382|0.385|0.392|
>
> We observe that as $D$ increases, AUC increases and around D=10, it becomes stable.

---

> > ### Author Response · Authors · 2025-11-27
> >
> > Dear Reviewer,
> >
> >
> > Thank you for your review. We were wondering if you could kindly let us know whether our rebuttal has addressed your concerns. This would allow us to have sufficient time to respond in case there are any remaining issues.
> >
> >
> > Thanks,
> >
> > Authors

---

> > ### Comment · Reviewer_3FEM · 2025-11-27
> > **Thanks for your rebuttal**
> >
> > The authors have provided a comprehensive rebuttal that effectively addresses my main concerns.
> >
> > Given the novel theoretical contribution (exchangeability) and the now-strengthened empirical evaluation, I am raising my score.

---

> > > ### Author Response · Authors · 2025-11-28
> > >
> > > Thank you for encouraging comments and increasing the score. Please note that we already included all the results in the paper too.

---

### Official Review · Reviewer_nx4Q · 2025-10-31

**Soundness:** 4
**Presentation:** 4
**Contribution:** 3
**Rating:** 6
**Confidence:** 2

**Summary:**

The paper introduces a novel approach, GraphHash, addressing the graph retrieval problem. It demonstrates that the node embedding dimensions for a wide class of GNN methods are invariant under permutation operations applied along the dimension axis. This property is referred to as exchangeability, and the authors design a locality-sensitive hashing (LSH) framework by leveraging this property in order to efficiently approximate transportation-based graph similarities. The proposed method is supported by theoretical analysis and evaluated on four benchmark datasets for the graph retrieval task.

**Strengths:**

- The paper is well-structured and clearly presented, so the technical content is easy to follow.
- It provides a rigorous theoretical analysis supporting the proposed framework and its underlying assumptions.
- It includes an experimental evaluation demonstrating the effectiveness of the method across multiple datasets.

**Weaknesses:**

- The experimental evaluation is limited to relatively small benchmark datasets and lacks large-scale or real-world scenarios.
- The paper focuses primarily on retrieval performance but does not examine the memory and computational requirements.

**Questions:**

- Does the method require a fixed number of nodes per graph? Since the hash function depends on sorting node embeddings, how is hashing handled if graphs undergo updates that change the embedding order?
- What is the computational and space complexity of the proposed method? and how does it scale with graph size and embedding dimensionality compared to existing approaches?
- Based on Proposition 7, how does varying $D$ influence the performance? Similarly, what is the impact of $dim_h$ and $dim_T$ for different values?
- Since the hashing relies on random Fourier features and random hyperplanes, how stable is the performance across different seeds? Could the authors report variance for different runs?
- Could the authors elaborate on how sensitive the exchangeability property is to violations of the assumptions, such as non-i.i.d. initialization?
- It would be helpful to include the exact (non-approximate) transportation-based similarity in the comparison to better quantify the accuracy loss.
- In Figure 2, it is difficult to distinguish the performances clearly due to overlapping points. It would be better to summarize retrieval performance using a single aggregate score, such as area under the curve or a related metric.

**Additional comments:**
- In Line 163, it could be nice to mention that $\Psi$ and $ \Theta$ indicate the weight matrices in order to avoid the confusion that $\Psi$ is an input feature.
- Line 1034, “invovled” -> involved

---

> ### Author Response · Authors · 2025-11-21
> **Response to Reviewer nx4Q (1/3)**
>
> We thank the reviewer for their suggestions. Sec 5.1 and Appendix H.1 now contain more experiments on exchangeability. Sec 5.2 and Appendix H.2.4 onwards contain more experiments on graph retrieval. Section E.1.6 contains additional discussions about theoretical setup.
>
> > fixed number of nodes per graph?  how is hashing handled if change the embedding order?
>
> We only require an upper bound on the number of nodes in a graph. As we show next, if we change the order of node embeddings $X$, the distribution of new hashcode does not change.
>
> We sort the node embeddings to obtain $v=\text{sort}(X)$ and apply a Fourier map $\mathcal{F}$ to obtain $t=\mathcal{F}(v)$. Next, we multiply a random hyperplane W on $t$ to obtain the hash code $h = sign(Wt)$.  Suppose, $X$ changes its order, then $v$ undergoes some changes in its order.  Since $\mathcal{F}$ is applied pointwise, $t$ is equivariant to the permutation of $v$. Hence, if for some permutation $P$, the transformation $v\to Pv$ implies $t\to Pt$.
>
> Now, $W[i,j]$ are i.i.d. random variables--- therefore, $W$ and $WP$ have the same distribution. Hence, $Wt$ and $WPt$ will have the same distribution. Therefore, the density of $Wt$ remains invariant, if $t$ changes its order.
> Hence, even when the embeddings are sorted, performance of hashing methods do not change too much.
>
> > What is the computational and space complexity of the proposed method?
>
> We first analyze the time complexity.  Recall the number of nodes is $n$, the embedding dimension is $D$, number of corpus graphs is $|C|$.
>
> (1) Time complexity for computing node embeddings $X _q$ is $O(K|E|)$.
>
> (2) Time complexity for sorting the emebdding entries of $X _q$ in one dimension $d\in [D]$ is $O(n\log n)$.
>
> (3) Time complexity for Fourier feature computation ($\hat{T}  _ {q,d}$) for each query graph is $O(4nM)$.
>
> (4) The complexity of query time $O(|C|^{\gamma})$ where $\gamma$ is the efficiency factor in LSH, i.e., $\gamma = \log(1/p)/\log(1/ p') < 1$, where $p$ is the lower bound of hash-collision of high similarity regime and $p'$ is the upper bound of hash-collision of low similarity regime. Proof of Theorem 18 in Appendix contains exact specification of p and p'.
>
> Item (1) is computed in a tensorized manner (not sequentially for each dim of embeddings.). Note that the constant factor depends on the architecture).
> Items 2,3 can be executed on each dimension, independently of other dimensions. Therefore, these computations can be parallelized across dimensions. Hence the total time complexity is $O(K|E|) + O(n\log n) + O(4nM) + O(|C|^{\gamma})$, if we use full matrix operations in the first stage and subsequent parallelization in next stages.
>
> Next, we analyze space complexity. We keep $D$ indices, each for one LSH trial.  These indices can only keep the IDs of the corpus items against each hash bucket and trials. Hence, the space complexity is $O(D|C|)$.  In practice, our indices are quite lightweight: for 100K graphs,  it only takes 3.5M. Their size can be further reduced by applying compression techniques.
>
> >  Scaling with graph size and large dataset
>
> We increase the number of nodes to 2x.  Following tables shows the efficiency in terms of  fraction of graphs retrieved to achieve atleast a certain MAP=$m^*$ expressed as a percentage of the exhaustive MAP for subgraph matching task. Lower k/C  indicates higher efficiency.
>
> ptc-fr
> |MAP$^*$|Graphhash|Fhash|RH|DiskANN|IVF|
> |-|-|-|-|-|-|
> |60\%|**0.39**|0.49|0.61|0.82|0.80|
> |70\%|**0.55**|0.60|0.73|0.82|0.80|
> |80\%|**0.63**|0.94|0.80|0.82|0.80|
>
> Cox2
> |MAP$^*$|Graphhash|Fhash|RH|DiskANN|IVF|
> |-|-|-|-|-|-
> |60%|0.38|**0.35**|0.62|0.78|0.78|
> |70%|**0.48**|0.51|0.69|0.78|0.78|
> |80%|**0.60**|0.69|0.80|0.78|0.78|
>
> We observe that our method achieves the same MAP with greater efficiency in majority of the cases.
>
> Next, we evaluate our method on a large *number* of corpus graphs (|C|=1M).  As the reviewer suggested, we also report AUC of trade of curve between MAP and efficiency as follows.  It shows that our method is better in most cases.
>
> |Dataset|Graphhash|Fhash|RH|IVF|
> |-|-|-|-|-|
> |cox2 (SM)|**0.361**|0.332|0.213|0.274|
> |cox2 (GED)|**0.230**|0.222|0.190|0.154|
> |ptc-fm (SM)|**0.347**|0.322|0.161|0.216|
> |ptc-fm (GED)| **0.284**|0.270|0.231|0.186|
> |ptc-fr (SM)|**0.333**|0.317|0.157|0.217|
> |ptc-fr (GED)|**0.284**|0.270|0.231|0.186|
> |ptc-mr (SM)|**0.337**|0.288|0.122|0.205|
> |ptc-mr (GED)|0.320|**0.339**|0.256|0.200|
>
> > Variation with  embedding dimensionality
>
> We change embedding dim D and compute the area under the trade off curve (AUC, higher is better).
>
> ||our, D=5 |our, D=6|our, D=7 |our, D=8|our, D=9|GH, D=10 |Fhash|RH|IVF|DiskANN|
> |-|-|-|-|-|-|-|-|-|-|-|
> |PTC-FM|0.331|0.337|0.346|0.347|0.355|0.354|0.339|0.174|0.225|0.247|
> |COX2|0.374|0.377|0.384|0.382|0.385|0.392|0.344|0.229|0.285|0.289|
>
> We observe that our method performs better than baselines. As $D$ increases, the performance becomes better.

---

> ### Author Response · Authors · 2025-11-21
> **Response to Reviewer nx4Q (2/3)**
>
> > Based on Proposition 7, how does varying D influence the performance?
>
>  In the following, we vary $D$ and present results for
>
>
> (1) MeanErr=$\frac{1}{D}|\sum _d\mathrm{sim} _d(G _c,G _q) -  \mathrm{sim}(G _c,G _q)|$.
>
>
> (2) Err(d)=$|\mathrm{sim} _d(G _c,G _q) - \frac{1}{D}\mathrm{sim}(G _c,G _q)|$ for an arbitrary $d$, we set $d=1$.
>
>
>
> Cox2 (Subgraph Matching)
> |   D | MeanErr|Err(d)|
> |-|-|-|
> |2|1.19|1.16|
> |5|1.07|1.07|
> |10|1.17|1.18|
> |30|0.69|0.67|
>
>
> PTC_FM  (Subgraph Matching)
> |D|MeanErr|Err(d)|
> |-|-|-|
> | 2  | 1.08 | 1.05 |
> | 5  | 1.35 | 1.38 |
> | 10 | 1.21 | 1.22 |
> | 30 | 0.88 | 0.87 |
>
> As $D$ increases, we observe that the difference between the Euclidean similarity and transportation based similarity decreases, thereby improving the quality of approximation.
> We further measure the quality of Eucledian similarity for different $D$ in terms of the trade off between MAP and efficiency  for subgraph isomorphism task. We present the area under the curve (AUC) as follows.
>
>
>
> |Dataset|D=5 |D=6|D=7 |D=8|D=9| D=10 |
>  |-|-|-|-|-|-|-|
> |PTC-FM|0.331|0.337|0.346|0.347|0.355|0.354|
> |COX2|0.374|0.377|0.384|0.382|0.385|0.392|
>
> We observe that as $D$ increases, AUC increases.
>
>
> > Similarly, what is the impact of dim_h and dim_t for different values?
>
> dim_h indicates the number of hash bits. Hence, by changing dim_h, one can sweep through the trade off plot. Therefore, to generate the scatter plot for the trade-off, one should vary the value of dim_h (along with the number of trials).
> Here, we present the AUC over varying $\mathrm{dim}_h$, but considering all points with $\mathrm{dim} _h \leq H$.
> There are cluttered points in the scatter plot. Therefore, occasionally higher H may result in small decrease in AUC.
> We observe that AUC becomes stable after dim_h =20.
>
>  | $H$ such that $\mathrm{dim}_h \leq H$  |   cox2 (SM) |   cox2 (Uneq. cost GED) |   ptc-fm (SM) |   ptc-fm (Uneq. cost GED) |
> |-|-|-|-|-|
> | 1 |          0.028 |           0.028  |            0.025 |             0.071 |
> | 5 |          0.355 |           0.213 |            0.329  |             0.277 |
> | 10 |          0.390 |           0.239 |            0.362  |             0.298  |
> |20 |   0.391 |           0.242  |            0.360 |             0.306 |
> |30 |          0.391 |    0.244 |            0.356 |             0.306 |
> | 40 |          0.392 |           0.244  |            0.354 |             0.306 |
> |50 |          0.392 |           0.244  |            0.354 |             0.306 |
> |            60 |          0.392 |           0.244 |            0.354 |             0.305 |
>
>
> Next we obtain AUC of the trade off curve against different values of dim${} _T$ (Fourier feature dimension) as follows. We parameterize dim${} _T$ as $=4nM$, where $n$ is the graph size.
>
> |M| cox2 (SM) | cox2 (Uneq cost GED) | ptc-fm (SM) | ptc-fm (Uneq cost GED) |
> |-|-|-|-|-|
> | 2|0.283|0.202|0.260|0.207|
> | 4|0.285|0.168|0.250|0.209|
> | 6|0.303|0.203|0.261|0.195|
> | 8|0.291|0.167|0.287|0.194|
> | 9|0.304|0.192|0.293|0.227|
> |10|0.392|0.244|0.354|0.305|
>
>
> We observe that there is a sharp decrease in performance as we reduce $M$ from $10$.
>
> >  how stable is the performance across different seeds? Could the authors report variance for different runs?
>
> We report the mean AUC for trade off curve  for 10 runs, including std dev, for two datsets across both subgraph matching (SM) and unequqal cost GED prediction tasks.
>
> ||mean AUC|std dev|
> |-|-|-|
> ptc-fm (SM)|0.343|0.007
> cox2 (SM)|0.370|0.009
> ptc-fm (Uneq cost GED)|0.290|0.008
> cox2 (Uneq cost GED)|0.238|0.005
>
> We observe that the performance is stable across different runs.
>
>
>
> > It would be helpful to include the exact (non-approximate) transportation-based similarity in the comparison to better quantify the accuracy loss.
>
>
> We compute MAP* for transportation-based similarity and MAP for approximate similarity and compute the % accuracy loss in terms of (MAP*-MAP)/MAP*.
>
> |Dataset| (MAP*-MAP)/MAP* (in percentage)|
> |-|-|
> |ptc-fr|8.44\%|
> |ptc-fm|8.18\%|
> |ptc-mr|6.73|
> |cox2|11.89\%|
>
>
> We observe that the accuracy loss remains bounded by $12\%$.
>
>
>
> >  summarize retrieval performance using  area under the curve
>
>
> We report AUC  for Subgraph Matching based retrieval as follows:
>
> |method|ptc-fm|cox2|ptc_fr|ptc_mr|
> |-|-|-|-|-|
> |GraphHash|**0.354**|**0.392**|**0.350**|**0.352**|
> |FourierHashNet|0.339|0.344|0.330|0.299|
> |RH|0.152|0.212|0.171|0.133|
> |DiskANN|0.247|0.289|0.248|0.224|
> |IVF|0.225|0.285|0.230|0.208|
> |Random|0.234|0.269|0.230|0.228|
>
>
> We report AUC  for Unequal cost GED based retrieval as follows:
>
> |method|ptc-fm|cox2|ptc_fr|ptc_mr|
> |-|-|-|-|-|
> |GraphHash|**0.305**|**0.244**|**0.328**|0.339|
> |FourierHashNet|0.280|0.229|0.324|**0.348**|
> |RH|0.242|0.195|0.264|0.246|
> |DiskANN|0.153|0.118|0.167|0.167|
> |IVF|0.194|0.158|0.193|0.211|
> |Random|0.171|0.134|0.189|0.188|
>
>
> We observe in subgraph matching task, our method performs the best for all datasets. In unequal cost GED task, our method performs the best in all datasets except one.

---

> > ### Author Response · Authors · 2025-11-21
> > **Response to Reviewer nx4Q (3/3)**
> >
> > > Could the authors elaborate on how sensitive the exchangeability property is to violations of the assumptions,
> >
> > We imposed a few simplifying assumptions only for brevity. In fact, our exchangeability results continue to hold even when these conditions are not explicitly met.
> >
> >
> > ### Permutation invariance of loss function
> >
> > Even when we consider settings where the loss is not permutation invariant  (for example, a node classification task), exchangeable 'representations' exist within the middle of the network rather than at the end.
> >
> > We may partition such network into two parts--- the embedding network, and the classifier/prediction head. We may write $\mathbf{X} = \mathrm{NN}(G)$ where we refer to $\mathbf{X}$ as the embeddings and  $\hat{\mathbf{y}} = \mathrm{Clf}(\mathbf{X})$ where $\hat{\mathbf{y}}$ is the prediction label vector across nodes.  We can characterize and prove the exchangeability of $\mathbf{X}$ for this setting.
> >
> > Let the parameters of the entire network at $t$ timesteps be represented by $\theta   = (\theta  _{\mathrm{NN} },\theta  _{\mathrm{Clf} })$, coresponding to the parameters of either network. Let us also define the permutation inducing transformation as $\Gamma  _{\pi} = \Gamma  _{\mathrm{NN},\pi} \otimes \Gamma  _{\mathrm{Clf}, \pi}$, i.e. $\Gamma  _{\pi} (\theta) = (\Gamma  _{\mathrm{NN}, \pi}(\theta  _{\mathrm{NN}}),\Gamma  _{\mathrm{Clf}, \pi}(\theta  _{\mathrm{Clf}}))$. Given the dataset, we may reparameterise the loss function as $\mathcal{L}(\mathbf{X},\mathrm{Clf})$, or equivalently, $\mathcal{L}(\mathbf{X},\theta  _{\mathrm{Clf}})$.
> >
> > Given any permutation $\pi$, if there exists $\Gamma  _{\mathrm{NN},\pi},\Gamma  _{\mathrm{Clf},\pi}$ such that
> > - $\mathbf{X}\mapsto\mathbf{X}\mathbf{\pi}$ under $\Gamma  _{\mathrm{NN}, \pi}$, and
> > - the loss is invariant under $(\pi,\Gamma  _{\mathrm{Clf},\pi})$, i.e. $\mathcal{L}(\mathbf{X},\theta  _\mathrm{Clf}) = \mathcal{L}(\mathbf{X}\mathbf{\pi},\Gamma  _{\mathrm{Clf},\pi}(\theta  _\mathrm{Clf}))$
> >
> > Under these conditions, exchangeability follows with the same steps - exchangeability at initialisation, equivariance of gradient, equivariance of update step.
> >
> > To illustrate this, consider a three class classification task with a single layer for both $\mathrm{NN}$ and $\mathrm{Clf}$. Let the input feature be $\mathbf{feat}$. Let us focus on one channel/node of $\mathbf{X}$ denoted as $\mathbf{x} = \mathbf{X}[:,\bullet]$ and $\hat{\mathbf{y}}[\bullet] = y$. We have: $\mathbf{x} = \mathrm{NN}(\mathbf{feat}) = \sigma(\mathbf{feat}\mathbf{\Theta}  _{\mathrm{NN}})$. Hence, we will have:
> > $\hat{y} = \mathrm{Softmax}\left([(\mathbf{x} \cdot\mathbf{w}  _1)\, \,(\mathbf{x}\cdot\mathbf{w}  _2)\,(\mathbf{x}\cdot\mathbf{w}  _3)]\right)$.
> >
> > The transformation $\Gamma  _{\text{NN},\pi}$ can then represented as, $\Theta  _{\mathrm{NN}} \mapsto \Theta  _{\mathrm{NN}}\mathbf{\pi}$ and $[\mathbf{w}  _1, \mathbf{w}  _2, \mathbf{w}  _3] \mapsto [\mathbf{\pi}^\top\mathbf{w}  _1, \mathbf{\pi}^\top\mathbf{w}  _2, \mathbf{\pi}^\top\mathbf{w}  _3]$. Under this transformation $\mathbf{x}\mapsto\mathbf{x}\mathbf{\pi}$ but $\hat{y}$ remains invariant---therefore, the loss is invariant.
> >
> > Hence, our results remain valid even when the loss itself is not permutation-invariant. This is because such losses may still exhibit invariance under a joint transformation consisting of a permutation of intermediate representations together with a corresponding permutation-induced transformation of the parameters.
> >
> >
> > ### IID initialization
> >
> > IID initialization is not stricly required. As long as our initial model parameters are exchangeable, our results will hold true, even if the model parameters are dependent.

---

> > > ### Comment · Reviewer_nx4Q · 2025-11-27
> > > **Thanks for the clarification and additional experiments**
> > >
> > > I thank the authors for the additional experiments and for the comments addressing my concerns. I raised my score.

---

> > > > ### Author Response · Authors · 2025-11-28
> > > >
> > > > Dear Reviewer nx4Q,
> > > >
> > > > Thank you for increasing the score. Please note that we already included all the results in the paper too.

---

### Author Response · Authors · 2025-12-03
**Address to AC**

Dear AC,


Our paper shows that GNN node representations are exchangeable across embedding dimensions. Under mild conditions, if $X \in \mathbb{R}^{N \times d}$ is the node embedding matrix, then $p_X = p_{X\pi}$ for any permutation $\pi \in \\{0,1\\}^{d \times d}$ acting on the feature dimensions. This property has critical consequences for designing indexing and hashing techniques in the context of graph matching and retrieval. Due to exchangeability, the neural graph similarity expressed via Earth Mover Distance can be well-approximated by a Euclidean similarity between dimension-wise sorted node embeddings. This allows us to develop locality sensitive hashing (LSH) for graph retrieval.


Pre-rebuttal reviews were already positive (nx4Q=6, 3FEM=4, knCv=8, UMLi=6). We addressed all their questions/points in a comprehensive manner. Some of these key changes are listed below.

*  We theoretically show that our results extend well beyond the already mild assumptions (nX4Q, 3FEM, knCv and UMLi): They may work when the loss function is not permutation invariant and neural nets involve batch or feature normalization

* We reported on additional experiments, which involve hyperparameter sensitivity analysis (nX4Q, UMLi), comparison with additional baselines (3FEM, UMLi), experiments on large scale datasets (nX4Q, 3FEM), experiments with stronger statistical validation for exchangeability (3FEM, UMLi), extension to downstream task (3FEM), robustness across seeds (nX4Q), etc.

* We explained theoretically and empirically why inter-dimensional correlation of embeddings does not affect our analysis (knCv).


Other clarifications include space and time analysis, reasoning about zero padding etc.


Each reviewer acknowledged our rebuttal very positively. **nX4Q and 3FEM improved their scores from 6→8 and 4→ 6 respectively (Please refer to their response statement about increase in the score)**.  In their response to the rebuttal, knCv mentioned that this paper is exciting. UMLi clearly wrote most of their concerns were addressed and asked two minor questions, which we answered. We believe that UMLi would have responded again, but the discussion was prematurely terminated.

------

Thus, our paper’s ratings had moved to **8,6, 8, 6 (avg = 7)** from 6, 4, 8, 6. Unfortunately, the recent reversal brought it back to 6, 4, 8, 6 again.

 We stressed on the scores because

(1) a score of 7 might would probably place a paper well above the poster bracket in normal circumstances.

(2) our rebuttal comprehensively addressed all the concerns of reviewers, leading two of them  to raise their scores which reflect strong approval from the reviewers after rebuttal.

We kindly request that you ensure this post-rebuttal opinion is considered during the decision process.

While the message is primarily addressed to AC, we are keeping it visible to all reviewers for full transparency. Note that we have already obtained explicit approval from the PC chairs (via one-to-one email) to mention the post-rebuttal scores.




Best regards,

Authors

---

### Meta-Review · Area_Chair_QTdA · 2025-12-28

**Summary:**

In this paper, the authors show that graph neural networks (GNNs) exhibit certain exchangeability in their dimensionality, i.e., the node embeddings remain invariant with respect to permutations along the dimension axis. Based on this finding, the authors propose approximations so that Euclidean similarities can be applied, enabling locality-sensitive hashing (LSH) to graph tasks such as subgraph matching and graph edit distance.

All reviewers agree that the paper presents an interesting angle for GNNs, as existing methods typically focus on the permutation of node IDs. Main concerns are largely twofold: whether this finding holds in theory beyond the assumptions presented in the paper, and requiring more experimental verifications. The authors have presented extended discussions in the rebuttal, including more experiments and more theoretical explanations. All reviewers are satisfied with the rebuttal. Given the positive initial score and successful rebuttal, it is clear that the paper is above the bar for acceptance.

**Reviewer Concerns:**

Most concerns are addressed, particularly those involving more theoretical justifications and experiments.

**Reviewer Scores:**

For Reviewer nx4Q, the initial rating is 6, and it is likely to increase to 8.

For Reviewer 3FEM, the initial rating is 4, and it is likely to increase to 6.

For Reviewer knCv, the initial rating is 8, and it is likely to stay at 8.

For Reviewer UMLi, the initial rating is 6, and it is likely to stay at 6 or increase to 8.

---

### Decision · Program_Chairs · 2026-01-26

Accept (Oral)